# Predicting Ocean Waves along the U.S. East Coast During Energetic Winter Storms: sensitivity to Whitecapping parameterizations

5   Mohammad Nabi Allahdadi[1], Ruoying He[1], and Vincent S. Neary[2]

[1] North Carolina State University, Department of Marine, Earth, and Atmospheric Sciences, Raleigh, NC27695, USA

[2] Sandia National Laboratories, P.O. Box 5800, Albuquerque, NM 87185 - MS1124, USA

*Correspondence to*: Mohammad Nabi Allahdadi (mallahd@ncsu.edu)

**Abstract:** The performance of two methods for quantifying whitecapping dissipation incorporated in the SWAN wave model is evaluated for waves generated along and off the U.S. East Coast under energetic winter storms with a predominantly westerly wind. Parameterizing the whitecapping effect can be done using the Komen-type schemes, which are based on mean spectral parameters, or the saturation-based (SB) approach of van der Westhuysen (2007), which is based on local wave parameters and the saturation level concept of the wave spectrum (we use "Komen" and "Westhuysen" to denote these two approaches). Observations of wave parameters and frequency spectra at four NDBC buoys are used to evaluate simulation results. Model-data comparisons show that when using the default parameters in SWAN, both Komen and Westhuysen methods underestimate wave height. Simulations of mean wave period using the Komen method agree with observations, but those using the Westhuysen method are substantially lower. Examination of source terms shows that the Westhuysen method underestimates the total energy transferred into the wave action equations, especially in the lower frequency bands that contain higher spectral energy. Several causes for this underestimation are identified. The primary reason is the difference between the wave growth conditions along the East Coast during winter storms and the conditions used for the original whitecapping formula calibration. In addition, some deficiencies in simulation results are caused along the coast by the "slanting fetch" effect that adds low-frequency components to the 2-D wave spectra. These components cannot be simulated partly or entirely by available source terms (wind input, white capping, and quadruplet) in models and their interaction. Further, the effect of boundary layer instability that is not considered in the Komen and Westhuysen whitecapping wind input formulas may cause additional underestimation.

## 1 Introduction

Spectral wave models, including Simulating Wave Nearshore (SWAN) (SWAN, 2015), solve the equation for conservation of wave action density in the frequency-direction, spatial, and time domains. This equation considers the time variation of spectral energy over the specified geographic domain by considering the local rate of change and transport terms as well as source terms. The source term in the wave action density equation is the algebric sume of several terms as follows:

$$S_{tot} = S_{wind} + S_{ds,w} + S_{nl4} + S_{nl3} + S_{ds,b} + S_{ds,br} \tag{1}$$

The terms on the right side of the equation are wave growth by wind, wave decay due to whitecapping, nonlinear transfer of wave energy through four-wave (quadruplet) and three-wave (triad) interactions, bottom friction, and depth-induced wave breaking, respectively. Three-wave interaction (triad) as well as bottom friction and depth-induced wave breaking are specific energy source and sink terms for the shallow to the very shallow coastal water environment, whereas the first three terms actively contribute to wave energy development and spectral evolution in both open-ocean and coastal environments. Quantifying these source terms has been a challenging task and the focus of active research, especially for the wave decay processes associated with whitecapping.

The Komen-type methods for resolving whitecapping dissipation are of the most popular approaches in coastal modeling applications, and are based on the initial study by Hasselmann (1974), formulated by Komen (1984), and modified by Janssen et al. (1991). This approach represents dissipation of spectral energy as a function of mean spectral frequency and steepness. It is an appropriate approach for simulation of wave height as a result of generation and growth by local wind, and the default method for resolving whitecapping dissipation in SWAN and other popular spectral models like WAM and Mike21-SW. To achieve higher simulation accuracies for wave height and wave period, calibrations of the model for the whitecapping parameter and the wave period parameter delta are necessary (Allahdadi et al., 2017; Siadatmousavi et al., 2011; Niroomandi et al., 2018; Allahdadi et al., 2004). However, wave model applications for different regions (including open ocean and shelf waters) show that Komen-type methods tend to underestimate both peak and mean wave periods (SWAN, 2015; Vledder et al, 2016; and Siadatmousavi et al 2011). For the case of pure wind wave growth, this problem is the result of underestimating the spectral energy at low frequencies. In the presence of low-frequency swells with lower spectral steepness, this method contributes to higher rates of swell energy dissipation and under-prediction of wave period (van der Westhuysen et al., 2007). To address these shortcomings of the Komen-type models, Westhuysen et al. (2007) introduced an alternative whitecapping method based on a modified approach by Alves et al. (2003) using the concept of local saturation of spectra instead of mean spectral parameters. This method, known as the saturation-based (SB) approach, was successfully applied to several cases of sea-swell combinations and outperformed the Komen-type approaches (van der Westhuysen et al., 2007, hereafter W007; Mulligan et al., 2008). W007 evaluated the above two types of whitecapping formula by comparing SWAN modeling results and field observations for coastal and inland waters including the shelf waters of North Carolina and the shallow Lake IJssel, the Netherlands. Mulligan et al (2008) employed SWAN to study the evolution of waves in the semi-enclosed Lunenburg Bay in Nova Scotia under the effect of local waves generated by

an extratropical storm and swells from the northwest Atlantic. However, these applications were mostly implemented for settings with limited fetch lengths, such as lakes, bays, and small coastal areas (with limited fetch-lengths in the order of 100 km or smaller), and shorter study periods, so that time variations of the wind field were not considered.

During the recent years, several combinations of newer wind input/whitecapping formulations were developed and incorporated in the WAVEWATCHIII (WWWIII) model. These models including ST2, ST4, and ST6 were developed to be more consistent with physics of wind-wave/swell dissipation rather than only the mathematical balance that was the base for the previous formulations like Komen-type and Westhuysen (Babanin and van der Westhuysen, 2008; Tolman et al, 2014). The most recent formulation ST6 is basically an observation-based scheme that also includes the effect of negative wind input and wave-turbulence interaction. Although the ST6 physics package has been included in SWAN 41.20, yet there is limited experience with this physics package in coastal waters. Furthermore, the range of applicability of W007 for larger fetches with inhomogeneous wind fields is poorly known regardless of the fact that this formulation as well as the Komen-type approaches are still extensively being used for wave modeling of different regions all over the world. Therefore, it is imperative to examine these application ranges for different regions. Otherwise, extensive efforts may be required for model sensitivity analysis and calibration (Chaichitehrani, 2018; Allahdadi et al., 2019).

Other than W007 and Mulligan et al. (2008), studies that address the direct comparison between the Komen-type and SB whitecapping formulations based on real regional simulations are rare. A recent study by Vledder et al. (2016) for the North Sea during the severe storm of December 2013 showed that the Komen-type method with default parameters as incorporated in SWAN performed slightly better in the simulation of wave height, period, and frequency spectra than the SB method. This conclusion contradicts the results of previous studies. This suggests that there is no generally applicable source term setting exists requiring tuning. In the coastal regions of the North Sea also shallow water effects like bottom friction are a significant component of the total source term balance. Although Vledder et al. (2016) examined spatial and temporal variations of source terms regarding different whitecapping approaches, no specific reason was given to explain the better performance of the Komen-type formulation. The vast area of the simulation, with fetch lengths over 1000 km from the model boundary to the coastal areas where model results were evaluated, and the time variations of the wind field, are significant differences of this study from those of W007 and Mulligan et al. (2008). Hence, more studies are needed to determine the appropriate ranges of application for each whitecapping method. Moreover, wave growth and dissipation can be significantly affected by variabilities in the wind speed and direction (gustiness), instabilities in the air-sea boundary layer because of air-sea temperature difference, and slanting fetches over the coastal areas (Ardhuin et al., 2007; Donelan et al., 1985). Numerical tests of Ardhuin et al. (2007) showed that introducing wind speed gustiness may increase the simulated wave energy by up to 50%. They also reported that 10° of variability in wind direction might produce a similar effect that 10% variability in the wind speed causes. For smaller values of non-dimensional fetch, it was shown that a specific formulation of Komen-type whitecapping ($S_{ds}^{JHHK}$) was generally able to reproduce the measured fetch-growth curves of Kahma and Calkoen (1992) and Walsh et al. (1989) after correction for wind direction variability as well as the instability induced by the air-sea boundary layer. The mentioned variabilities in wind speed and direction (gustiness) occur within short time

slots (seconds to minutes) that cannot be considered by atmospheric models with hourly to several hourly outputs (Abdallah and Cavaleri (2002)). To be consistent with the practical modeling efforts and real applications, we did not correct the wind field for gustiness. Therefore, in our simulation we used a wind field with 1 hour temporal resolution that cannot include such short term variations (see section 4 for details).

The present study was motivated by a recent 31-year simulation that authors implemented for the U.S East Coast to characterize wave energy resources (Allahdadi et al., 2019). We used SWAN for this modeling since due to complex coastal geometry we could benefit from the flexible mesh option made available by SWAN. The only whitecapping formulations that are available by SWAN-ADCIRC (we had to use this coupled version to implement the domain decomposing needed for the high-performance computation over more than 4,300,000 computational grid points) are Komen-type and Westhuysen approaches. During the study we had to perform extensive sensitivity and calibration to select the most appropriate whitecapping approach amongst the ones available in SWAN and a part of these efforts is reported here. The main research questions that will be answered through the present study are: what are the appropriate ranges of applications for Komen and Westhuysen whitecapping formulations (see section 2 for detailed formulations) for the geographical extent of the U.S East coast and under the outbreak of the winter storms? And what are the effects of different wind conditions and other factors like coastal geometry and instabilities in the air-sea boundary layer on evolution of waves under this circumstances? The relatively large fetches and instationary large wind fields that are common for the real cases of wave generation and propagation over extensive regions are examples of the cases that need further attention.

It should be noted that the main goal of the present study is examining the performance of these whitecapping formula within a real simulation framework, not a detailed examining of the theoretical basis. Through this study, the performance of these two whitecapping formulations is evaluated against in situ observations using simulations for the U.S East Coast coastal ocean. Wind and wave fields over this area follow a seasonal pattern with most energetic wind wave events in late fall and early winter (Allahdadi et al., 2019). During the summer, the effect of swells with longer periods is more pronounced over the study area, an additional component that may cause differences in the performance of Komen-type and SB whitecapping approaches. Hence, a separate model performance evaluation for each season is warranted. The present paper is dedicated to implementing this evaluation during January 2009. The persistent offshore-ward wind field during this month (Allahdadi et al., 2019) along with large fetch lengths over the modeling region provide an appropriate condition to study different features of wind field and waves including fetch-limited and fully developed sea states.

## 2    Source term quantification for whitecapping and wind input

The default approach for quantifying whitecapping dissipation in SWAN is the pulse-based, quasi-linear model of Hasselmann (1974) that was formulated by Komen et al. (1984):

$$S_{ds,w}(\sigma,\theta) = -C_{ds}\left((1-\delta) + \delta\frac{k}{\bar{k}}\right)\left(\frac{\bar{s}}{\bar{s}_{PM}}\right)^p \tilde{\sigma}\frac{k}{\bar{k}}E(\sigma,\theta) \tag{2}$$

$$\tilde{S} = \tilde{k}\sqrt{E_{tot}} \tag{3}$$

where $C_{ds}$ is the whitecapping coefficient and $\delta$ is a parameter for partially adjusting wave period that varies between
0 and 1 with the default of 1 in SWAN. It may also affect the wave height and one may need to change it in agreement with $C_{ds}$). Rogers et al (2003) reported that by changing $\delta$ from 0 to 1 the accuracy for prediction of wave energy corresponding to low frequencies increases. Parameter $p$ is a constant, $k$ is wave number with the average of $\tilde{k}$, $\sigma$ is the angular wave frequency with the average of $\tilde{\sigma}$, $E_{tot}$ is the total energy of the wave spectrum, $\tilde{S}$ is the mean spectral steepness, and $\tilde{S}_{PM} = (3.02 \times 10^{-3})^{1/2}$ is the mean spectral steepness due to the Pierson-Moskowitz
spectrum (SWAN, 2015). Average angular frequency and average wave number are calculated by integration over the

frequency-directional spectrum as $E_{tot}/\iint \sigma^{-1}E(\sigma,\theta)d\sigma d\theta$ and $\left[E_{tot}/\iint k^{-\frac{1}{2}}E(\sigma,\theta)d\sigma d\theta\right]^2$ respectively. The strong dependency of the formulation to the mean spectral parameters is maintained through $\tilde{k}, \tilde{\sigma}$, and $\tilde{S}$. Two sets of values for $C_{ds}$ and $p$ were found by Komen at al. (1984) and Janssen (1992) by balancing the energy equation through fetch growth tests. The Komen-type whitecapping method is conjugated with a wind input term that includes both
linear and exponential growth terms (SWAN, 2015):

$$S_{in}(\sigma,\theta) = A + BE(\sigma,\theta) \tag{4}$$

In SWAN, the linear term (A) is estimated by a formulation of Cavaleri and Malanotte-Rizzoli (1981). The exponential growth term(the term including the coefficient B) is a function of the spectral energy and accounts for the main energy input by the wind. The coefficient B for Komen et al. (1984) is calculated based on the wave age inverse $\frac{u_*}{c}$ and
angular frequency:

$$B = \max\left[0, 0.25\frac{\rho_a}{\rho_w}\left(28\frac{u_*}{c}\cos(\theta - \theta_w) - 1\right)\right]\sigma \tag{5}$$

Where $u_*$ is the wind shear velocity, $c$ is wave phase speed, $\rho_a$ and $\rho_w$ are air and water densities respectively, $\theta$ is the direction of spectral component for which wind input is calculated, and $\theta_w$ is the wind direction.

       The SB method for resolving whitecapping dissipation was developed in response to shortcomings in Komen-
type methods due to the dependency of this process on the mean spectral parameters. The attempts mostly focused on removing the dependency on the mean spectral steepness and wave number. Alves and Banner (2003) presented a new form that related the wave groups to the whitecapping dissipation. This form was adopted and modified by W007 and was incorporated into SWAN as the SB model for whitecapping. This approach assumes that whitecapping dissipation affects wave groups when reaching a specific threshold:

$$S_{ds,break}(\sigma,\theta) = -C_{ds}\left[\frac{B(k)}{B_r}\right]^{p/2}[\tanh(kd)]^{\frac{2-p_0}{4}}\sqrt{gk}\,E(\sigma,\theta) \tag{6}$$

$$B(k) = \int_0^{2\pi} c_g k^3 E(\sigma,\theta)d\theta \tag{7}$$

$$p = \frac{p_0}{2} + \frac{p_0}{2}\tanh\left[10\left(\sqrt{\frac{B(k)}{B_r}} - 1\right)\right] \tag{8}$$

In the above equations, $p_0$ is a function of the wave age inverse $\frac{u_*}{c}$, $d$ is water depth, $g$ is acceleration due to gravity,
and $c_g$ is the wave group velocity. $B(k)$ is defined as the azimuthal-integrated spectral saturation and $B_r = 1.75 \times$

$10^{-3}$ is the threshold saturation level. If $B(k) > B_r$, waves break due to whitecapping. The dependency of the dissipation equation on group velocity and the associated saturation level for each wave group leads to separate estimation of the whitecapping dissipation for seas and swells and reduces their unrealistic interaction that significantly affects the whitecapping dissipation over different parts of the spectrum. In the version of SB whitecapping that is incorporated in SWAN, an additional term has been used for inclusion of dissipation caused by turbulence and interaction between long and short waves. Thus, the final whitecapping dissipation term is a weighted sum of the dissipation due to breaking and non-breaking waves:

$$S_{ds,w}(\sigma, \theta) = f_{br}(\sigma)S_{ds,break} + \lceil 1 - f_{br}(\sigma) \rceil S_{ds,non-break} \tag{9}$$

In the above equation, $f_{br}(\sigma)$ is a function of $B(k)$ and $B_r$ that provides a smooth transition between the breaking and non-breaking components of the dissipation.

Finally, a consistent wind input expression that considers the exponential wave growth by wind, suggested by Yan (1987), is used in conjunction with the SB whitecapping approach. This expression is obtained based on the laboratory and field measurements that show quadratic growth for $\frac{u_*}{c} > 0.1$ (e.g., Plant, 1982; Pierson and Belcher, 2005) and linear growth for $\frac{u_*}{c} < 0.1$ (Snyder et al., 1981; Hasselmann and Bösenberg, 1991):

$$B = D\left(\frac{u_*}{c_{ph}}\right)^2 \cos(\theta - \theta_w) + E\left(\frac{u_*}{c_{ph}}\right)\cos(\theta - \theta_w) + F\cos(\theta - \theta_w) + H \tag{10}$$

where $D, E, F,$ and $H$ are coefficients of the fit (W007).

### 3    Modeling domain and field data

The modeling area includes the U.S East Coast from the Gulf of Maine to south Florida and the offshore areas west of 61.0˚ W (Figure1). This area is characterized by seasonal variations of winds (Allahdadi et al., 2019). For example during January 2009, the average wind direction along and off the U.S East Coast was northwesterly to westerly with average wind speed of 6.5 m/s, while for July 2009 average wind direction was southwesterly with average wind speed of 3.5 m/s. During the late fall and the entire winter, the study area, especially the northern part, is significantly affected by extratropical storms with strong westerly winds that generate coastal and offshore waves with the general direction of west to east (Allahdadi et al., 2019). In the present study, the evolution of waves is investigated during January 2009.

The performance of the two whitecapping methods was evaluated at four NDBC buoys, including two in the north part of the modeling area (44017 nearshore and 44011 offshore) and two in the south (41004 nearshore and 41048 offshore) (Figure 1 and Table1). Wave height, peak wave period, mean wave period, wind speed, frequency spectra, and other met-ocean parameters were collected at these buoys.

Time variations and wind roses of observed wind at stations in the northern and southern parts of the modeling area during January 2009 show a dominant eastward wind direction, which was the result of winter storm spreading over

the area (Figure 2). With the dominant direction of the wind from west to east, it is less likely that low-frequency swells from the Atlantic Ocean could propagate toward the U.S. East Coast. Therefore, this is an ideal period to investigate the performance of each whitecapping approach based on the traditional fetch-limited framework. This assumption was examined by inspecting measured frequency spectra at different stations over the modeling area in January 2009. It was confirmed that at most times during this month measured frequency spectra were single-peaked that are generated by local winds (See Figure3 for examples of the measured frequency spectra at NDBC 44011). Model performance can be evaluated for both short and long fetch lengths. In both the northern and southern regions of the model domain, wind speeds of 20 m/s or larger were observed. The average 10-meter wind speeds during this month were 9.7 m/s for station 44011 and 8.9 m/s for 41048.

## 4 Model setup

A high-resolution unstructured SWAN model (coupled with ADCIRC for implementing the domain decomposition in the parallel mode) with coastal resolution of 200 m was developed and applied in this study. Details of the model set up are given in Allahdadi et al. (2019). Model bathymetry was prepared using two data sources, including a high-resolution database from NOAA's Coastal Relief Model with a spatial resolution of 3 arc-sec (~90 m) for the coastal areas and NOAA's ETOPO1 Global Relief Model with a spatial resolution of 1 min (~1700 m) for deep and offshore areas. The model was forced by wind fields from the NCEP Climate Forecast System Reanalysis (CFSR) with a spatial resolution of 0.312° (almost 32 km for the East Coast region) and temporal resolution of 1 hour. Evaluation of CFSR wind fields at different buoys showed the good accuracy of this data for the East Coast Region (Figure 4). For the four stations for which the CFSR wind field were evaluated in Figure 4, the correlation coefficient varies between 0.88 and 0.92 and the bias is as low as -0.06 m/s that shows the general underestimation trend of the simulated wind speed by the CFSR. This underestimation is especially pronounced for higher wind speeds at two buoys located in the southern half of the modeling area (buoys 41004 and 41048). These results are consistent with evaluations of Yang et al (2017) and Allahdadi et al (2019). In the present study, no correction was applied on the wind field and the original CFSR data were used for forcing the wave model. Three-hourly snapshots of CFSR wind fields around two reference times (t1 and t2) are shown in Figure 5. Times t1 (1/8/2009 12:00) and t2 (1/21/2009 06:00) correspond to storms in the northern and middle parts of the model, respectively. At time t1, a severe extratropical storm (wind speed > 17 m/s) spread over the modeling area north of 36˚ N, while at time t2 the storm affects coastal and offshore areas from New York to Florida. These two times have been selected to maintain the fully-developed or fetch-limited criteria of the sea state (Coastal Engineering Manual, 2006) at offshore buoys 44011 and 41048. These conditions require small variations of wind speed and direction within several hours timeframe depending the fetch length and wind speed. At each buoy, fetch lengrths were estimated as the distance between the shoreline and the location of buoy in the direction of the sustained offshoreward wind. They will be used for further examination of model results in next sections. Along the open boundaries, the model was forced using the wave parameters obtained from a global WAVEWATCHIII model with a spatial resolution of 0.5° and temporal resolution of 3 hours. Following Whalen and Ochi (1978), Ochi (1998), and Allahdadi et al. (2004b), a JONSWAP frequency-spectra with the average enhance parameter of $\gamma$=3.3 was chosen for converting parametric wave data to 2-D spectra along the boundary. Due to the dominant west-to-east

wind over the modeling area during the simulation period, it is less likely for boundary waves to propagate toward the modeling area. Nevertheless, realistic boundary data were used in this study. The number of spectral directions and frequencies for discretization of 2-D spectra were 24 and 28, respectively. Simulation was done using a minimum frequency of 0.04 Hz, maximum frequency of 1.00 Hz, a computational time step of 10 minutes, and three

computational iterations per time step (Allahdadi et al., 2019). Source terms for whitecapping dissipation and their associated wind input formulation were examined based on the two types of whitecapping dissipation approaches discussed in section 2. For the rest of source terms including quadruplets, triads, depth-induced wave breaking, and bottom dissipation, the default methods in SWAN were used.

**5   Results**

Based on the model setup described in section 4, twin simulations were performed for January 2009 using Komen (1984) and van der Westhuysen (2007) to supply the formulation for quantifying whitecapping dissipation (Komen and Westhuysen hereafter). For both approaches, only the default SWAN parameters were used (For Komen $C_{ds} = 2.3 \times 10^{-5}$ and $\delta = 1$, while for Westhuysen $C_{ds} = 5 \times 10^{-5}$). Simulated significant wave height and mean period

for both approaches were compared with measurements at NDBC stations 41004, 41048, 44017, and 44011 (Figure 6). Spectral definitions of the simulated mean wave period that is used for model evaluation as well as the mean wave direction that is later used for representing the wave vectors are presented below:

$$T_{m02} = 2\pi \left( \frac{\iint \omega^2 E(\omega,\theta) d\omega \, d\theta}{\iint E(\omega,\theta) d\omega \, d\theta} \right)^{-1/2} \tag{11}$$

$$Dir = arctan \left[ \frac{\int \sin \theta \, E(\omega,\theta) d\omega \, d\theta}{\int \cos \theta \, E(\omega,\theta) d\omega \, d\theta} \right] \tag{12}$$

In the above equations $T_{m02}$ is the mean wave period, $\omega$ is the radian frequency of a specific wave energy component, $\theta$ is the direction of wave energy component, $E(\omega, \theta)$ is the corresponding wave energy for this spectral component, and $Dir$ is mean wave direction. Comparisons with field data show that both whitecapping approaches underestimate wave height and wave period (less pronounced for Komen) at all stations. For all stations, Westhuysen simulated smaller wave heights compared to both observations and Komen. (Figures 6a to 6d). While at all four stations

Westhuysen significantly underestimated the wave period (Figures 6e to 6h), wave periods from Komen differed from observations at some stations. Comparison results for wave height and period as obtained from measurements and simulation scenarios for t1 and t2 show similar patterns (Table2). It should be noted that SWAN uses a prognostic high frequency tail for integration over a full frequency range that can increase the integration range to 10 Hz (SWAN, 2015). Since buoys integrate parameters over a narrow spectral range (in the case of NDBC buoys of the present study

the range is 0.02-0.485 Hz) some additional discrepancies may be introduced to the comparisons between model and buoy parameters. Akpinar et al (2012) showed that these discrepancies are negligible for wave height and could only be important for lower values of wave period $T_{m02}$ (approximately lower than three seconds). Since the measured wave periods during our simulations at all four buoys are larger than three seconds for most times (Figure 6e-6h), we can safely neglect this discrepancy for the wave period comparison.

Simulation results for two whitecapping methods compared to buoy measurements are further investigated through scatter plots (Figures 7 and 8). Comparisons are quantified using standard metrics for model performance including correlation coefficient (R), bias, root mean square error (RMSE), and scatter index (SI) (Tehrani et al., 2013). Statistics for wave height show that while the correlation coefficient of the match-up comparison is slightly larger for the Westhuysen, at all four buoys, the average errors of the simulated wave heights (bias) and the average distance from the ideal agreement line (RMSE) are significantly smaller for Komen (Figure 7). The only exception is the RMSE for buoy 44017, for which the corresponding value of RMSE from Komen is just slightly larger than that of Westhuysen (0.52 for Komen and 0.49 for Westhuysen, see Figure 7c). Scatter indices, which show the scattering of simulated values around the ideal match-up line, are smaller at buoys 41004, 41048, and 44011 for simulated wave heights by Komen. Again, the exception is buoy 44017. This different behavior is due to the complex coastal geography upwind of the station that causes the slanting fetch effect when the prevailing wind is from the land toward offshore (Ardhuin et al., 2007). This effect will be further examined in Section 6. For simulated mean wave periods, the correlation coefficients between the two scenarios are very similar at all buoys (Figure 8). However, the remaining performance statistics significantly favor the Komen whitecapping predictions. For all buoys, Westhuysen substantially underestimates the mean wave period with the RMSE values between 1.1 and 1.6 seconds, while for the Komen method, RMSEs range between 0.85 to 1.05 seconds.

Simulated wave heights and periods using Komen and Westhuysen whitecapping approaches at times t1 and t2 are also investigated by examining snapshots of results over the modeling area (Figure 9 for the results at time t1). It is worth noting that times t1 and t2 were not selected arbitrary. They were selected so that at offshore buoys 44011 and 41048 almost spatially uniform wind fields with sufficient durations occurred between the land and the location of buoys so that the fetch-limited sea states are achieved at these buoys (Coastal Engineering Manual, 2006). This specific sea states will later be used for further discussions on the behavior of the whitecapping formula. At time t1, significant differences are observed between wave heights from the twin simulations, especially within the extensive region in the north that was affected by the intense storm winds (Figures 9a and 9b). Similarly, at time t2 (Figures 9c and 9d), significant differences result for the extensive areas offshore of North Carolina to New Jersey that is close to the instantaneous center of the storm. At both t1 and t2, substantial differences are observed between simulated wave periods (Figure 9e to 9h). At t1 (Figures 9e and 9f), wave periods off the New York coast are significantly underestimated by Westhuysen compared to Komen (period of 7 sec for Westhuysen and 9 sec for Komen), a pattern that is also observed for time t2 (Figures 9g and 9f) for all offshore areas off the Florida to Massachusetts coast.

To examine the performance of each whitecapping approach in the simulation of wave energy distribution, frequency spectra from two experiments were compared with measured spectra at each buoy and for t1 (Figure 10) and t2. Hourly frequency spectra at the buoys are available from observations for the frequency band of 0.02-0.485 Hz. However, spectral energy corresponding to frequencies smaller than 0.06 Hz were zero. To minimize the effect of measurement noises at t1 and t2, measured spectra were averaged within a three-hour time window (W007). At each location, frequency spectra were also presented in semilogarithmic scale on the energy axis to more clearly show the differences.

At t1 at buoy 41011, both methods appropriately simulated the general shape of the single-peaked spectrum and the value of the peak frequency (a slight overestimation by Westhuysen for peak frequency). While Komen simulated an almost identical peak energy, Westhuysen underestimated it by 18%. The peak of energy is also maintained by Komen for the other offshore station (NDBC 41048), but with significant underestimation of the peak frequency in the simulated spectra (about 30%). For both coastal stations (44017 and 41004), the peak of energy is significantly under-predicted by both experiments (34-75% depending on the station and simulation experiment) while the peak frequencies were off by -15 to 23%. At all buoys, Komen simulated larger peaks of energy, which are closer to the measurements. The consistency of the simulated frequency spectra at 44011 with that of measurements at time t1 is due to the persistent winds with almost constant speed and direction from the coast toward the station at this time and several hours before it (at least 6 hours; Figure 5). This wind condition can produce the fetch-limited wave growth with the well-developed single-peaked spectrum (Hasselmann et al., 1973) that can likely be simulated by different whitecapping formulations because they are evaluated and calibrated mainly based on the measured fetch-limited growth curves (W007, Ardhuin, et al. 2007). Discrepancies at the two coastal stations are caused by the effect of land roughness on the CFSR wind over the coastal areas (Allahdadi et al., 2019), non-persistence of wind field over these areas, and effect of slanting fetch (Ardhuin et al., 2007). The fetch-limited wave growth at 44011 is of particular interest due to available field observations and modeling studies (for instance, Kahma and Calkoen, 1992). As mentioned above, at this station at t1 the Komen approach shows almost identical values for the peak of energy and peak frequency to the measurements, while Westhuysen underestimates the peak of energy and overestimates the peak frequency. These results are in contradiction to the simulation result of W007 for a wave evolution test off the coast of North Carolina, USA. Their result showed that in the absence of offshore swells, the SB approach (Westhuysen) simulated higher levels of spectral energy corresponding to the peak frequency than that of Komen. Also, the simulated peak frequency from the SB model was more consistent with measurements. These different behaviors could be due to different growth conditions and wave age stages discussed in the next section.

Similar patterns as time t1 for comparison of simulated frequency spectra based on twin simulations and measurements are observed at time t2 (not shown). Because at this time the most persistent winds occur in the middle part of the modeling area, NDBC41048 shows the best consistency for spectral energy and peak frequency.

## 6 Discussion

### 6.1 Examining source terms

A part of discrepancies in the simulation results from both whitecapping formulations is caused by inaccuracies in the wind field (see Figure4) mainly due to general underestimation of the wind speed by the CFSR wind that results in underestimations in the simulated wave heights and wave periods. A calibrated model for the same study region as this paper implemented by Allahdadi et al (2019) resulted in the average bias of 0.11 m for significant wave height at different buoys. For the present study the average bias values for wave height are 0.19 m and 0.33 m for simulations with Komen and Westhuysen whitecapping respectively. If we conservatively assume that the whole bias in the calibrated model is attributed to the wind, more than half of the bias in simulation with Komen and one third of the

bias for simulation with Westhuysen whitecapping in the present study would be related to the wind. However, still significant differences are observed between simulation results from Komen and Westhuysen considering the fact that both used the same wind field. Simulation results presented in the previous section clearly show that compared to the in situ observations, the Komen whitecapping approach results in higher accuracy for both wave height and period. Over the modeling area, especially close to the instantaneous center of the storms at times t1 and t2, simulated wave heights and periods from Komen are larger than those of Westhuysen (Figure 9).

Spatial and temporal variations of source terms (integrated source term magnitudes) for wind input ($S_{wind}$), whitecapping dissipation ($S_{wc}$), and quadruplet ($Snl_4$) were obtained from SWAN simulations and diagnosed at these two times for both simulations to illustrate the contribution of source terms in the simulation results (Figure 11). For each modeling simulation, the three essential source terms are of the same order of magnitude and show similar values. This is consistent with Bouws and Komen (1983) and van Vledder et al. (2016). The quantified source terms by Westhuysen are significantly larger than those of Komen. For example, off the coast from New York Harbor to the Gulf of Maine, the estimated $S_{wind}$ by Komen varies between $1.5 - 2 \times 10^{-4} \; m^2/s$ whereas the simulated wind input source term by Westhuysen approach is at least twice as large as Komen's. This is because the wind input term is a direct function of $\frac{u_*}{c}$ in both formulations, but the wind input formulation for Komen (equation 5) is a linear function of this parameter and is mostly appropriate for weaker wind speeds up to 12 m/s (W007). Conversely, the wind input associated with Westhuysen whitecapping (Yan, 1987; equation 10) is appropriate for both weak and strong wind forcing and includes generation of wind energy as a function of both $\frac{u_*}{c}$ and $(\frac{u_*}{c})^2$. The wind input formulation for each whitecapping approach has been selected to be consistent with the scaling of the whitecapping to appropriately simulate the observed shape of the evaluated frequency spectra (W007) and keeping the total balance appropriate. Particularly for the spectral tail with frequencies 1.5 times higher than the peak frequency, Resio and Perrie (1991) reported that the dominant shape of the spectrum is a form which is a function of $f^{-4}$ ($f$ is wave frequency) for both weakly and strongly forced waves. This shape results from the stabilizing effect of the quadruplet interactions. Hence, spatial variations of whitecapping dissipation for each approach are of the same order of magnitude as their wind input counterpart. Similar to the wind input, the simulated whitecapping using Westhuysen shows higher values than those simulated by Komen. Compared to wind input and whitecapping, estimated quadruplet source terms as a result of using Komen and Westhuysen are closer in value.

Estimated source terms from the two simulations were also compared by examining variations in the frequency space at buoy 44011 at t1 (Figure 12). In addition to the main source terms of wind input, whitecapping dissipation, and quadruplet, their algebraic sums (sum of the first three right-hand terms in equation 1) are also compared in the frequency domain. The oscillatory variations of the quadruplet term with frequency, especially the ones for Westhuysen simulation could be due to oscillations of the whitecapping term between frequencies of 0.1 to 0.35 Hz (see Figure 12b). This pattern has also been simulated by Mulligan et al (2008). Like the integrated values of these source terms over the modeling area (Figure 11), variations of source terms versus frequency show larger values of wind input and stronger whitecapping dissipation by the Westhuysen approach (Figures 12a and 12b). The algebraic sum of the source terms is the ultimate energy amount that is produced at each time step due to source term interactions

and is subjected to spatial and temporal variations based on the equation of wave action conservation. Hence, variations of this term in the frequency domain can be consistent with the shape of the energy-frequency spectra of Figure 10. Komen simulated a larger sum of source terms at the peak frequency and all frequencies below that (Figure 12d). This result is consistent with Figures 10a that shows higher spectral energies at this time by Komen compared to Westhuysen. The consistent spectral energies from Komen and Westhuysen for the high-frequency spectral tail in Figure 10 can also be explained using Figure 12d. Compared to the peak frequency, simulated sums of source terms for frequencies larger than the peak frequency are half or smaller and their difference is not large enough to cause different spectral shapes in the tail of spectra. The above statements show that although the wind input counterpart of the Westhuysen approach (Yan, 1987) resulted in larger energy input to the sea surface than that of Komen, larger whitecapping by the Westhuysen formulation balances the excess input. For the case of our study, this dissipation from Westhuysen approach causes underestimation of the spectral energy in the wave spectrum.

### 6.2 Effect of wind field and growth conditions

In this section, the deficiencies associated with the Komen and Westhuysen whitecapping methods are investigated based on wave growth conditions during the simulation period. The performance of these two approaches for quantifying whitecapping dissipation and their wind input counterparts highly depends on the spatial and temporal variations of the wind field and the spatial scale of the modeling area, which both affect wave growth. Hence, developed approaches for wind input and whitecapping are primarily calibrated and verified using observed growth curves. These growth curves are represented in the form of non-dimensional energy and non-dimensional frequency both versus non-dimensional fetch $X^* = gX/u_*^2$ where $X$ is the fetch length. W007 verified both the SB (Westhuysen) and Komen (using the default SWAN parameters like the present study) whitecapping approaches versus the growth curves of Kahma and Calkoen (1992) (for fetch-limited growth) and Pierson-Moskowitz (1964) (for the fully-developed sea state) and determined the default calibration parameters for the SB model. The comparisons showed that using the default parameters for whitecapping, both approaches performed well during the fetch-limited growth when the value of the non-dimensional fetches are $<10^7$, although for $X^*$ values between $10^4$ and $10^5$, the Westhuysen approach was more consistent with observations. This study also indicated that for the fully-developed part of the growth curve ($X^* > 10^7$), the Komen approach with default parameters simulated higher amounts for the non-dimensional energy than Westhuysen. Although simulated non-dimensional energy by Komen was more consistent with observations, both approaches underestimated it. Furthermore, both whitecapping approaches overestimated the non-dimensional peak frequency for $X^* > 10^7$ that leads to lower wave periods in simulation. These cases of inconsistency could partly contribute to the underestimation of wave height and period obtained from Komen and Westhuysen whitecapping approaches with default parameters. Among the four NDBC buoys used for model result verification in this study, two (41011 and 41048) are offshore 600 and 1300 km from the shoreline in the east-west direction. Hence, because of the dominant offshore-ward direction of the wind during the simulation period in January 2009 (Allahdadi et al., 2019), large values for the non-dimensional fetch are resulted at these locations. This could be corresponding to the fully-developed sea state that is the zone of inconsistency based on the above discussions. At t1,

a strong wind with a westerly direction affected the East Coast and offshore areas north of 33°N. The consistent wind direction with the average speed of about 15.5 m/s from the coast to buoy 41011 produced a fully-developed sea state with $X^* \approx 2 \times 10^7 > 10^7$. Hence, underestimation in both wave height and wave period is expected. However, in this area of the growth curve, Komen generates higher levels of energy, i.e. higher wave heights result (Figure 6). For the other offshore station (41048), even larger values for $X^*$ on the order of $10^8$-$10^9$ are obtained that correspond to

larger underestimations that are also evidenced in Figure 6. At t2 and 6-10 hours before that, the wind at buoy 41011 was consistently from the northeast with average speed of 7 m/s, corresponding to a strong fully-developed sea state with $X^* \approx 10^8$. At this time, the wave height was significantly underestimated by both whitecapping approaches, especially by Westhuysen (Table 2). At coastal stations, however, due to the generally short fetch lengths during the winter storm outbreak, fully-developed sea states were less likely. For instance, at both t1 and t2, the persistent wind

at buoy 41017 corresponded to $X^* = 5 \times 10^6$ and $4.3 \times 10^6$ respectively indicating fetch-limited growth.

       Wind input and whitecapping source terms for both Komen and Westhuysen are direct or indirect functions of the wave age inverse $\frac{u_*}{c}$ (equations 5, 8, and 10). Multiple studies reported that with increasing wave age (decreasing the wave age inverse), dissipation due to whitecapping decreases (W007; Longuet-Higgins and Smith, 1983; Katsaros and Ataktürk, 1992). Wave age inverse is also an appropriate manifestation of the sea state and an indicator of whether

the sea state is in the forcing phase or fully-developed. Volov (1970) and Oost (1998) suggested and Drennan and Graber (2003) later confirmed that a developing sea corresponds to $\frac{u_*}{c} > 0.05$, while $0.033 < \frac{u_*}{c} < 0.05$ indicates a fully-developed sea state. For offshore buoy 44011 and nearshore buoy 44017, variations of simulated hourly whitecapping dissipation with the inverse wave age for two experiments are plotted in Figure 13. Since the scaling of the whitecapping formula in Komen and Westhuysen differ (Figure 11), simulated whitecapping values on the vertical

axes are normalized based on the maximum value in each case. At both stations and for both whitecapping methods, whitecapping dissipation increases with increasing inverse wave age, although the nearshore station has more scattering due to the fetch length variations caused by the coastline irregularities. At the offshore station 41011(Figures 13a and 13c), significant numbers of events are included in the fully-developed zone. The density of the simulated incidents in this zone decreased for the coastal station due to smaller fetch lengths. Based on the criteria specified by

Volov (1970) and Oost (1998) and the calculated values of $\frac{u_*}{c}$ for simulation outputs, frequency of occurrence (FO) for two main sea states including "developing" and "fully-developed" were calculated (Figure 14) at offshore station 44011 and the coastal station 44017. Also, measured wave peak period and wind speed at the location of these two NDBC stations were used to calculate the FO's for two sea states. Comparisons showed that at both locations the FO for "developing" sea state was significantly overestimated by both models, although Komen simulated more consistent

values of FO with those of buoys. This "developing" sea state during which wind energy is actively transferred from wind to water can be corresponding to either fetch-limited or duration-limited wave growths.  For the "fully-developed" sea states Komen and Westhuysen compare different at the costal and offshore buoys. At the offshore buoy 44011, Komen simulated almost the same FO as the buoy observations, while Westhuysen underestimated the FO by 7%. Both models underestimated the FO for the "fully-developed" sea state at the coastal station, but again

Komen's performance is slightly better. These results show that for the two major sea states resulted from generation

and propagation of wind-waves, especially for the "developing" state, Komen and Westhuysen whitecapping formulas, may present substantially different features with those of measurements.

The above discussion shows that for the East Coast, a significant part of the deficiencies at offshore buoys (and to some extent at nearshore buoys) is caused by the spectral energy underestimation/peak frequency underestimation by these approaches during fully-developed sea states. This could be fixed by revisiting the models' calibration process and selecting smaller amounts for the default whitecapping parameter ($C_{dis}$) corresponding to the large values of the non-dimensional fetches. The default value for Komen whitecapping as presented by Komen et al. (1984) is $2.3 \times 10^{-5}$, while for the Westhuysen approach, W007 suggested $C_{ds} = 5 \times 10^{-5}$ based on comparisons with field measurements. However, the modified whitecapping parameter for the fully-developed condition may cause inconstancies in the fetch-limited zone of the growth-curve regarding the fact that the simulated non-dimensional energies and peak frequencies already match the measurements. Hence, it is suggested that in future modifications, if possible; fully-developed and fetch-limited conditions are treated independently, so that models could be able to calculate the whitecapping parameters based on the instantaneous non-dimensional fetches. Furthermore, within an extensive modeling area with a high spatially and temporally variable wind field, an ideal fetch-limited condition is less likely to occur, at least for offshore areas, for the East Coast during winter storms. The large fetch lengths for these areas need several hours of persistent winds with small variations in speed and direction to develop a fetch-limited condition. If variations of wind speed and direction occur often, the conditions for reaching a fetch-limited or fully-developed sea state are violated (Coastal Engineering Manual, 2006). However, spatial and temporal variations of the wind field over this area cannot generally stimulate such a condition. In fact, times t1 and t2 were two infrequent cases for which the persistent winds were dominant over a part of modeling area for several hours. It means that for many points on Figure 13, the values of $\frac{u_*}{c} > 0.05$ may represent duration-limited wave growth that is not a part of the calibration process during the development of the whitecapping approaches, especially for Westhuysen. Revisiting the calibration process and including the duration-limited growth curves (non-dimensional wind duration instead of non-dimensional fetch) lead to updated and more consistent calibration parameters. For coastal buoys, the coastal geometry may influence model accuracy as discussed in the next section.

### 6.3 Effect of coastal geometry

Similar to the offshore regions, deviations from the fetch-limited condition in coastal areas can contribute to the underestimation of wave height and period, although due to shorter fetch lengths it is more likely for the coastal areas to reach fetch-limited growth (Hasselman et al., 1973). However, significant underestimation for both wave height (0.32 < RMSE < 0.52 m) and period (0.85 < RMSE < 1.37 sec) were observed from simulation results at the two coastal stations (41004 and 44017; see Figures 7 and 8 for details). Due to relatively deep water at these locations (38 and 52 m respectively), shallow water phenomena are not likely to affect simulation results. However, the proximity to land may contribute to the underestimation in several ways. First, although the 32 km spatial resolution of CSFSR wind that was used for the present simulation is one of the finest available resolutions for the East Coast, interpolation of wind land points over the mesh in the coastal areas may significantly underestimate wind speed used in SWAN

(Dobson et al., 1989; Taylor and Lee, 1984). Second, regarding the performance of whitecapping and wind input approaches and their interaction with the quadruplet source term over the coastal areas, several studies highlighted the effect of the fetch geometry and the deviation of the wind direction from the shore-normal direction on wave evolution (e.g., Ardhuin et al., 2007; Donelan et al., 1985). Ardhuin et al. (2007) used the term "slanting fetch" for such a condition. Based on wave measurements at several coastal stations along the North Carolina and Virginia coast, they observed that even with small deviations in offshore-ward wind direction from the shore-normal, two distinct wind-sea systems are produced. The low-frequency systems propagate alongshore in the approximate direction of the slanting fetch, while the higher frequency wave system propagates downwind. For resolving the quadruplet term they used the Direct Interaction Approximation (DIA) method which is the same as the default method in SWAN and was used in the present simulation. Bottema and Vledder (2008) showed that using the exact quadruplet methods (Xnl) results in stronger changes in the coastal wave directions compared to the case that DIA is used. However, using Xnl needs significantly higher computational resources that is not practical for regular uses.

Buoy's frequency-directional spectra (reconstructed from the Fourier coefficients that can be distilled from the buoys time series) at 44017 and for times t1 and t2 (Figures 15a and 15b) illustrate this behavior. From a modeling perspective, whitecapping approaches and their wind input counterparts when interact with the quadruplet term may fail partly or entirely to simulate the part of the spectra with higher directional spreading from the mean wind direction (Ardhuin et al., 2007). Ardhuin et al. (2007) reported that the directional distribution associated with the wind input term of Jensen (1991) is too narrow. Therefore, it is not able to simulate enough energy for directional bands away from the mean wind direction. Consequently, less energy is transferred to the directions close to the slanting fetch compared to observations and this may contribute to a further underestimation of wave height and period at the location of coastal stations. The simulated frequency-directional spectra at buoy 44017 using Komen and Westhuysen approaches at t1 and t2 are compared with those from observations in Figure 15 c-f. At t1, the local wind direction is from west to east, and the measured spectrum (Figure 15a) shows a wide spectral band extended from 90 to 300 in the clockwise direction with the high energy zone formed at directions close to the wind direction. At the same time, a lower frequency spectral band from 330 to 85 with the main direction parallel to the coastline (Long Island is to the north of 44017) is produced as a separate wave system. The simulated wave spectra using both whitecapping approaches, however, capture only the higher frequency portion of the spectrum generated downwind and fail to simulate the lower frequency part produced by the slanting fetch effect. While their directional spreading (the total angle for which wave energy exists within the scale to 360 degrees) is almost the same (Komen's spectra is slightly wider), as expected, Komen results in higher energy levels. Although at t2 simulated spectra were able to reproduce the main portion of the low-frequency spectral zone caused by the slanting fetch effect, they both failed to include that portion of the low-frequency wave system that propagated from the northern quadrant (Figures 15d and 15f).

### 6.4   Effect of boundary layer instability

For both Komen and Westhuysen approaches, the associated wind input terms are quantified by assuming a stable air-sea boundary layer, i.e., air temperature is assumed to be the same as or higher than the sea surface temperature.

However, there are many occasions, especially during the winter, when the air is colder than the water. This negative temperature difference can cause instability at the air-sea boundary layer and lead to higher rates of wind energy transfer to the water surface. This instability effect has been studied by several researchers to modify the quantification of the wind input source term (e.g., Abdalla and Cavaleri, 2002; Tolman, 2002). Tolman (2002) suggested a relationship for correcting wind speed based on the air-sea temperature difference that increases the input wind speed to the model if this difference is negative. Ardhuin et al. (2007) applied this relationship to an unstable case with dT = -10 (dT=air temperature-water temperature) to correct the wind speed and were able to successfully reproduce the unstable growth curve of Kahma and Calkoen (1992). For this amount of the air-sea temperature difference, Tolman's relationship showed about a 24% increase in the input wind speed compared to the neutral case. In the present study, field measurements of air temperature and sea surface temperature at different locations in the model domain showed that during the simulation period in January 2009, there were many events with an unstable boundary layer. These instability incidents are more frequent and stronger for buoys in the north of the model domain (Figure 16a for 44011). To illustrate the potential effect of temperature instabilities on simulation results, observed and simulated wave heights based on both simulations are presented in Figure 16b. For most days during January 2009, the air-sea temperature difference was negative, indicating an unstable boundary layer. Temperature differences were as low as -10°C with the average of -5°C, corresponding to an increase in wind speed from 15-24% based on Tolman's relationship. In the present simulation, wind speed correction due to the boundary layer instability was not considered. As shown in Figure 16b, the underestimation of wave height by both whitecapping approaches in many cases coincided with the negative temperature difference occurred several hours to 1-2 days before the peak of wind/wave. The comparison between the simulated wave height deviation by both whitecapping approaches from the measured wave height (dHs) with air-sea temperature difference (dT) at NDBC 44011 shows a relatively strong correlation between the wave height underestimation and negative values of dT (Figure 17). It should be noted that in addition to the boundary layer instability, other factors as mentioned in the previous sections contribute to the wave height underestimation, hence the plots in figure 17 include the effect of several phenomena. However, the correlation between dHs and dT account for a general trend of increasing negative dHs with increasing negative dT. These temperature-related deficiencies in the simulated wave height are caused due to the fact that Komen and Westhuysen wind input/whitecapping formula fail to consider the effect of boundary layer instabilities. Hence, by incorporating appropriate modifications that include the effect of boundary layer instability on the wind field, higher wave heights that are more consistent with observations are expected.

## 7    Summary and Conclusion

Selecting appropriate modeling approaches for wind input and whitecapping source terms is essential for high accuracy wave modeling. Available methods have some limitations regarding the wind climate over the modeling area, spatial scales, coastal geometry, and presence of swells. The Komen-type whitecapping methods produce spurious results under a combination of seas and swells. The SB model of W007 (Westhuysen) was developed to modify this spurious effect. For an extensive modeling area like the U.S. East Coast and its offshore areas, the performance of each type of whitecapping method and its associated wind input terms should be evaluated during

varied meteorological conditions. Since the wind conditions of the East Coast are very different between winter and summer, seasonal investigations need to be done separately. During the winter, wind direction is mostly offshore-ward and along the coast, and Atlantic swells are less likely to propagate over the model domain, while during the summer, wind power significantly weakens and swells predominate.

The present paper evaluates model performance during an outbreak of winter storms in January 2009. Simulation
results showed that using either Komen or Westhuysen to resolve whitecapping led to an underestimation of wave height at coastal and offshore stations, although Komen resulted in larger wave heights that were more consistent with observations. While simulated mean wave periods using Komen were in a good agreement with observations, Westhuysen significantly underestimated wave periods at all four buoy locations used for model evaluation. Examining the quantified source terms over the modeling area indicated that for each whitecapping approach, the
source terms for wind input, whitecapping dissipation, and quadruplet have the same order of magnitude and follow similar spatial and temporal variations. For the wind input formulation of Yan (1987), which is associated with the Westhuysen whitecapping method, the wind input source term was modified for the intense wind speeds that include the energy generation as a function of both $(\frac{u_*}{c})^2$ and $\frac{u_*}{c}$. Hence, the resulting wind input at the peak of the storm was 2-3 times larger than that of Komen, which only scales the wind input as a linear function of $\frac{u_*}{c}$. For both methods,
quantification of the whitecapping dissipation terms (and thereby calculation of the quadruplet term) accords with the scaling of the wind input terms. The algebraic sum of source terms (that is transferred to the equation for the conservation of the wave action density) from Komen includes higher amounts of energy, especially for lower frequencies and at peak frequency. This leads to higher spectral energies from Komen whitecapping available to the frequency-directional spectra that contributes to larger wave heights and periods compared to Westhuysen. Several
reasons contribute to this underestimation over the coastal and offshore areas. Generally, the whitecapping formulas and their wind input counterparts are developed and tested to comply with the traditional fetch-limited and fully-developed growth curves. For the specific case of the saturation-based whitecapping and to some extent Komen-type whitecapping, the numerical tests of W007 showed that the calibrated models based on growth curve of Kahma and Calkoen(1992) underestimate spectral energy within the fully-developed part of the growth curve. This behavior
corresponds to the underestimation of wave height and period at the offshore buoys, where the large fetches during the offshore-ward wind events are more likely to produce fully-developed growth compared to coastal stations. For many events that do not correspond to the fully-developed sea state at the offshore and coastal stations, wave parameters are still underestimated. This could be partly because of the transient wind field that produces duration-limited growth, a condition that was not included in the calibration and verification of whitecapping approaches during
their development phase. Therefore, re-visiting the calibration process for both methods and representing new default parameters for whitecapping is highly recommended. The default parameters should be presented for different wave development conditions including fetch-limited, duration-limited, and fully-developed. The duration-limited condition should be especially considered since it has not been included in previous studies of developing and testing the whitecapping methods.

For the coastal stations, the deviation of the wind direction from the shore-normal direction (directionally dependent fetch-lengths) that is very likely due to the complicated coastal geometry (variations in the coastline direction) along with variations of wind direction over the coastal areas, causes the "slanting fetch" effect that transfers part of the wind-induced energy to the low frequencies and wave propagation along the shoreline. Generally, the source balance in SWAN resulted from the interaction of wind input, whitecapping, and quadruplet is not able to simulate large

spreading from the mean wind direction and this alongshore counterpart of the 2-D spectra may be overlooked. Comparison with observed 2-D spectra at coastal stations showed that source balance resulted from both whitecapping approaches partly or completely fails to include this low frequency part, further contributing to the underestimation of wave parameters.

    Instabilities in the air-sea boundary layer induced by colder air temperature than sea surface temperature may

significantly increase wind energy transfer to waves, i.e., create larger wave heights. Although during January 2009, this temperature difference at the offshore station 44011 reached -10°C, none of the wind input approaches are able to include this intensifying effect.

    In the present study, low frequency swells from the Atlantic were less likely to propagate toward the modeling area under the prevailing west to east wind direction. Hence, the evaluation was mostly limited to the pure wind-wave

generation during January 2009. More studies are required to address the spurious effect (unrealistic lower or higher whitecapping dissipation that is produced in the presence of swells) of low-frequency swells on whitecapping dissipation resulting from the Komen-type models over this study area. Therefore, similar simulations and analyses for the summer will be conducted. Results will be reported in future correspondence.

    The present study and the future planned studies for other seasons are required to provide more scientific support

when applying two different whitecapping formulations in the context of available schemes in SWAN. Including the newer more consistent physics packages of ST4 (ST6 has already been included) in SWAN will add more options for SWAN users to choose the best formulations based on their specific regions and wave climates.

## 8   Acknowledgments

This study was funded by the Wind and Water Power Technologies Office (WWPTO) within the Office of Energy Efficiency and Renewable Energy (EERE), U.S. Department of Energy. Sandia National Laboratories is a multi-mission laboratory managed and operated by National Technology and Engineering Solutions of Sandia LLC, a wholly owned subsidiary of Honeywell International Inc. for the U.S. Department of Energy's National Nuclear Security Administration under contract DE-NA0003525. This paper describes objective technical results and

analysis. Any subjective views or opinions that might be expressed in the paper do not necessarily represent the views of the U.S. Department of Energy or the United States Government.  RH also acknowledges research support provided by NSF grant OCE1559178 and NOAA grant NA11NOS0120033. Authors sincerely appreciate the constructive comments from the anonymous reviewers that significantly improved the quality of this paper. We thank Jennifer Warrillow for editorial assistance with the manuscript

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

**Table1: Information on NDBC stations used for evaluation of model results**


| Buoy | Depth(m) | Description | Longitude | Latitude |
|---|---|---|---|---|
| 41004 | 38.4 | EDISTO - 41 NM Southeast of Charleston, SC | -79.099 | 32.501 |
| 41048 | 5340 | WEST BERMUDA - 240 NM West of Bermuda | -69.590 | 31.86 |
| 44011 | 82.9 | GEORGES BANK 170 NM East of Hyannis, MA | -66.619 | 41.098 |
| 44017 | 52.4 | MONTAUK POINT - 23 NM SSW of Montauk Point, NY | -72.048 | 40.694 |

**Table 2: Simulated wave heights and mean periods using Komen and Westhuysen whitecapping methods at reference times**
**t1 and t2 compared to observations at the four NDBC buoys.**

| t1(01/08/2009 12:00) | | | | | | |
|---|---|---|---|---|---|---|
| | Wave height(m) | | | Wave Period(sec) | | |
| Buoy | Measurement | Komen | Westhuysen | Measurement | Komen | Westhuysen |
| 41004 | 2.02 | 1.76 | 1.48 | 4.74 | 3.99 | 3.61 |
| 444017 | 5.05 | 3.53 | 3.17 | 7.19 | 5.78 | 4.89 |
| 41048 | 3.44 | 3.39 | 2.76 | 6.93 | 7.65 | 5.74 |
| 44011 | 6.20 | 6.04 | 5.57 | 7.71 | 8.25 | 6.83 |
| t2 (01/21/2009 06:00) | | | | | | |
| | Wave height(m) | | | Wave Period(sec) | | |
| Buoy | Measurement | Komen | Westhuysen | Measurement | Komen | Westhuysen |
| 41004 | 1.61 | 1.55 | 1.02 | 4.91 | 3.74 | 3.23 |
| 444017 | 1.87 | 1.56 | 1.49 | 4.52 | 3.99 | 3.84 |
| 41048 | 6.07 | 5.59 | 5.02 | 8.62 | 8.11 | 6.70 |
| 44011 | 2.26 | 2.21 | 1.94 | 5.52 | 5.01 | 4.58 |


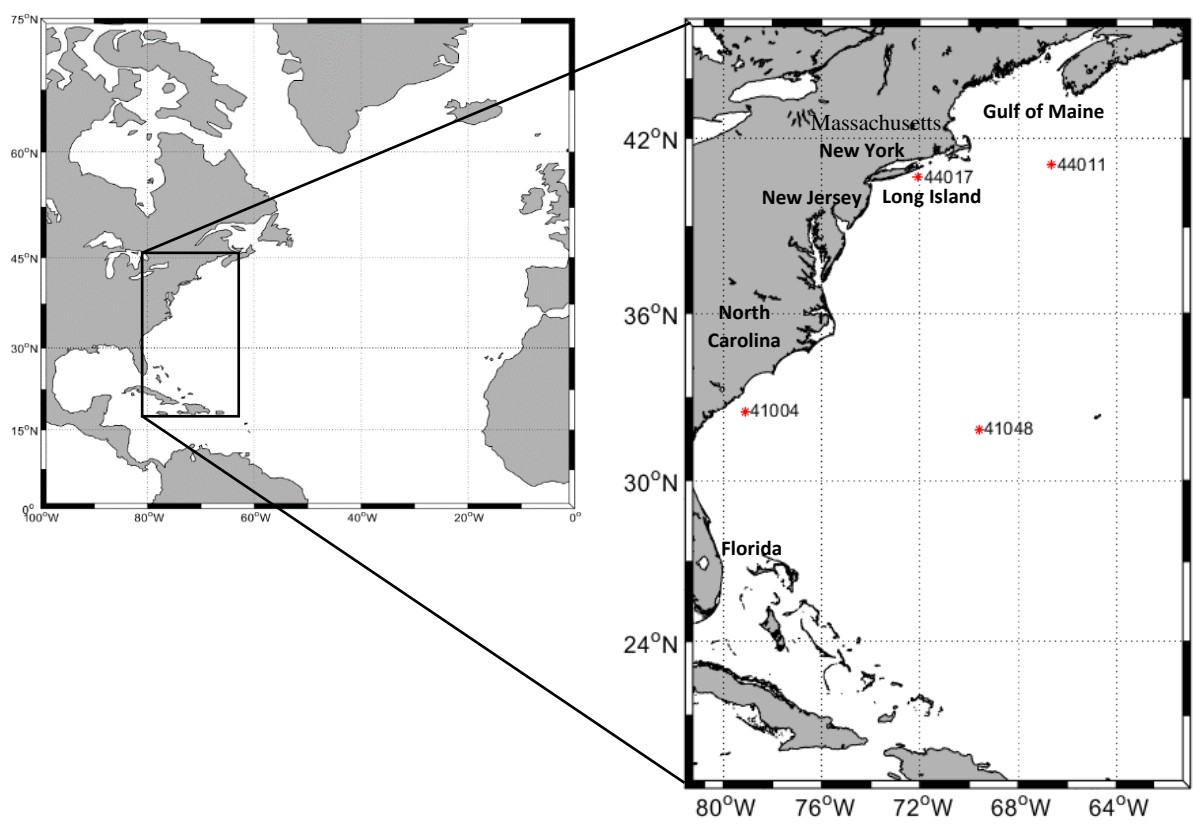

**Figure1: Modeling area and locations of NDBC buoys used for evaluation of whitecapping models**



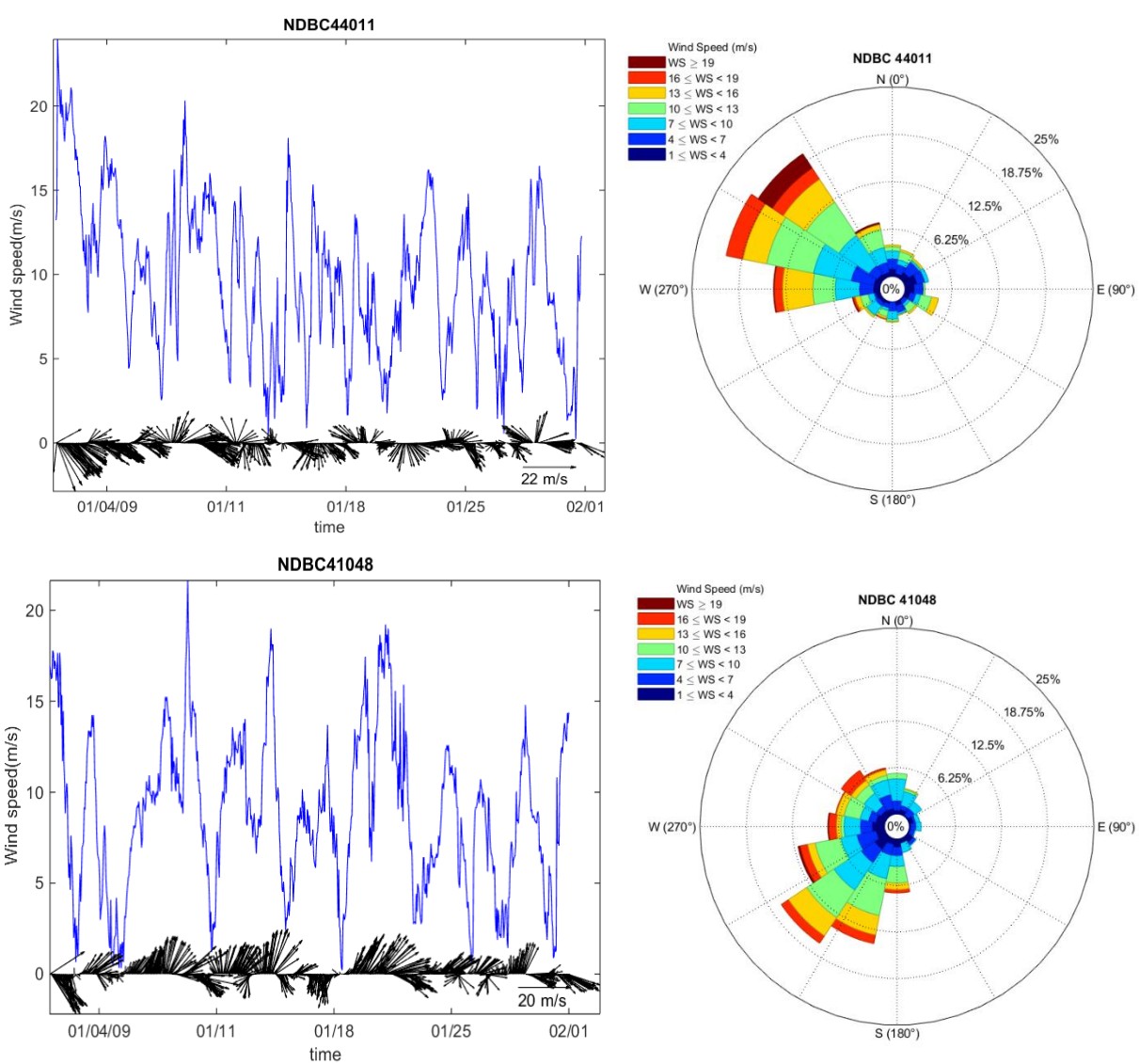

**Figure2: Time series of measured wind speed (lines) and vectors (arrows) and the associated wind roses at NDBC buoys 44011 (upper) and 41048 (lower).**




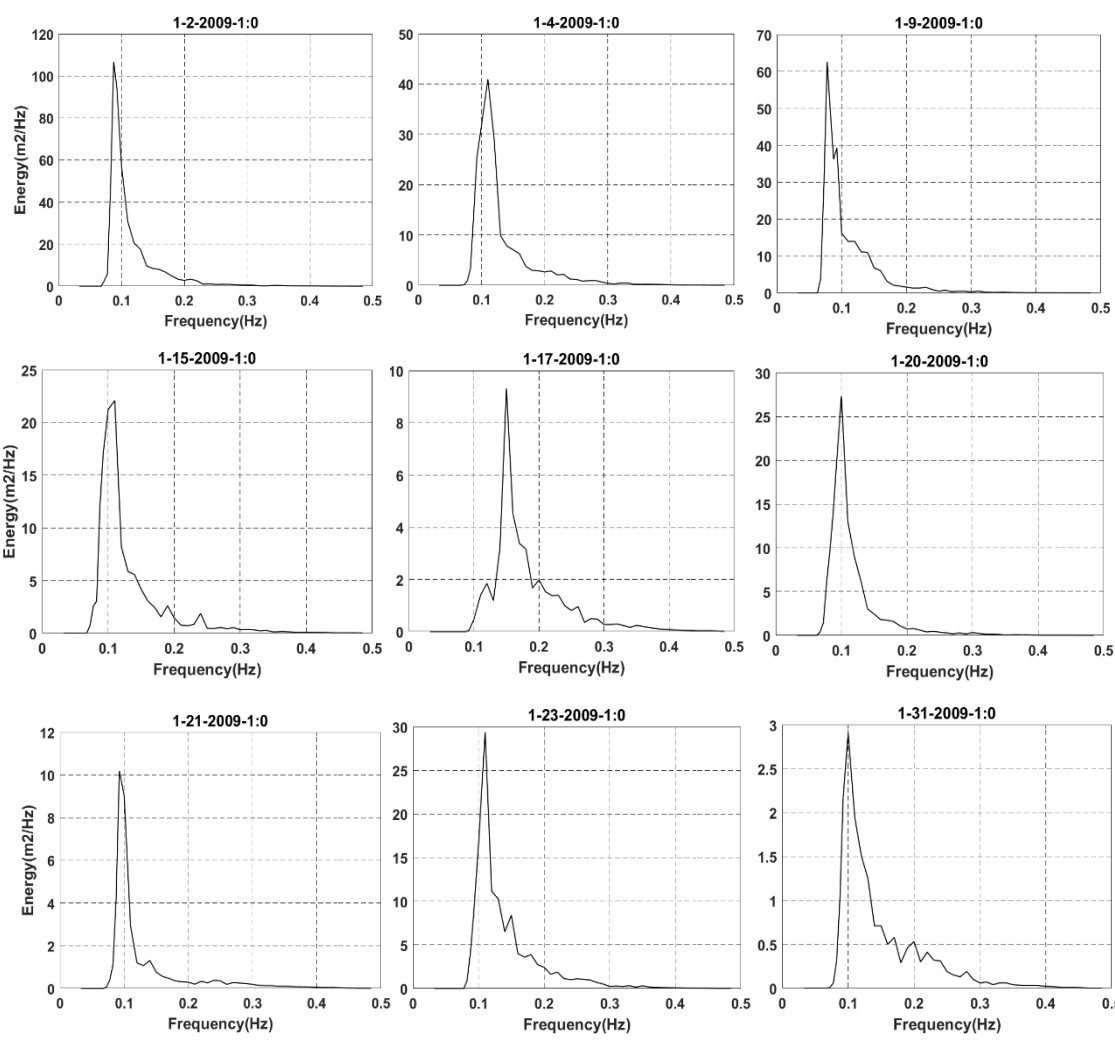

**Figure3: Examples of single-peaked frequency spectra measured at NDBC 44011 in January 2009. The title for each panel shows the date and time of measurement with the format month-day-year: hour: minute**


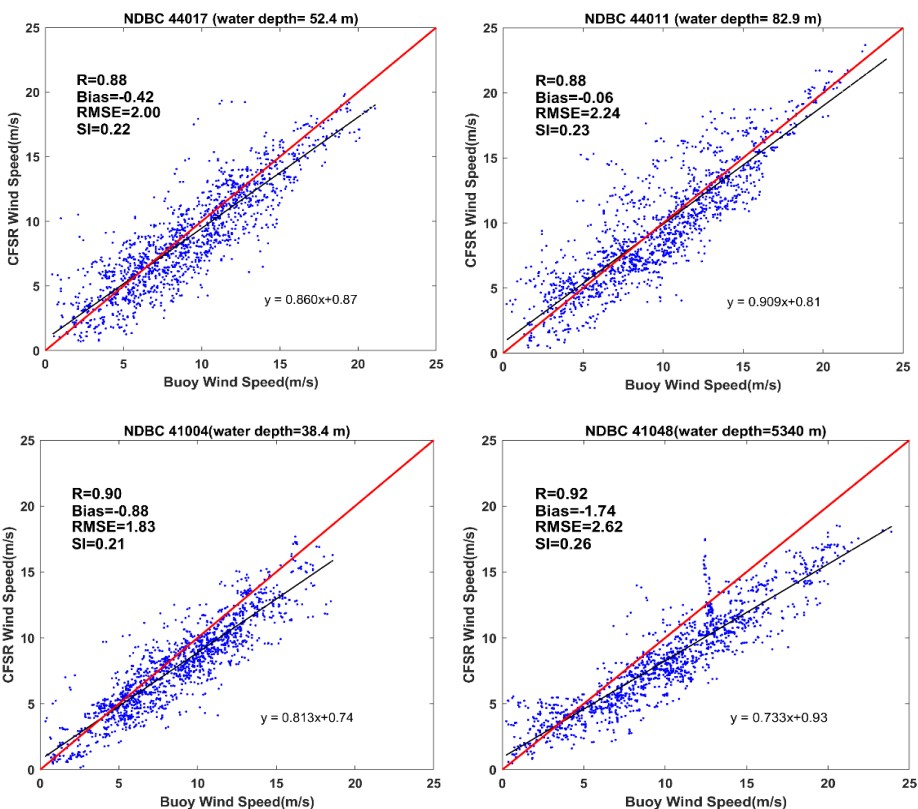

**Figure4: Evaluation of the CFSR wind field versus measured wind by NDBC buoys at four stations shown in Figure1**







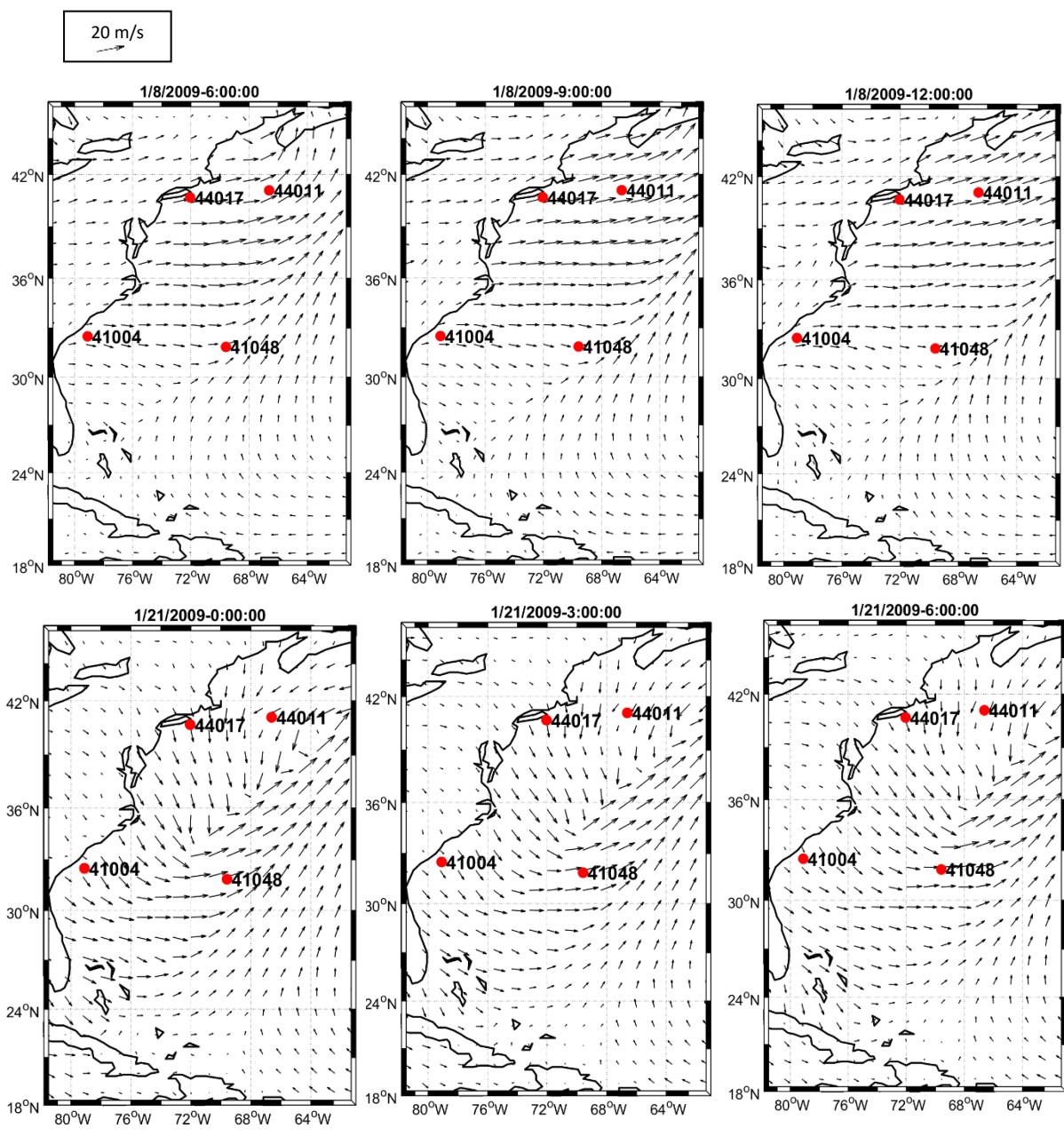


**Figure5: Three-hourly snapshots of CFSR wind fields over the modeling area ending at times t1 (1/8/2009 12:00, upper panels) and t2 (1/21/2009 06:00, lower panels).**



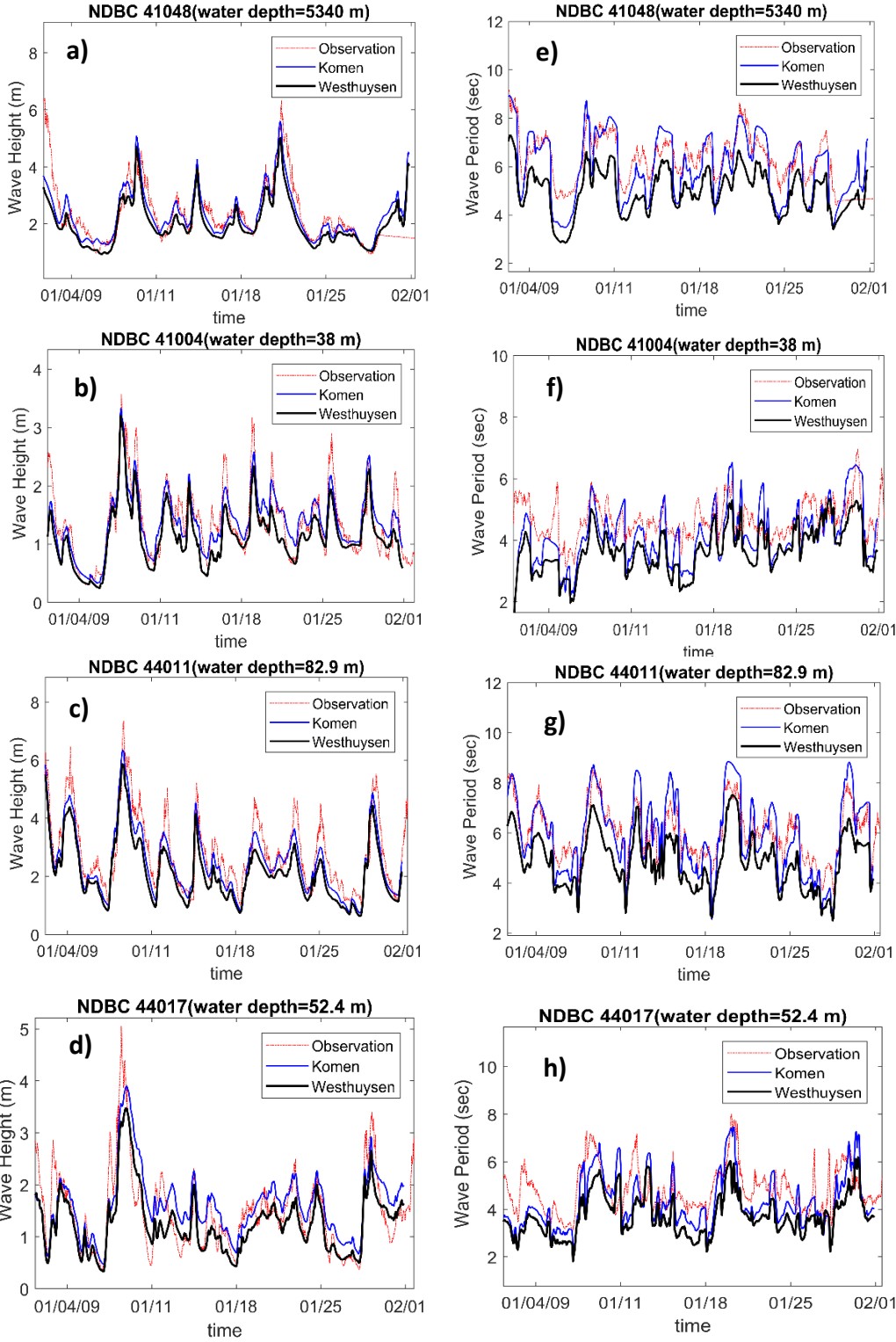

**Figure6: a-d) Time series of simulated wave heights and e-h) simulated mean wave periods, using Komen (blue lines) and Westhuysen (black lines) whitecapping formulas compared to measurements at the four NDBC buoys (dashed red lines).**

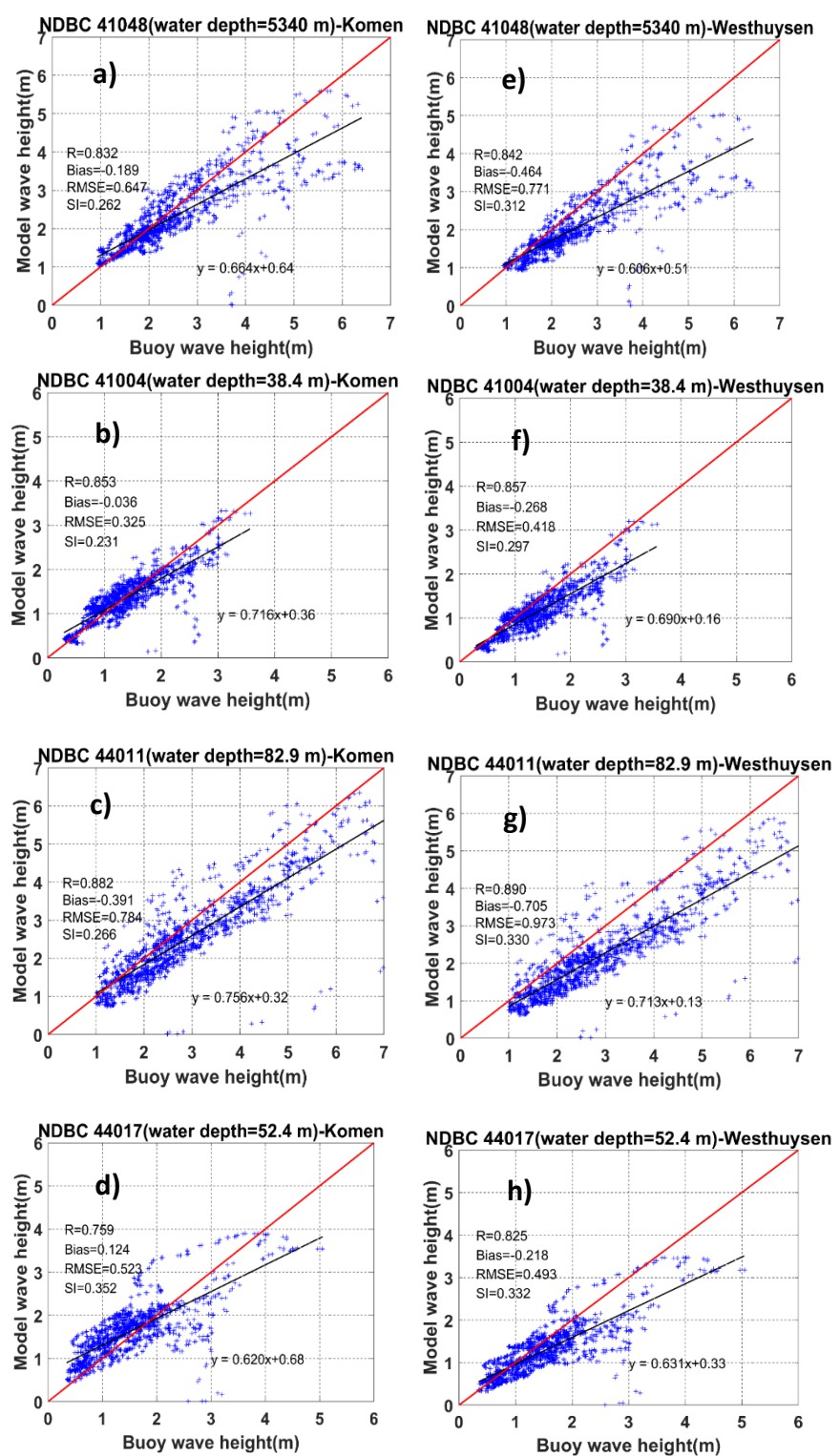


**Figure7: Scatter plots and model performance metrics for simulated wave heights using a-d) Komen whitecapping, and e-h) Westhuysen whitecapping at the NDBC buoys. The red and black lines are 1: 1 line and regression line respectively.**

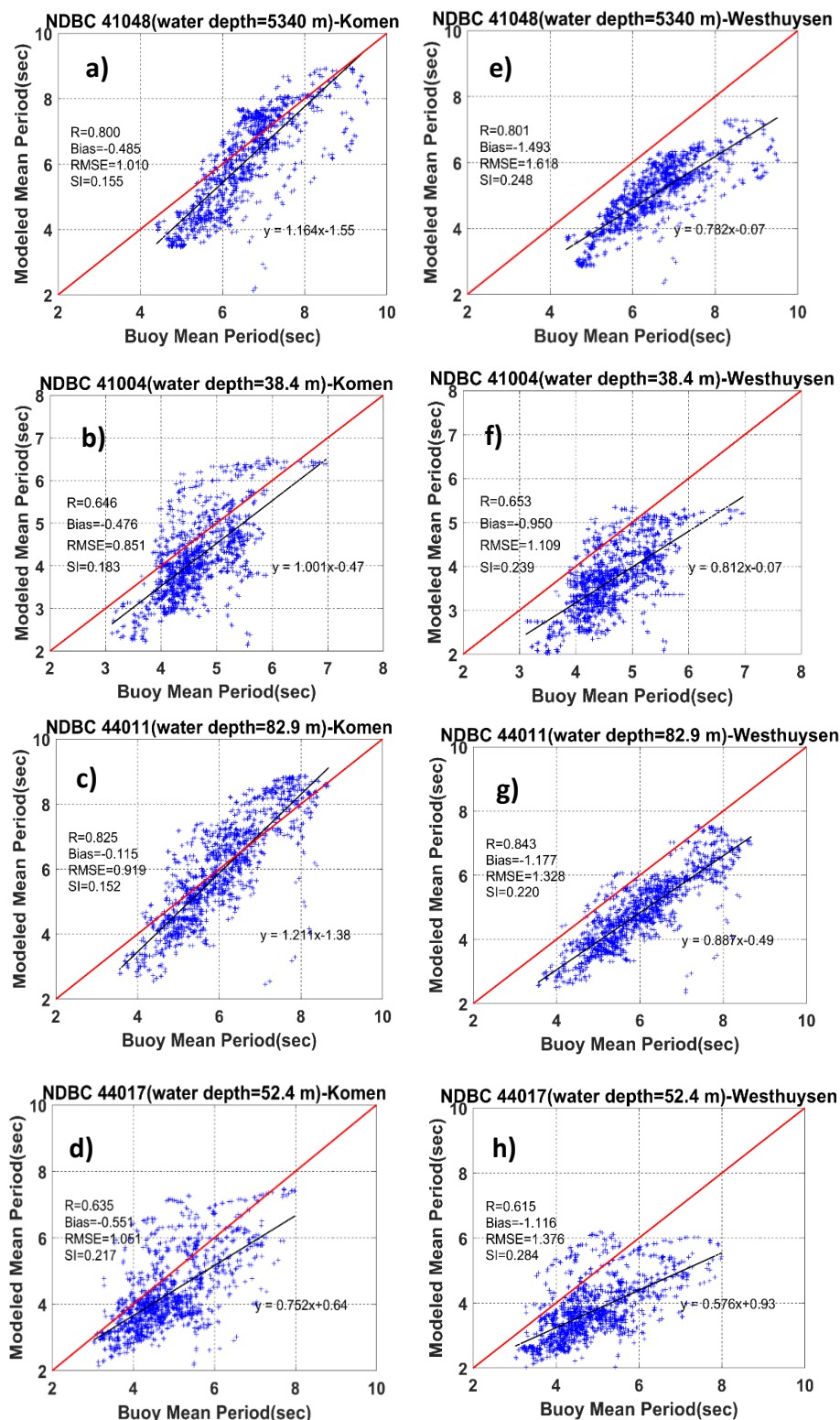

**Figure8: Scatter plots and model performance metrics for simulated mean wave period using a-d) Komen whitecapping, and e-h) Westhuysen whitecapping at the NDBC buoys. The red and black lines are 1: 1 line and regression line respectively.**


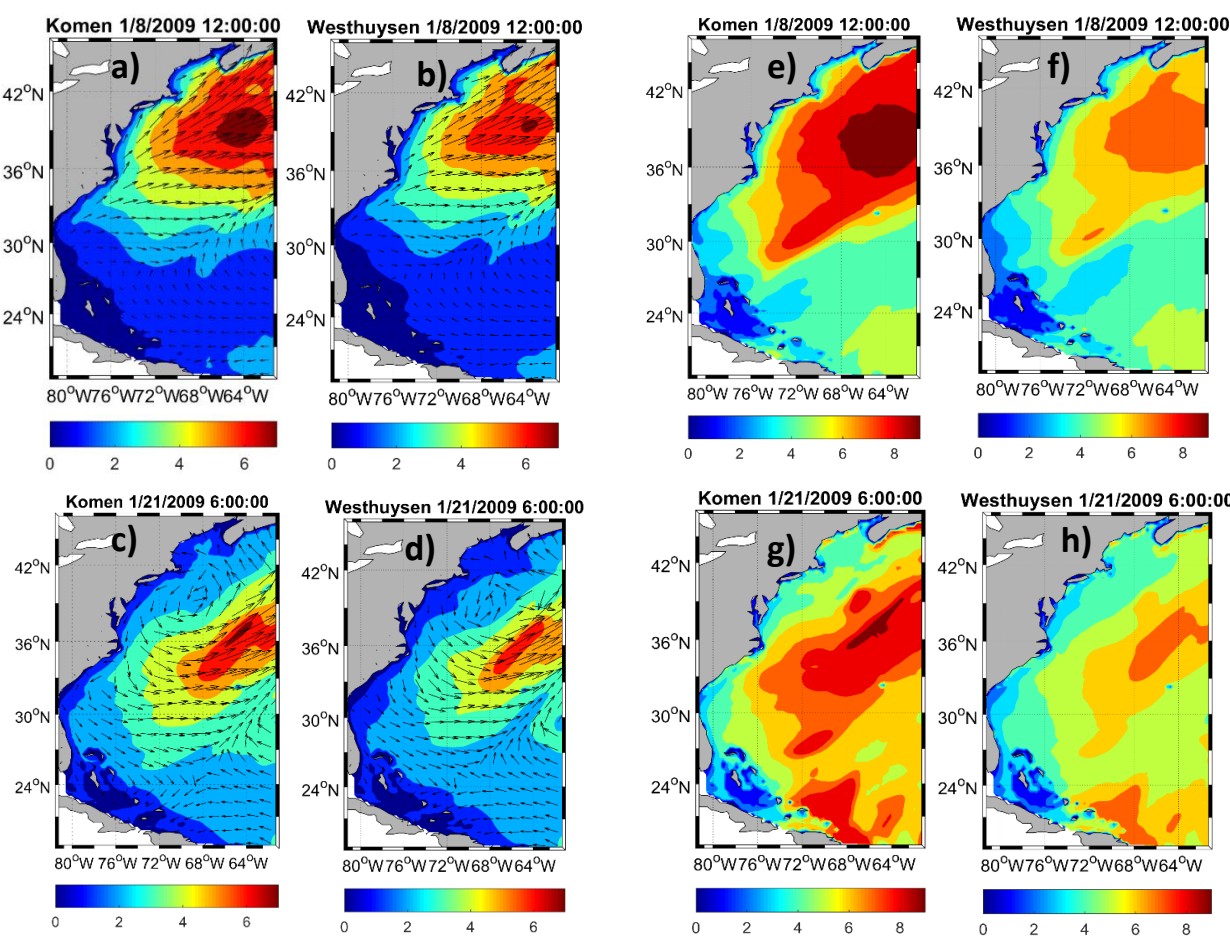


**Figure9: Simulated wave height (H<sub>m0</sub>) and direction over the modeling area using Komen and Westhuysen whitecapping formula for times t1 (a and b) and t2 (c and d) and simulation results for mean wave periods (Tm<sub>02</sub>) for times t1 (e and f) and t2 (g and h).**




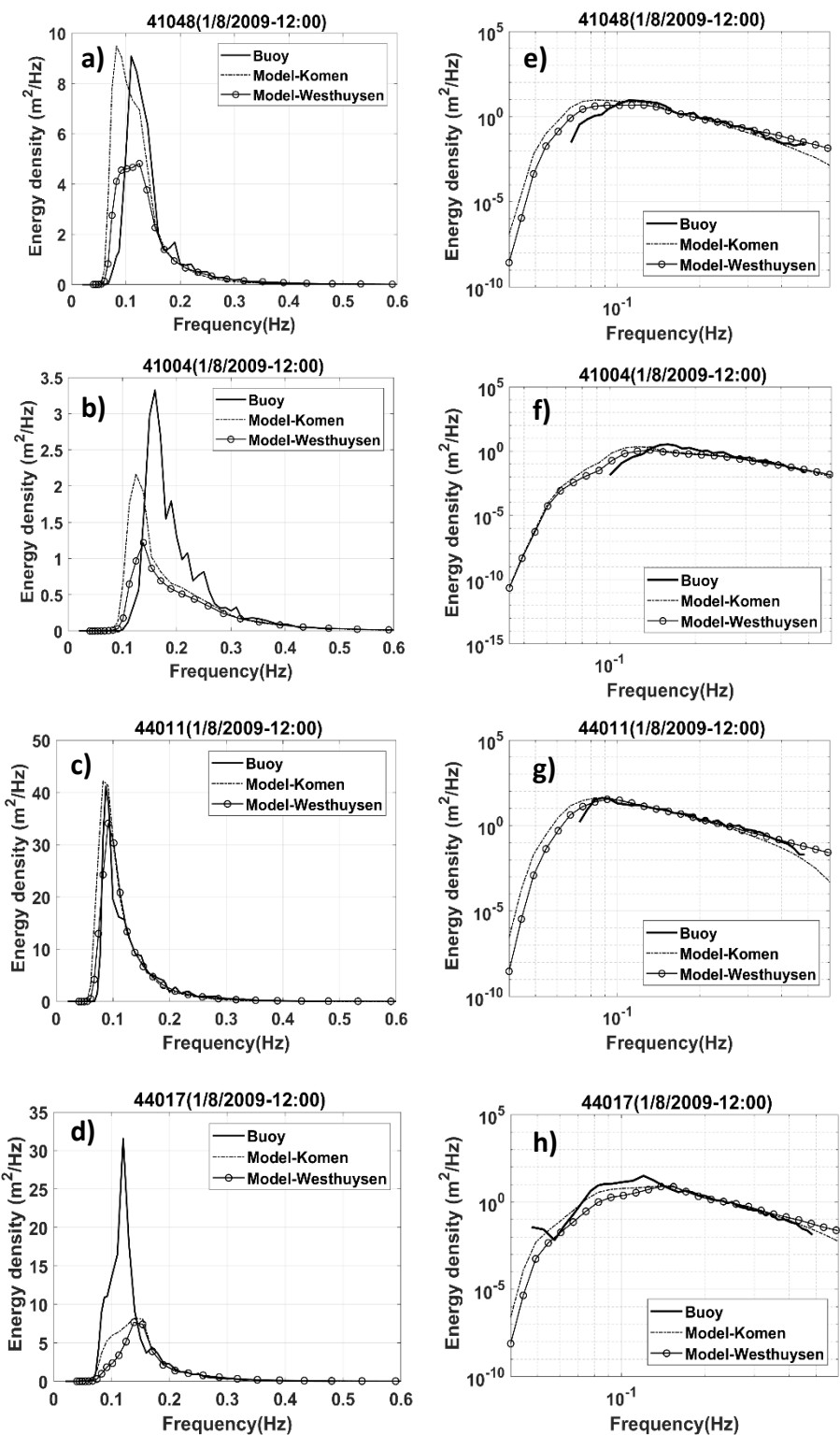

**Figure10: Comparison between the observed (solid lines) and simulated (dashed for Komen, circles for Westhuysen) frequency spectra at t1 at the four NDBC stations: a,c,e,g) linear scale for the energy axis and b,d,f,h) logarithmic scale for the energy axis.**


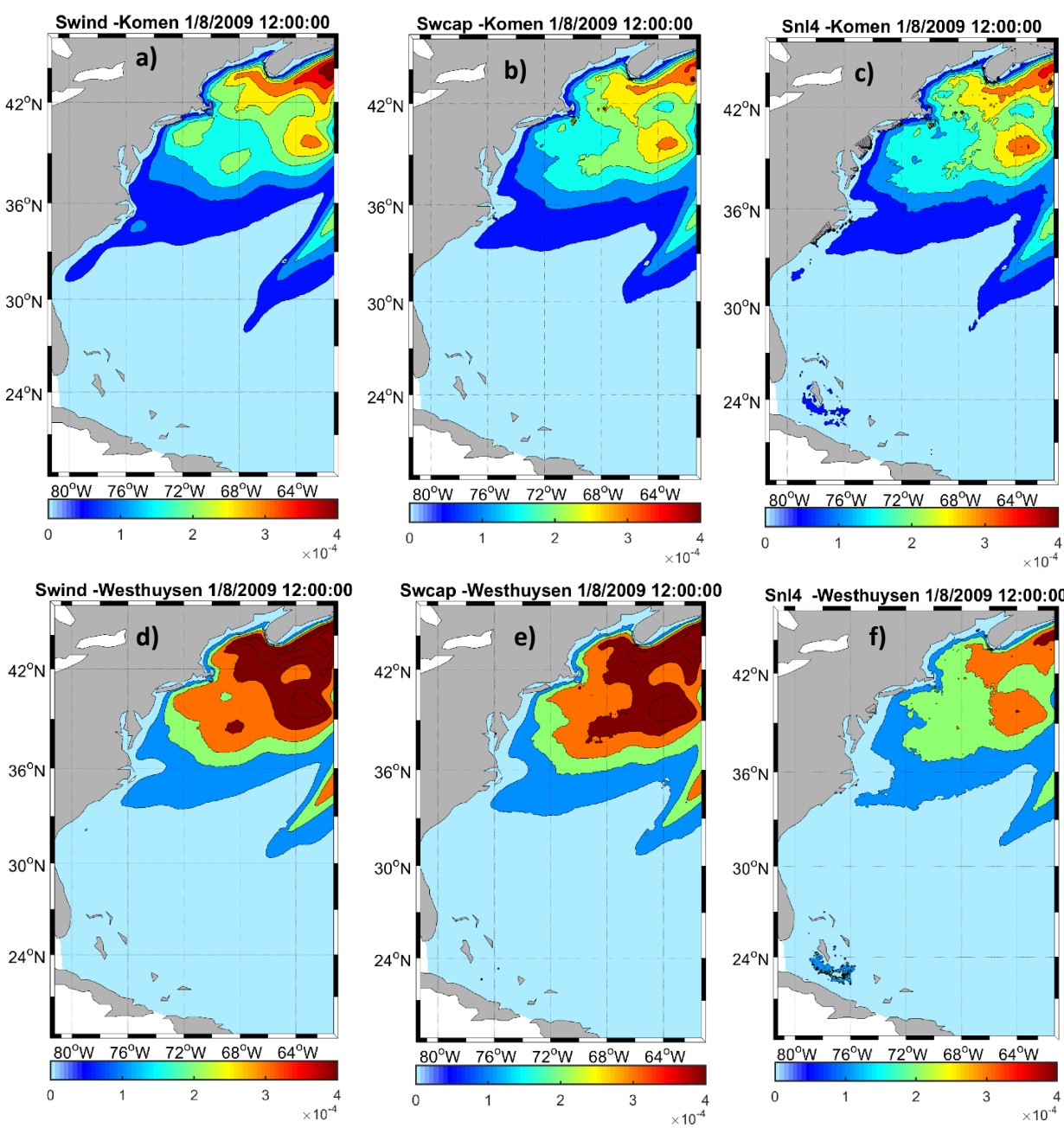

**Figure11: Spatial variations of source terms (wind input, whitecapping dissipation, and quadruplet in $m^2/s$) integrated in the spectral domain over the modeling area at time t1, for a-c) Komen and d-f) Westhuysen.**


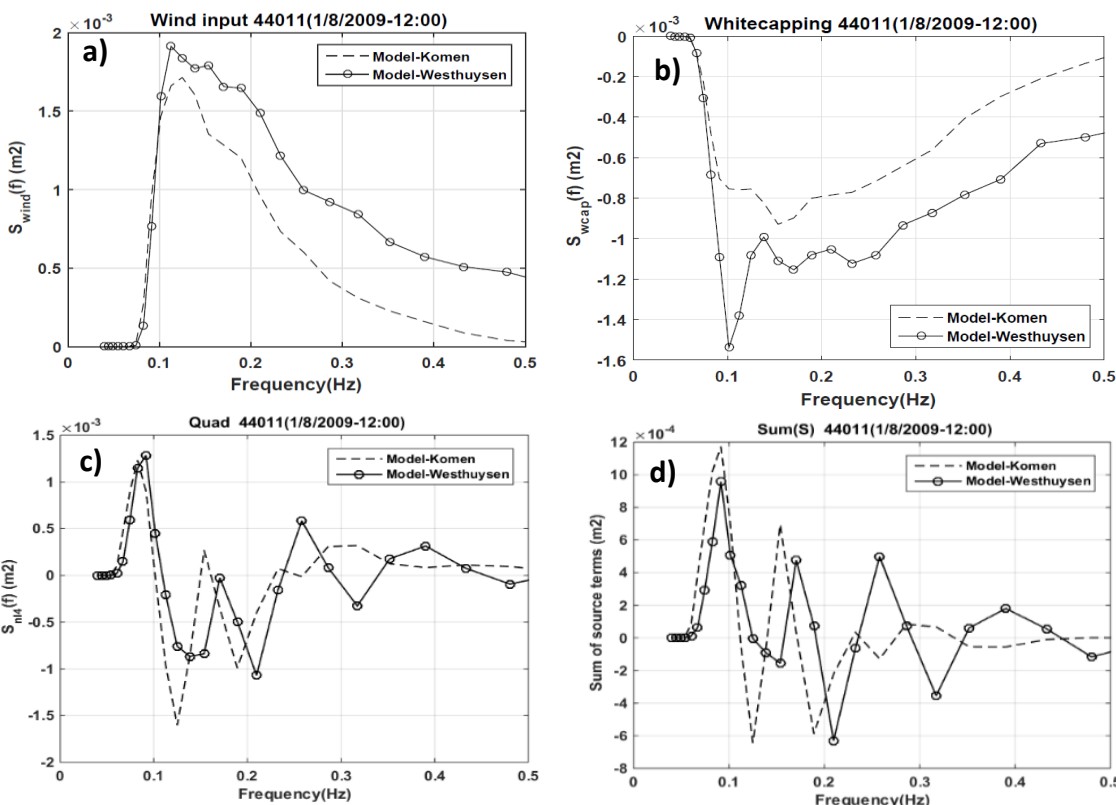

**920**

**Figure12: Variations of simulated source term components with frequency at buoy 44011 for t1: a) wind input, b) whitecapping dissipation, c) quadruplet, and d) algebraic sum of these terms.**

**925**

**930**

**935**

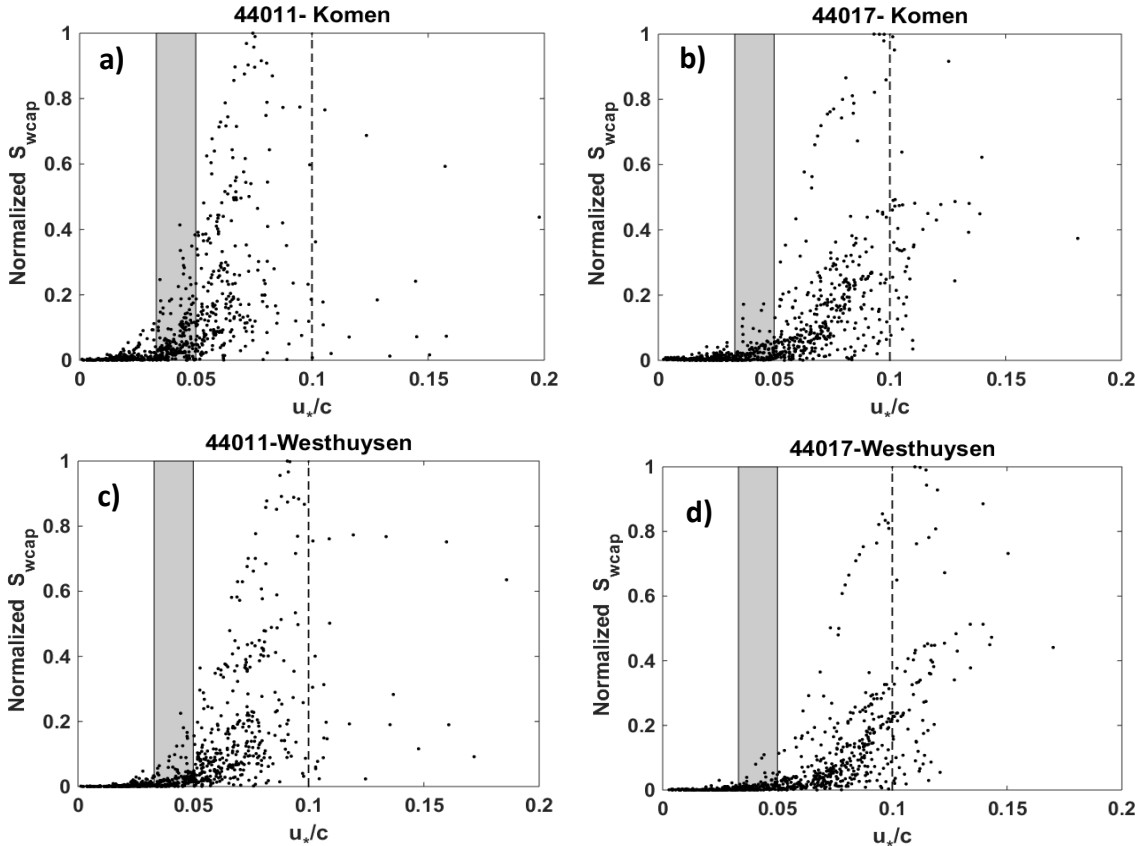

**Figure13: Variations of the normalized whitecapping dissipation ($S_{wcap}$) from Komen and Westhuysen simulation scenarios with the inverse wave age $\frac{u_*}{c}$ at a) and c) NDBC 44011, and b) and d) NDBC 44017. Gray boxes indicate the fully-developed zone ($0.033 < \frac{u_*}{c} < 0.05$). The dashed line separates zones for linear and quadratic growth based on Yan(198y) as indicated by $\frac{u_*}{c} = 0.1$.**

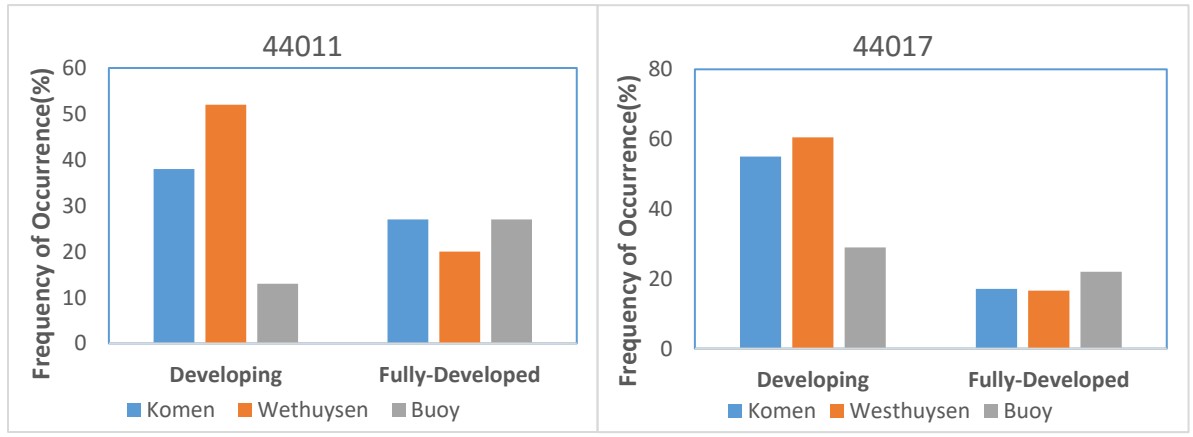

**Figure14: Frequency of occurrence for "Developing" and "Fully-Developed" sea states (based on Volov (1970) and Oost (1998)) at stations 44011 and 44017 from buoy data and simulations using Komen and Westhuysen whitecapping formula**


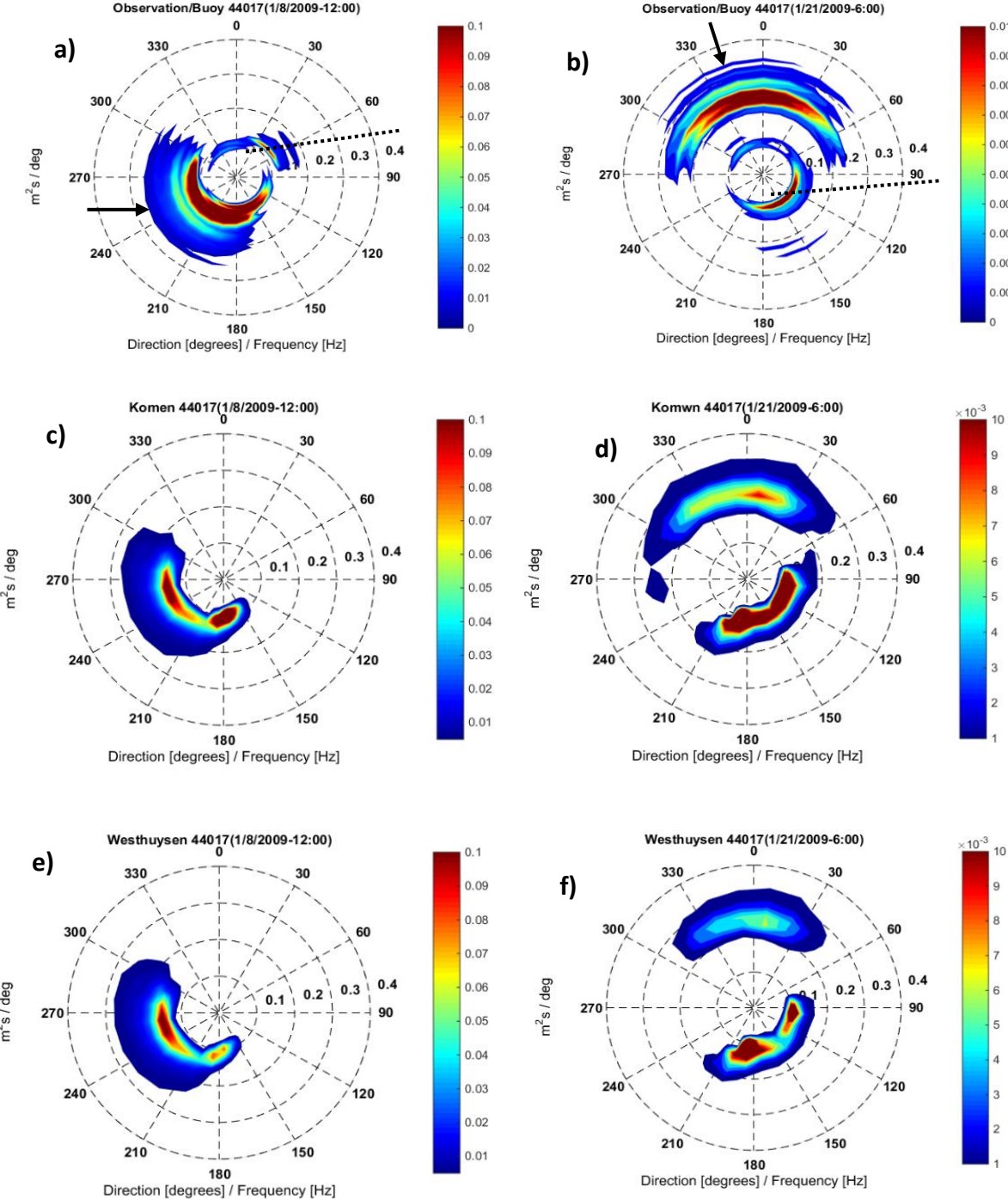

**Figure15: a,c,e) Frequency-directional spectra from observation, Komen simulation, and Westhuysen simulation, respectively at 44017 for time t1, b, d, f) the same spectra at t2. The solid arrows show the direction of observed CFSR wind at the buoy. The direction of the shoreline in the vicinity of the buoy is shown with dotted lines.**


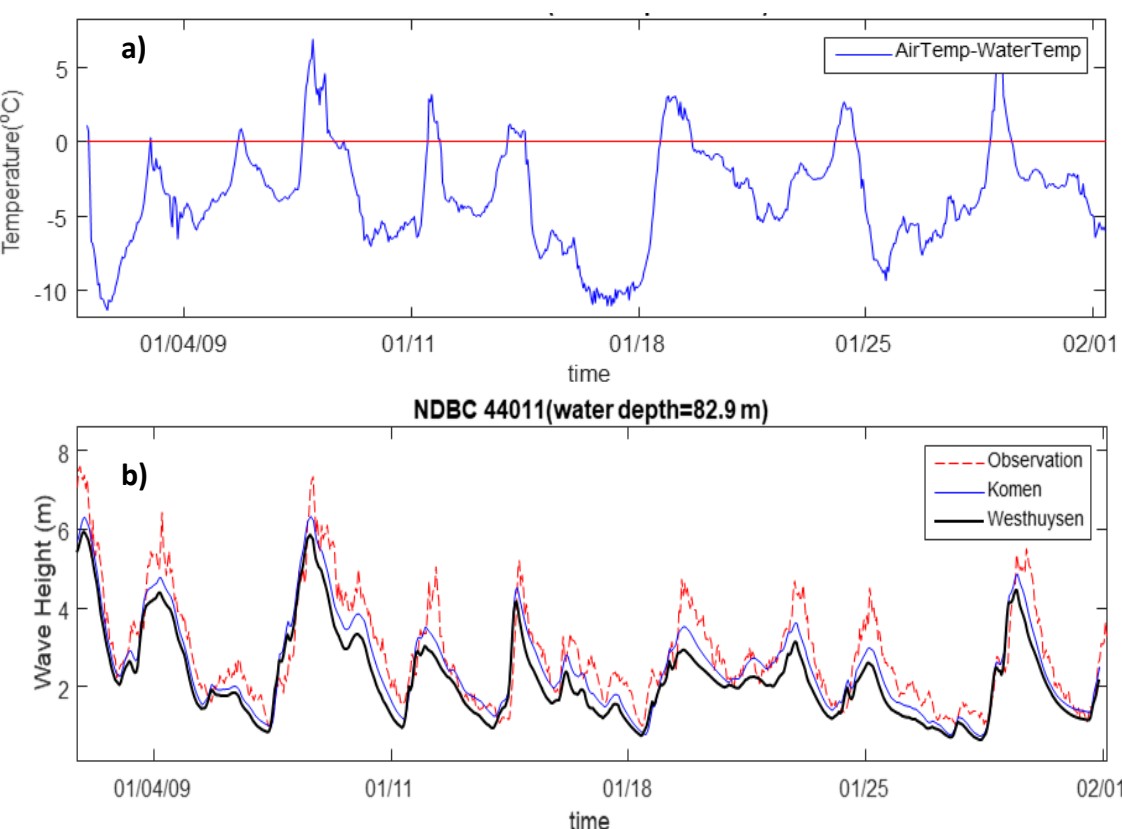

**Figure16: a) Time variations of the observed temperature difference between air and sea surface during January 2009 at buoy 41011. b) Time series of observed and simulated wave heights during this period.**


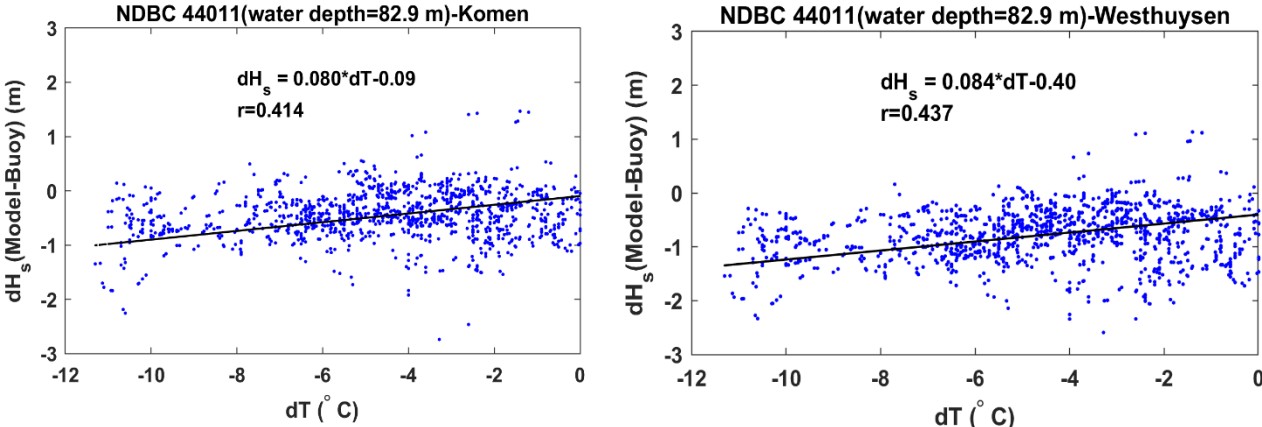

**Figure17: Correlation between the air-sea temperature difference and deviations of the simulated wave height from the measurements at NDBC44011 for two whitecapping formulations**
