# Peer review of "Predicting Ocean Waves along the U.S. East Coast During Energetic Winter Storms: sensitivity to Whitecapping parameterizations"

_Ocean Science, 2018_

## Referee Comment (RC1) · Anonymous Referee #1 · 24 Nov 2018

The authors present the results of a sensitivity study of applying two wave model settings concerning the whitecapping parameterisations. As the results of the different settings show (as expected) different results, the authors performed an analysis to explain the differences and determined the range of applicability. Although the authors show some knowledge about spectral wave modelling, they fail to make a sound manuscript and I cannot recommend it for publication. One of the reasons is that no proper research question is posed. Just comparing different model outcomes is a trivial exercise, but the lessons learned are now too vaguely investigated or described.

[Figure]

This becomes apparent as the words 'could be' occur at too many places (9 counted). In general, the manuscript is too descriptive and too less an in-depth analysis. For instance, in the discussion of growth curves, a possible problem is suggested, but no work has actually been done to make a further step. A similar remark can be made about the effect of atmospheric stability.

Another objection is that a large part of the analysis is to try to explain differences in source term behavior. There are some nice observations, but as the spectra themselves are already different it makes no sense to draw conclusions about source term differences. A sounder comparison is to start with equal spectra followed by investigating the source term response.

Concerning the choice of whitecapping source terms, various aspects are missing in their description. An essential part of the Komen type whitecapping is the use of a mean wave steepness. This is briefly mentioned when introducing the Westhuysen term, but hardly any systematic analysis is done. It is known for many years that this gives problems in bi-model seas with a swell and a wind sea component. Playing with the delta parameter, see for instance, Rogers et al. (2003, JPO, 33) and Pallares et al. (2014, CSR, 87) discuss this effect. Concerning the Westhuysen setting, the author do not seem to be aware of Babanin and Westhuysen (2008, JPO, 38). Moreover, recent developments in source terms, for ease referred to as the ST4 and ST6 versions in Wavewatch, are not mentioned at all.

The quality of any wave model hindcast is largely dependent on the quality of the wind fields. In this study no attempt has been made to validate the wind.

The notion that source terms should be recalibrated for different type of fetches or limitations is an interesting notion, but a more important conclusion is that the chosen formulations suffer a number of (already known) shortcomings, which can be remedied by introduction of more sound physics, viz. the parameterisations. In that sense, the present manuscript adds nothing new and recommends the wrong approach to improve

spectral wave modelling in coastal areas.

The use of snapshots of spectra and then draw conclusions about model behavior is not sound science. This is statistically seen of no value.

The mean wave period is not defined. It is not clear whether this is Tm01, Tm02 or even Tm-10 (also referred to as TE). Same, for the definition of mean wave direction.

The authors suggest in the introduction (#26) that certain slanting fetch effects can not be generated due to short comings in the wind field. This notion is wrong, as it is a complex interplay of all source terms, see Ardhuin et al. (2007).

**41 Quantifying the source terms still is a challenging task, just look at the developments around more modern source terms like ST4 and ST6. Having said that, I feel the choice of the present authors is outdated.**

**61 What are small coastal areas?**

**64, #67 Reference missing in list of references, or is there a typo?**

**78 which specific formulations are meant.**

**87 The authors discuss seasonality as being important, but their analysis is based on only one month of data. Again, the authors miss an opportunity to study their problem in depth.**

**95 The description of the role of the delta parameter is wrong. Yes, it has an effect on periods, but this parameter is not the related to another basic effect**

**145 Looking up Allahdadi et al. (2018) gave only an abstract. This makes it impossible to judge its content and how it relates to the present manuscript.**

**154 A wind rose would reveal in proper detail what is dominant.**

**156 Local wind and distant swell are unrelated. Therefore, this reasoning is flawed, unless the others refer to different phenomena. Authors could also have used spectral**

partitioning to select events of interest. Now, it is possible that their data are full of 'swell' noise.

**182 Also convergence criterion important, like number of iterations per time step**

**201 The added value of the referenced papers is unclear. You do not need such references for basic statistics.**

**209 'Could be' An example of speculation**

**216 Snapshots are insufficient to draw conclusions**

**232 The $E(fp)$ is not a suitable parameter to draw conclusions, moreover as only one event is shown. It has only one degree of freedom. More robust methods are using bulk statistics of different period measures like Tm01 and Tm-10, which have different emphasis on weighting frequency bands**

**238 'could be ' another example of speculation.**

**284 quadruplet terms cannot be predicted by Komen or Westhuysen source terms.**

**305 it is unclear how the actual fetch lengths are estimated in the open ocean.**

**311 what are the defaults. As they may change, it is better to specify them for instance in an appendix.**

**346 'Presumably', again speculation**

**355 'may cause', speculation**

**362 why are small variations in speed and direction relevant in this context**

**368 I doubt whether revisiting the calibration process is the proper way. This can be a short-term fix, but rather difficult to obtain sound results. This notion is more an indication that the present physics in Westhuysen and Komen is incomplete. Sticking to outdated source terms does not seem a viable option**
**380 Concerning land-sea effects on the development of wind speed, the authors can refer to papers from e.g. Dobson et al. (1989, Atm_Ocean, 27) and Taylor and Lee (1984).**

**402 it is not how the 2D-spectra were reconstructed from the measurements in Figure 12**

**420 The discussion on temperature effects is interesting, but nothing is done with this. Moreover, this is effect is known already.**

**452 Do these so-called intense wind speeds occur in the present hindcast, and does the Yan parameterization really have an effect at the buoy locations**

**455 I do not understand the reasoning about the computation of the quadruplets.**

**476 What makes a coastline complicated? It all has to do with scales. The interpretation of the causes of the slanting fetch effect is incomplete. Directionally dependent fetch-lengths and the quadruplet interactions also play a role.**

**479 See Ardhuin et al. (207) for an in-depth discussion**

**480 Reasoning is incorrect. In case low-frequencies are underestimated, this may be due to the effect of the whitecapping term. Now the authors suggest it leads to an overestimation.**

**488 What do the authors imply mean 'a spurious effect'**

**490 Good idea to study all seasons. As in this manuscript only one month is used, the conclusions are not backed with sufficient data.**

Table 2 Just providing data for one time instant is statistically insufficient and completely meaningless. Furtherm the mean wave period is not defined

Figure 2 A wind and wave rose would be more informative.

Figure 3 The snapshots are too similar to be of value. A possible link with the determination of a fetch length could add some value to this manuscript.

Figure 5 The causes of the discrepancies can also lie in incorrect winds. But nothing is said about the quality of the winds. Some details, add grid lines and close the rectangle.

Figure 6 Same as for Figure 5

Figure 7 The arrows with the mean direction are too small. Further, what is the definition of mean wave direction?

Figure 9 Nice pictures, but of little value as the underlying spectra are also different. Then the source terms will also be different and no firm conclusions can be drawn.

Figure 10 Why are the Snl4-terms oscillatory?

Figure 13 This figures confirms known concepts, but nothing is done in this study.

[Figure]

---

## Referee Comment (RC2) · Anonymous Referee #2 · 28 Nov 2018

The authors evaluated the performance of two different methods for parameterizing the whitecapping effect: 1) Komen which is based on the mean spectral parameters, 2) and van der Westhuysen approach which is based on the saturation level concept of the wave spectrum. This sensitivity analysis was performed in the winter, covering the storms of January 2009. They performed match-up comparison between their results at four NDBC stations. Their results showed that both approaches underestimate wave height at all stations, but still Komen performed better than the Westhuysen. Also, the Westhuysen approach underestimates mean wave period and the total wave energy

as well, whereas Komen did a good job in the modeling of the wave period. In addition, the authors discussed the effect of coastal geometry, the effect of boundary layer instability, as well as the effect of wind field and growth conditions. The results and the methods are applicable, and useful for physical oceanographers using SWAN itself or Delft3D model, since the SWAN model is embedded in Delft3D model. Although other models such as WWIII model seems also appealing to researchers, but still SWAN model can be more efficient due to its flexible mesh option while these two models sharing a very similar theoretical background. I believe SWAN was a suitable choice and computationally more efficient in this study, considering the areas were modeled. Komen and Westhuysen are still available in SWAN and are still widely used for wave simulation all of the world, so that's worthwhile to evaluate. The results from this paper are very interesting and scientifically valuable. Results are presented in concise and convincing ways. Their results can be used for other areas, for example in the Gulf of Mexico, where recently folks have a difficulty to select the most appropriate method to parameterize the whitecapping effect. I would like to recommend this paper with minor revisions.

**51: the author mentioned that Komen method tends to underestimate in different regions. The author should be specific about these regions. Are these regions are shallow water, deep water? Also, at least two references should be added to this part. #63: Some more findings from Mulligan et al., (2008) and W007 must be mentioned #77: Did the author observe such this variability in their wind data, if so, how much? # general comment: be consistent with swell and swell waves. #146: In Figure 1, label the locations of places they mention in the text. Such a Gulf of Main, Rhode Island, and others. #146: how much variation? any idea? #168: provide the results for CFSR evaluation. Such as R2 or RMSE, or others. Support your evaluation results in providing other references other than the mentioned one. #191: wave height? do you mean significant wave height, be more specific. #201: you don't need to put these references. # In Figure 5 and 6: explain what those are black and red lines (regression line and 1: 1 line) in the captions. #Again, label Florida and Massachusetts**

coast on Figure 1 #228: Could you provide any references that support the averaging method could minimize the noises? #270: Show the location of very northeast edge on the map and explain about the wind condition during the simulation period on this location. #314: state the default values. #374: supply this paragraph with appropriate references. #406: remove the space between 12 and d. And be consistent about the figure's captions (e.g., Figure 10 or Figure10) #Add a few lines (2 or 3lines) discussing if the mentioned results considering these two approaches are applicable for other seasons since the authors only performed the sensitivity test for the winter.

---

## Referee Comment (RC3) · Anonymous Referee #3 · 6 Dec 2018

Authors made significant efforts to evaluate the two mostly used white capping parameterizations in SWAN, viz., Komen and Saturation based Westhuysen approach. The simulation results clearly support the fact that Komen approach is the best for simulating bulk wave parameters as well as the spectra, during the winter weather conditions, for the US East coast. The horizontal resolution for the NCEP/CFSR wind input is $\sim$ 32 km for the study area, which would be sufficient for capturing the offshore directed wind forcing. However, as the authors observed, quoting Ardhuin et al., (2017), CFSR wind data is insufficient for capturing the short period gusting in the wind field, which

is quite common during January-March period in the study area. The simulated wave conditions at NDBC 44017 showed distinct variations in terms of its sensitivity to the two white capping parameters. As authors pointed out, the coastal geography played a significant role in the wave generation and its evolution for this station. Authors also provided a detailed discussion on comparing the varying source terms; effect of wind field and its effect on the wave growth; effect of coastal geometry, and finally effect of boundary layer instability etc. The study confirmed that, both Komen and Westhuysen implementations in SWAN and WAM models are comparable in their performance, with Komen approach consistently showed better agreement with in situ observations. In order to examine the slanting fetch effect on the coastal buoy locations, two things could be considered, which may not necessarily required for this study. It would be ideal to simulate wave conditions for easterly wind episodes and compare the performance for the two parameterization schemes; data assimilation also would be an option, see (Orzech et al., 2013; Almeida et al., 2016.). Figure 3; a wind vector is required for scaling purpose Figure 8: please correct the units along Y-axis

---

## Author Comment (AC1) · 16 Dec 2018

Dear Reviewer

Thank you very much for your time reviewing our paper and for your constructive comments. Your comments for revision of the paper are addressed below:

**51: the author mentioned that Komen method tends to underestimate in different regions. The author should be specific about these regions. Are these regions are shallow water, deep water? Also, at least two references should be added to this part.**

[Figure]

The underestimation associated with the Komen-type whitecapping models occurred in both deep and shallow water regions. Examples of this behavior are presented by several studies, for example, van Vledder (2016), Allahdadi et al(2017), and Siadat-mousavi et al(20111). These details will be included in the modified manuscript.

**63: Some more findings from Mulligan et al., (2008) and W007 must be mentioned**

More details from these two papers that evaluated Komen and Westhuysen white-capping formulas over both closed and marginal seas will be added to the revised manuscript.

**77: Did the author observe such this variability in their wind data, if so, how much?**

The mentioned variabilities in wind speed and direction (gustiness) occur within short time slots (seconds to minutes) that cannot be considered by atmospheric models with hourly to several hourly outputs (Abdallah and Cavaleri (2002)). In our simulation, we used the CFSR wind field with 1-hour temporal resolution that cannot include such short-term variations. We will mention it in the revised manuscript. We mentioned the gustiness as an example for a shortcoming of the traditional whitecapping models, but including this effect was not a part of our study scope.

**general comment: be consistent with swell and swell waves.**

We believe that swell wave is a redundant expression. We will consistently use swell in the revised manuscript.

**146: In Figure 1, label the locations of places they mention in the text. Such a Gulf of Main, Rhode Island, and others.**

The labels will be included in Figure 1 in the revised manuscript

**146: how much variation? any idea?**

This sentences about the amount of seasonal variations of wind will be added to the revised manuscript: "For example during January 2009, the average wind direction

along and off the U.S East Coast was northwesterly to westerly with average wind speed of 6.5 m/s, while for July 2009 average wind direction was southwesterly with an average wind speed of 3.5 m/s."

**168: provide the results for CFSR evaluation. Such as R2 or RMSE, or others. Support your evaluation results in providing other references other than the mentioned one.**

The evaluation results along with the statistical parameter will be included in the revised manuscript. Additional reference will be added to support the results.

**191: wave height? Do you mean significant wave height? be more specific.**

Yes, this is significant wave height. Will be modified in the text.

**201: you don't need to put these references.**

The references will be removed

**In Figure 5 and 6: explain what those are black and red lines (regression line and 1:1 line) in the captions. Figure captains will be modified to include definitions for the 1:1 line and regression line.**

**Again, label Florida and Massachusetts coast on Figure 1 Labels will be added to the figure. #228: Could you provide any references that support the averaging method could minimize the noises?**

Time-averaging of the spectra has been suggested by van der Westhuysen et al (2007) that will be mentioned in the revised manuscript.

**270: Show the location of very northeast edge on the map and explain about the wind condition during the simulation period on this location.**

The location of "very northeast edge" will be marked on the map. Wind condition at this location will be discussed based on the snapshots in Figure 3

**314: state the default values.**

For the saturated-based mode the defaults for the whitecapping parameter the saturation threshold are Cds=5.0×10−5 and Br=1.75×10−3, respectively. For Komen, the default for the whitecapping parameter is Cds=2.35×10−5. These vales will be included in the revised manuscript.

**374: supply this paragraph with appropriate references.**

The appropriate reference for this paragraph (Hasselman et al.(1973) ) will be included in the revised manuscript .n

**406: remove the space between 12 and d. And be consistent about the figure's captions (e.g., Figure 10 or Figure10)**

Modifications will be applied to the revised manuscript.

**Add a few lines (2 or 3lines) discussing if the mentioned results considering these two approaches are applicable for other seasons since the authors only performed the sensitivity test for the winter.**

The likely different behaviors of models during the summer with more occurrence of Atlantic swells have been mentioned in the last paragraph of Summary and Conclusion (486-490) of the original submission.

Regards Allahdadi et al

––––––––––––––––––––––

---

## Author Comment (AC2) · 17 Dec 2018

Dear Reviewer Thank you very much for your time and efforts for reviewing our papers. Responses to your comments are as follows:

comment: However, as the authors observed, quoting Ardhuin et al., (2007), CFSR wind data is insufficient for capturing the short period gusting in the wind field, is quite common during January-March period in the study area

Response: As you mentioned, the timescale for the wind gusting is very short. In fact,

it is in the scale of seconds to minutes (Abdallah and Cavaleri (2002)) that cannot be captured by the common wind models with hourly to 6-hourly output. Modification of wind input and whitecapping formula to include the effect of these fluctuation even in case of using wind model outputs is one of the ways that can increase the accuracy of wave models. It was not the focus of our study and was just mentioned as an example about the shortcoming of the available whitecapping formula in SWAN or other models. We will discuss it more in the paper introduction.

Comment: In order to examine the slanting fetch effect on the coastal buoy locations, two things could be considered, which may not necessarily be required for this study. It would be ideal to simulate wave conditions for easterly wind episodes and compare the performance for the two parameterization schemes; data assimilation also would be an option, see (Orzech et al., 2013; Almeida et al., 2016.).

Response: We agree that considering easterly wind episodes can shed more lights on the effect of the coastal geometry on wave condition at coastal buoys. As we mentioned in the summary and conclusion section, we considered a complementary paper to study the performance of two whitecapping approaches during other seasons including summer. They could include events of the easterly winds that will be studied in detail. About the data assimilation option: again we agree with the reviewer. However, our main goal in this study was only detecting the problem and probable causes, not diagnosing the solutions. That is why we only used the default parameter to challenge the original parametrization of the approaches. Calibration and its advanced form as data assimilation could be very helpful for improving the performance of these model like the calibration process that we completed in Allahdadi et al (2018).

Comment: Figure 3; a wind vector is required for scaling purpose Figure Response:Reference wind vector will be added to Figure 3 in the revised manuscript.

Comment: Figure8: please correct the units along Y-axis Response: The unit of Y-axis will be corrected in the revised manuscript.
Regards Allahdadi et al

---

## Referee Comment (RC4) · Anonymous Referee #2 · 25 Dec 2018

Dear editor, The authors replied to all comments, clarified unclear parts, and modified the manuscript based on my suggestions and requests. I recommend this paper for publication. Regards

---

## Author Comment (AC3) · 27 Dec 2018

Dear reviewer

Thank you very much for your thorough and constructive comments on our paper. Followings are responses to your comments and concerns: ——————————————————————————————————————————

*One of the reasons is that no proper research question is posed. Just comparing different model outcomes is a trivial exercise, but the lessons learned are now too

vaguely investigated or described:

We agree that the research question were not posed directly. However, they have been mentioned as gaps in science indirectly(pleases see #70-#75) as reads:

"Hence, more studies are needed to determine the appropriate ranges of application for each whitecapping method. Moreover, wave growth and dissipation can be significantly affected by variabilities in the wind speed and direction (gustiness), instabilities in the air-sea boundary layer because of air-sea temperature difference, and slanting fetches over the coastal areas (Ardhuin et al., 2007; Donelan et al., 1985)." Obviously our research questions are: what are the appropriate ranges of application for each of Komen and Westhuysen whitecapping formulations and what are the effects of different wind conditions and other factors like coastal geometry and instabilities in the air-sea boundary layer. Throughout the manuscript, these questions have been answered. We will mmention this question directly in the introduction part. Although our paper outline generally follows some previously published work (ex. Vledder et al. (2016), but with much more details about the source terms and justification of differences), will modify the manuscript to make the discussion points more clear.

This becomes apparent as the words 'could be' occur at too many places (9 counted).

Frequent use of 'could be' will be revisited and fixed. In some cases, we don't even need to do the speculations since we have technical references to approve the point we made (ex. 205-210, 235-240, 240-245 ), and therefore by adding those references we will make the point absolute and clear. Some other cases were not actually speculation and only the sentence need to be modified since we had enough evidences to make the argument absolute(ex. 320-325, 350, 350-355, 365-370). In one case (355-360) it is not even speculation, but figure of speech. We will fix the manuscript to avoid these speculation-like expressions as much as possible.

**In general, the manuscript is too descriptive and too less an in-depth analysis. For instance, in the discussion of growth curves, a possible problem is suggested, but no

work has actually been done to make a further step. A similar remark can be made about the effect of atmospheric stability.

For the case of the growth curves, we argued that there are two main reasons for deficiencies: 1. Cases of fully-developed events that deviated the results from the measured growth curves and 2) the frequent cases of duration-limited events that are not even included in the growth curves. The quantification in this part was presented based on the examined times t1 and t2 and comparing simulation results with the numerical values resulted from the growth curves of Calkoen (1992) (for fetch-limited growth) and Pierson-Moskowitz (1964) (for the fully-developed sea state). Quantification for the duration-limited case was not possible since by our knowledge and unlike the curves for other two sea states, measured growth curves for this sea state are not publicly available. However, the analysis related to figure 11 show the relative frequency of events for each case regarding the two examined whitecappinf models. We can elaborate more on interpreting this figure ex. the exact numerical values of the event frequency for each case and at each buoy. Regarding the presented results for studying the atmospheric instability, we agree that more analysis may be required. We will complete the discussion in this part by adding some statistical results for the correlation between the air-sea temperature difference and deviations of the simulation results from the measurements. Another objection is that a large part of the analysis is to try to explain differences in source term behavior. There are some nice observations, but as the spectra themselves are already different it makes no sense to draw conclusions about source term differences. A sounder comparison is to start with equal spectra followed by investigating the source term response.

The base for the analysis on the source terms is different wave height and wave periods that were resulted for same simulation conditions and two different whitecapping formulations. Hence, the spectra and source terms regarding specific times that exhibit significant differences between simulation results and observations were presented to figure out the reasons that cause the differences. This is a standard methodology that
was followed by many research studies (ex. van der Westhuysen et al., 2007; Mulligan et al., 2008, and Vledder et al.2016). The rational is simple: at a specific time for similar conditions, wave height and periods should be the same, but since the energy spectra and in fact the source terms are quantified in different ways, deviations in the integrated parameters were resulted. Our main argument in this paper was identifying the reasons that contributes to these differences in the context of assumptions and limitations associated with each whitecapping model. ***Concerning the choice of whitecapping source terms, various aspects are missing in their description. An essential part of the Komen type whitecapping is the use of a mean wave steepness. This is briefly mentioned when introducing the Westhuysen term, but hardly any systematic analysis is done.

We appreciate that the mean wave parameters including the steepness are the main contributor to the shortcomings associated with Komen approach for the cases of seas-swell combination. However, in the present study only wind-generated wave with single-peaked spectra (like Figure 8) are simulated due to the selected study period in January 2009. The main point of he present study is although Westhuysen whitcapping approach was introduced to overcome the inaccuracies associated with Komen method during the seas-swell combination, its performance for the purely wind-generated cases with no swell is lower than Koem which is consistent with Vledder et al. 2016. Furthermore, the main goal of the present study was examining the performance of these whitecapping formula within a real simulation framework, not a detailed examining of of the theoretical basis and suggesting a new form of the formulation. We will mention it in the introduction part to make the objectives more clear and avoid further complication for readers.

+Concerning the Westhuysen setting, the author do not seem to be aware of Babanin and Westhuysen (2008, JPO, 38). Moreover, recent developments in source terms, for ease referred to as the ST4 and ST6 versions in Wavewatch, are not mentioned at all.

We are aware of Babanin and Westhuysen(2008) paper when reviewing the theoret-
ical bases on the saturated-based model, although eventually we did not mention it in the manuscript. One reason that we did not mention it was that the focus of our research was not identifying the pitfalls in the theoretical background of the method, but examining the validity of both whitecapping approaches in a real simulation over an extensive region and prescribing the practical remedies for modelers. In the theoretical sense, Komen method has also several pitfalls in terms of inconsistency with observations, but again it was not our focus. The background behind our research is an extensive long-term wave modeling along the U.S East Coast for characterizing wave energy resources. We used SWAN for this modeling since due to complex coastal geometry we could benefit from the flexible mesh option made available by SWAN. The only whitecapping formulations that are available by SWAN+ADCIRC(we had to use this coupled version to implement the domain decomposing used in the high-performance coputation) are Komen-type and Westhuysen approaches and none of the physics packages incorporated in WWIII (ST2, ST4, and ST6) are available by this coupled version of SWAN. Although the observation-based physics package of ST6 has been recently added to the SWAN version 41.20, it has not been incorporated in the coupled SWAN+ADCIRC by the ADRIRC group. Hence, we need to deal with the available whitecapping approaches including Komen and Westhuysen if we want to use SWAN with the HPC ability which is vital for long-term modeling over an extensive modeling area with over 4,300, 000 computational nodes(the same modeling area of this manuscript, Figure1). To optimally use SWAN, sensitivity analysis on available whitecapping methods was absolutely necessary and the present manuscript shows a part of those extensive tests that we did to finalize the whitecapping parameters. By our experience, the results from this research are highly relevant for wave modeler all over the world simply due to the fact that SWAN is still being widely applied for wave modeling for different area. It could have saved many times for our simulation if such a research study was available during our modeling efforts. It should be noted that although more advanced model are consistent with observations and were not just developed based on solving the radiation transfer equation as an unknown,

yet their application in the real world simulation could be challenging and with lower accuracy compared to older models. One example is the study of Yan et al(2017) for the U.S West coast(the paper is attached to the response). Their one-year simulation showed that using ST6 physics package which is the newest in WWIII, does not actually improve wave parameters and resulted in lower model performance statistics compared to ST4 which is an older physics package. Regarding the WWIII physics packages (ST2, ST4, and ST6), we agree that we should have mentioned them in the manuscript. The related short summary about these packages will be added to the modified manuscript. The quality of any wave model hindcast is largely dependent on the quality of the wind fields. In this study no attempt has been made to validate the wind.

The authors have thoroughly verified the wind field versus wind measurements at the location of NDBC buoys (Allahdadi et al., 2018). Some verification results will be included in the modified manuscript.

++The notion that source terms should be recalibrated for different type of fetches or limitations is an interesting notion, but a more important conclusion is that the chosen formulations suffer a number of (already known) shortcomings, which can be remedied by introduction of more sound physics, viz. the parameterisations. In that sense, the present manuscript adds nothing new and recommends the wrong approach to improve

As mentioned above, the diagnosis about improving the two whitecapping approaches incorporated in SWAN; especially Wethuysen were regarding the fact that the more advanced whitecapping formulations like those are being used in WWIII, have not been incorporated in tne HPC-compatible version of SWAN yet, while SWAN is still being widely used all over the world. It means that still we need some reliable documentation of the shortcomings for each method based on the real-scale model simulations and the practical ways to overcome them. We will add notions in conclusion about the necessity for incorporating more advanced physics packages in SWAN to be more consistent with observations of whitecapping rates in the lab and field. Please also see
the response to + in the above.

+++The use of snapshots of spectra and then draw conclusions about model behavior is not sound science. This is statistically seen of no value.

The timstep corresponding the presented snapshots has not been selected arbitrarily, but selected based on a specific behavior of waves over the modeling area regarding the wind field. Please see 170-175 for details of the storm corresponding to time t1. The storm at this time resulted a fully-developed sea state at buoy 44011 that was the base for further analysis related to wave growth curves as presented in section 6.2. Based on our hypothesis on effect of sea state on simulation results, we examined several spectra at times with different sea states and observed similar behaviors for cases with fully-developed sea states at different buoys. The presented snapshot is a representative for these investigated case. We will make it more clear in the modified manuscript. Using just one snapshot as a representative of a specific behavior has also been used by other researches, ex. van der Westhuysen et al(2007), Mulligan et al.(200), and Vldder et al. (2016)

The mean wave period is not defined. It is not clear whether this is Tm01, Tm02 or even Tm-10 (also referred to as TE). Same, for the definition of mean wave direction.

The mean wave period used in this manuscript is the model-equivalent of zero-crossing wave period(T02). We will present the spectral definition on both mean wave period and mean wave direction used in this paper in the modified manuscript.

The authors suggest in the introduction (#26) that certain slanting fetch effects can not be generated due to short comings in the wind field. This notion is wrong, as it is a complex interplay of all source terms, see Ardhuin et al. (2007).

We agree that the effect of the "slanting fetch" is an interplay between the source terms since the source terms' quantifications are related to each other (especially whitecapping and wind input). We inferred expression of #26 from page 926 of the Ardhuin et al

(2007): "the wind input term of Janssen (1991) is found to be too narrow in direction, at least at the peak frequency, while the forms proposed by Snyder et al. (1981) and Tolman and Chalikov (1996) have a more appropriate directional shape in that range"

, but we will modify this in the manuscript based on the reviewer's comment that makes more sense based on the whole idea of Ardhuin et al(2007).

**41 Quantifying the source terms still is a challenging task, just look at the developments around more modern source terms like ST4 and ST6. Having said that, I feel the choice of the present authors is outdated.**

Please see the above responses on + and ***

**61 What are small coastal areas? We meant a single coastal area with limited fetch length, like the case studies by va der Westhhuysen et al(2007) at Virginia-North Carolina coast with maximum studied fetch lengths in the order of 100 km and smaller. It will be mentioned in the manuscript**

**64, #67 Reference missing in list of references, or is there a typo? Velder et al. (2016) Is a typo, the right one is Vledder et al(2016) that will be corrected in the manuscript.**

**78 which specific formulations are meant. This is the whitecapping source term $S\_ds^{JHHK}$ that was tuned for the WAM Cycle-4 model. This will be mentioned in the modified manuscript #87 The authors discuss seasonality as being important, but their analysis is based on only one month of data. Again, the authors miss an opportunity to study their problem in depth.**

As mentioned in this part of the paper, January is the representative of winter season. It was shown in Allahdadi et al(2018) that wind patterns and therefore wave and energy patterns from November to March are similar. During this time, the dominant wind field is generally winter storms that makes an offshoreward wind pattern over the model. However, we will try to be more specific just about January and less focused on the seasonality in the modified manuscript.

**95 The description of the role of the delta parameter is wrong. Yes, it has an effect on periods, but this parameter is not the related to another basic effect**

We implied this sentence about delta based in the SWAN 41.1 technical manual pages 20 and 21. We should have mentioned that it may also change the wave height and we need to change it in agreement with C_ds . We will make it more clear in the modified manuscript.

**145 Looking up Allahdadi et al. (2018) gave only an abstract. This makes it impossible to judge its content and how it relates to the present manuscript.**

The presentation is also available online through the Rsesrchgate. This file will be attached to the response to comment.

**154 A wind rose would reveal in proper detail what is dominant. We will include wind roses of those stations for January2009 in the modified manuscript.**

**156 Local wind and distant swell are unrelated. Therefore, this reasoning is flawed, unless the others refer to different phenomena. Authors could also have used spectral partitioning to select events of interest. Now, it is possible that their data are full of 'swell' noise.**

Our rational behind the argument of using this January 2009 for an almost swell-free analysis of whitecapping was based on our former long-term simulation for the East Coast and examining observed frequency-spectra. In that sense, during the simulation period, the dominant spectra at all four stations were single-peaked(please see for example Figure 8 of the original submission) that along with the dominant offshoreward wind direction shows just a wind-generated peak and no swell is present. We will include more descriptions in the modified manuscript to make it more clear. Also we can include more evidences of the single peaked spectra at different times. #182 Also convergence criterion important, like number of iterations per time step Will be mentioned in the modified manuscript

**201 The added value of the referenced papers is unclear. You do not need such references for basic statistics. The reference will be removed. #209 'Could be' An example of speculation We remove 'could be' from here and just use 'is' since we proved later that it was due to the slanting fetch effect.**

**216 Snapshots are insufficient to draw conclusions**

These snapshots in figure 7 are corresponding to a specific behavior of the wave field(fully-developed) that is representative of a group of incidents and are consistent with the wind field presented in Figure 3 of the original submission.

**232 The E(fp) is not a suitable parameter to draw conclusions, moreover as only one event is shown. It has only one degree of freedom. More robust methods are using bulk statistics of different period measures like Tm01 and Tm-10, which have different emphasis on weighting frequency bands**

We used E(fp), fp, and bulk ware parameters(Figures 4-7) together to conclude about the model behavior. E(fp) was only one of the parameters that we considered for making conclusion about the model behavior. It has also been used by van der Westhuysen et al (2007) to compare two whitecapping models. About the one timestep it was clarified before ( that it is a representative of a group with similar behavior.

**238 'could be ' another example of speculation. The expression will be removed by using some references to make it definite.**

**284 quadruplet terms cannot be predicted by Komen or Westhuysen source terms.**

This will be modified. We meant that the simulation using Komen and Westhuysen whitecapping resulted closer values of quadruplets when compared to differences between other two simulated source terms by two models.

**305 it is unclear how the actual fetch lengths are estimated in the open ocean.**

Calculation of actual fetch lengths that resulted the fetch limited and fully-developed

wave growth curves are described in Kahma and Calkoen (1992) for fetch-limited growth and Pierson-Moskowitz (1964) for fully-developed cases and is not in the scope of the present study. In our analysis we calculate the fetch length from land to the location of each buoy in the direction of the dominant wind that lasted for hours before times t1 and t2.

**311 what are the defaults. As they may change, it is better to specify them for instance in an appendix**

Default values for each whitecapping approach will be added to the modified paper.

**346 'Presumably', again speculation**

Since the effect of coastline on wave spectrum has later been discussed in section 6.3, we can remove presumably and make the argument definite.

**355 'may cause', speculation**

This speculation was made to show that future research studies need to treat fully-developed and fetch-limited conditions independently.

**362 why are small variations in speed and direction relevant in this context If time variations of wind speed and direction are too much, the conditions for reaching a fetch-limited or fully-developed sea state are violated(Coastal Engineering Manual, 2000). More details along with the reference will be mentioned in the modified manuscript. #368 I doubt whether revisiting the calibration process is the proper way. This can be a short-term fix, but rather difficult to obtain sound results. This notion is more an indication that the present physics in Westhuysen and Komen is incomplete. Sticking to outdated source terms does not seem a viable option Please see the answer to + and ++ above. #380 Concerning land-sea effects on the development of wind speed, the authors can refer to papers from e.g. Dobson et al. (1989, Atm_Ocean, 27) and Taylor and Lee (1984).**

The references and related descriptions will be added and discussed in the modified

manuscript.

**402 it is not how the 2D-spectra were reconstructed from the measurements in Figure 12 Measured and modeled spectra are different since model partly or completely fails to simulate the high frequency part of the spectrum generated as a result of the slanting fetch effect.**

**420 The discussion on temperature effects is interesting, but nothing is done with this. Moreover, this is effect is known already.**

For the first part of the comment please see the answer to ** Although this effect is already known, it has not been examined for the study area and here it was used as a complementary part along with other discussion points

**452 Do these so-called intense wind speeds occur in the present hindcast, and does the Yan parameterization really have an effect at the buoy locations**

Yes, we have large wind speeds over the modeling area. Please see #173 and Figure 2. The effect of Yan parametrization at buoy location is obvious looking at Figure 9d. This figure shows wind input term for the westhuysen formulation that uses Yan parametrization. Comparing it with Figure 9a for parametrization using Komen shows how the effect of high speed wind events are considered in Yan(1987) formulation. #455 I do not understand the reasoning about the computation of the quadruplets.

It is a standard way of examining source terms in simulations as used by van der Westhuysen et al(2007), van Veldder et al(2016), and several other studies. Here, it demonstrates that, although both simulation cases use the same computational approaches for calculating the quadruplet, still the amount of energy that is redistributed due to four-wave interaction is different between to methods due to different whitecappind approaches used for each simulation.

**476 What makes a coastline complicated? It all has to do with scales. The interpretation of the causes of the slanting fetch effect is incomplete. Directionally dependent**

[Figure]

fetch-lengths and the quadruplet interactions also play a role.

By "complicated coastal Geometry", we meant "variations in the coastline direction". We will modify it in the manuscript. "Directionally dependent fetch-lengths" that the reviewer mentioned is actually another manifestation of "deviation of the wind direction from the shore-normal direction" that we already mentioned in the text. We will directly mention this in the manuscript along with some discussions about the effect t of the quadruplet.

**479 See Ardhuin et al. (207) for an in-depth discussion More details will be added to the manuscript based on Ardhuin et al(2007).**

**480 Reasoning is incorrect. In case low-frequencies are underestimated, this may be due to the effect of the whitecapping term. Now the authors suggest it leads to an overestimation.**

We will modify the :underestimation of wave parameter" to "deviating simulated wave parameters from observations".

**488 What do the authors imply mean 'a spurious effect' We meant the lower or higher of whitecapping dissipation than the real value that is produced in the presence of swell waves (please see van der Westhuysen et al(2007))**

**490 Good idea to study all seasons. As in this manuscript only one month is used, the conclusions are not backed with sufficient data.**

We will limit our conclusion to only January 2009.

Table 2 Just providing data for one time instant is statistically insufficient and completely meaningless. Furtherm the mean wave period is not defined

Please see the answer to +++ Wave period is T02 that will be mentioned in the modified manuscript.

Figure 2 A wind and wave rose would be more informative. Wind roses will be included

in the modified manuscript

Figure 3 The snapshots are too similar to be of value. A possible link with the determination of a fetch length could add some value to this manuscript. Similarity of the wind field for the times before t1 and t2 as shown in Figure 3 is the main point of these snapshots to demonstrate that wind field was almost persistent so that can produce fetch-limited or fully-developed sea states. We will clarify this and the way we calculated the fetch length based on these wind fields. Figure 5 The causes of the discrepancies can also lie in incorrect winds. But nothing is said about the quality of the winds. Some details, add grid lines and close the rectangle.

Wind field evaluation will be added to the modified manuscript. Figures will be modified.

Figure 6 Same as for Figure 5

Figure 7 The arrows with the mean direction are too small. Further, what is the definition of mean wave direction? Figures will be re-plotted for larger vector sizes. The definition of the mean wave period will be added.

Figure 9 Nice pictures, but of little value as the underlying spectra are also different. Then the source terms will also be different and no firm conclusions can be drawn.

Presenting the spatial variations of main source terms simulated under two different whitecapping approaches is a significant part for demonstrating that different source term representation by these whitecapping approaches and their wind input counterparts are different. They show that although the wind input counterpart of the Westhuysen approach (Yan(1987)) resulted in larger input to the sea surface than that of Komen, larger whitecappin dissipation balances the excess input. But in the case of our study period this dissipation from Westhuysen approach is more than reality so that it may cause underestimation of the spectral energy in the wave spectrum. This implication will be added to the manuscript in section 6.1.

Figure 10 Why are the Snl4-terms oscillatory?

The oscillatory variations of the Snl4-term with frequency, especially the one for West-huysen simulation could be due to oscillation of the Whitecapping term between frequencies of 0.1 to 0.35 Hz (please see Figure 10b). This pattern has also been simulated by Mulligan et al(2008).

Figure 13 This figures confirms known concepts, but nothing is done in this study. Please see the answer to #420

Bets Regards Allahdadi et al.

Please also note the supplement to this comment:
https://www.ocean-sci-discuss.net/os-2018-112/os-2018-112-AC3-supplement.pdf
* * *
[Figure]

**Supplement:**

See discussions, stats, and author profiles for this publication at: https://www.researchgate.net/publication/324088629

**Allahdadi et al OSM2018**

**Presentation** · February 2018

DOI: 10.13140/RG.2.2.22754.43207

| CITATIONS | READS |
|---|---|
| 0 | 96 |

**5 authors**, including:

[Figure]

Nabi Allahdadi
North Carolina State University
**33** PUBLICATIONS   **48** CITATIONS

SEE PROFILE

Budi Gunawan
Sandia National Laboratories
**46** PUBLICATIONS   **252** CITATIONS

SEE PROFILE

Some of the authors of this publication are also working on these related projects:

Wave spectral modeliing View project

Wave Prediction Using a Mathematical model and comparing results with empirical prediction methods View project

All content following this page was uploaded by Nabi Allahdadi on 29 March 2018.

The user has requested enhancement of the downloaded file.

[Figure]

[Figure]

**High Resolution Wave modeling for Characterizing Wave Energy Resources along the U.S East Coast**

Nabi Allahdadi[1], Budi Gunawan[2], Jonathan Lai[2], Ruoying He[1], Vincent S. Neary[2]

1. North Carolina State University

2. Sandia National Laboratories

[Figure]

[Figure]

**Motivation**

- The global need to diversify energy portfolios, expand energy supplies and reduce carbon emissions

- Renewable energy supplied by ocean waves and swells can be converted to electricity using an assortment of technologies known as wave energy converters (WEC)

- The potential wave energy resource for the United States is about ten-percent of the global resource

- Resource characterization and assessment is an important first step to develop this wave energy resource potential and it supports a broad range of regional planning, project development and WEC design activities

[Figure]

Annual average wave power density along the East, Gulf, and West Coasts (EPRI 2011)

**Objectives**

◈ Preparing a long-term modeling database for waves along the East Coast based on a high spatial resolution computational mesh that is sufficient for high accuracy wave resource characterization

◈ Thorough calibration and verification of the model based on the 6 IEC parameters calculated at the location of coastal and offshore buoys and consistent with the test bed study along the Oregon Coast

◈ Presenting desired quantities including 6 IEC parameters and partitioned spectra at different locations and along the shoreline

**Buoy data**

- 18 NDBC buoys
- Both coastal and offshore
- All measured wave parameters including $H_s$, $T_p$, and $T_{02}$
- Only 7 of them measure directional spectra

| Buoy | Depth(m) | Description | Longitude | Latitude |
|---|---|---|---|---|
| 41002 | 3980 | SOUTH HATTERAS - 225 NM South of Cape Hatteras | -74.840 | 31.76 |
| 41004 | 38.4 | EDISTO - 41 NM Southeast of Charleston, SC | -79.099 | 32.501 |
| 41008 | 18.288 | GRAYS REEF - 40 NM Southeast of Savannah, GA | -80.868 | 31.4 |
| 41010 | 888 | CANAVERAL EAST - 120NM East of Cape Canaveral | -78.45 | 28.884 |
| 41013 | 23.5 | Frying Pan Shoals, NC | -77.743 | 33.436 |
| 41025 | 68.3 | Diamond Shoals, NC | -75.402 | 35.006 |
| 41047 | 5283 | NE BAHAMAS - 350 NM ENE of Nassau, Bahamas | -71.479 | 27.485 |
| 41048 | 5340 | WEST BERMUDA - 240 NM West of Bermuda | -69.590 | 31.86 |
| 44005 | 180.7 | GULF OF MAINE - 78 NM East of Portsmouth, NH | -69.128 | 43.201 |
| 44007 | 26.5 | PORTLAND 12 NM Southeast of Portland, ME | -70.141 | 43.525 |
| 44008 | 74.7 | NANTUCKET 54NM Southeast of Nantucket | -69.248 | 40.504 |
| 44009 | 43 | DELAWARE BAY 26 NM Southeast of Cape May, NJ | -74.703 | 38.461 |
| 44011 | 82.9 | GEORGES BANK 170 NM East of Hyannis, MA | -66.619 | 41.098 |
| 44013 | 64.5 | BOSTON 16 NM East of Boston, MA | -70.651 | 42.346 |
| 44017 | 52.4 | MONTAUK POINT - 23 NM SSW of Montauk Point, NY | -72.048 | 40.694 |
| 44018 | 217.3 | CAPE COD - 24 NM East of Provincetown, MA | -69.7 | 42.119 |
| 44025 | 40.8 | LONG ISLAND - 30 NM South of Islip, NY | -73.164 | 40.251 |
| 44027 | 178.6 | Jonesport, ME - 20 NM SE of Jonesport, ME | -67.307 | 44.287 |

**Computational mesh**

- Unstructured SWAN
- coastal resolution (within 20 km distance from the shoreline) : 200 meters
- Resolution along the offshore boundaries 10-15 km
- over 4,300,000 grid points

Map of spatial resolution

[Figure]

Mesh Module Grid spacing
- 2900.0
- 2600.0
- 2300.0
- 2000.0
- 1700.0
- 1400.0
- 1100.0
- 800.0
- 500.0
- 200.0

**Wind Field**

◈ Climate Forecast System Reanalysis (CFSR)

◈ Spatial resolution: 0.312 °

◈ Temporal resolution: 1 hour

◈ Seasonality

[Figure]

**Boundary Conditions**

◈ Global WWIII model implemented by NOAA (http://polar.ncep.noaa.gov/waves/hindcasts/nopp-phase2.php)

◈ Spatial resolution: 0.5 °

◈ Temporal resolution: 3 hours

◈ Physics package : ST4

◈ Wave parameters including height, peak period, and direction were prescribed along open boundaries

[Figure]

[Figure]

**Model run**

- 4,396,138  grid points
- 496 cores on the Sandia SkyBridge cluster~ 8700 node  for each core
- Each month of simulation took about 6 hours

[Figure]

**Model Sensitivity analysis**

- Several model parameters are site or scale specific
- Regional wind and wave climate may be important

| Parameter | Symbol | Examined values/Method | Selected value/Method |
|---|---|---|---|
| Computational time step | $\Delta t$ | 3, 5, 10, 20, 30, 50 minutes | 10 min |
| Number of iterations | Nit | 1, 3, 5, 7, 10 | 1 |
| Directional standard deviation | DSD | 20, 30, 50, 70 | 30 degrees |
| Frequency spectral shape | | JONSWAP: $\gamma = 1.1, 3.3, 6, 7$ Pierson-Moskowitz | 3.3 |
| Number of spectral frequencies | Nf | 18, 24, 28, 32 | 28 |
| Number of spectral directions | Nd | 18, 25, 36, 48 | 25 |

**Model Calibration**

Whitecapping dissipation and the associated parameters were used

Komen-type approaches (mean spectral parameters)
*Komen(1984) and Janssen(1991)*

$$S_{ds,w}(\sigma,\theta) = -\Gamma\tilde{\sigma}\frac{k}{\tilde{k}}E(\sigma,\theta)$$

$$\Gamma = C_{ds}\left((1-\delta)+\delta\frac{k}{\tilde{k}}\right)\left(\frac{\tilde{S}}{\tilde{S}_{PM}}\right)^p$$

$$\tilde{S}_{PM} = (3.02\times10^{-3})^{1/2}$$

$$\tilde{S} = \tilde{k}\sqrt{E_{tot}}$$

Saturated-based approach
*Van der Westhuysen(2007)*

$$S_{ds,break}(\sigma,\theta) = -C_{ds}\left[\frac{B(k)}{B_r}\right]^{p/2}\left[\tanh(kd)\right]^{\frac{2-p_0}{4}}g^{\frac{1}{2}}k^{\frac{1}{2}}E(\sigma,\theta)$$

$$S_{ds,w}(\sigma,\theta) = f_{br}(\sigma)S_{ds,break} + \lceil 1 - f_{br}(\sigma)\rceil S_{ds,non-break}$$

$$S_{ds,non-break}$$

accounts for dissipation by turbulence and short-wave-long-wave interaction

Three 1-month time periods were used for model calibration / measured waves at 18 NDBC buoys
different whitecapping formulations and different sets of parameters were examined

*Janssen(1991)* $C_{ds}$=2.7 $\delta$=0.9

**Model verification**

**Model performance metrics**

$$R = \frac{\sum_{i=1}^{N}(M_i - \bar{M})(P_i - \bar{P})}{\sqrt{(\sum_{i=1}^{N}(M_i - \bar{M})^2)(\sum_{i=1}^{N}(P_i - \bar{P})^2)}}$$

$$bias(P) = \frac{\sum_{i=1}^{N}(P_i - M_i)}{N}$$

$$RMSE = \sqrt{\frac{\sum_{i=1}^{N}(P_i - M_i)^2}{N}}$$

$$SI = \frac{RMSE}{\bar{M}}$$

M: measurement    P: model prediction

**IEC parameters**

Significant wave height          $H_s$

Energy period          $T_e$

Omnidirectiona wave power          $J = \rho g \sum_{i,j} c_{g,i} \, S_{ij} \Delta f_i \Delta \theta_j$          $[\frac{kW}{m}]$

Spectral width          $\epsilon_0 = \sqrt{\frac{m_0 m_{-2}}{m_{-1}^2} - 1}$

Direction of max power          $\theta_{Jmax}$

Directionally  coefficient          $d = \frac{J_{\theta_{Jmax}}}{J}$

i,j : number of spectral frequencies and directions, respectively

$m$ : spectral moments

*3-year verification from 2007 to 2009*

[Figure]

[Figure]

**Model performance metrics at selected stations**

| Buoy | Parameter | RMSE | SI | Bias | R |
|---|---|---|---|---|---|
| 41013 | | | | | |
| | Hs(m) | 0.31 | 0.24 | 0.13 | 0.91 |
| | Te(s) | 0.92 | 0.15 | 0.45 | 0.79 |
| | $\epsilon_0$(-) | 6.44 | 0.82 | 2.12 | 0.90 |
| | J(kw/m) | 0.07 | 0.20 | -0.01 | 0.59 |
| | $\theta_{Jmax}$(degrees) | 39.50 | N/A | 2.77 | 0.68 |
| | $d_\theta$ (-) | 0.14 | 0.19 | 0.07 | 0.37 |
| 41048 | Hs(m) | 0.38 | 0.20 | 0.14 | 0.93 |
| | Te(s) | 0.78 | 0.11 | 0.35 | 0.87 |
| | $\epsilon_0$(-) | 14.94 | 0.85 | 3.53 | 0.88 |
| | J(kw/m) | 0.05 | 0.17 | -0.02 | 0.65 |
| | $\theta_{Jmax}$(degrees) | 39.63 | N/A | 0.92 | 0.70 |
| | $d_\theta$ (-) | 0.09 | 0.13 | 0.02 | 0.77 |
| 44018 | Hs(m) | 0.46 | 0.31 | 0.01 | 0.86 |
| | Te(s) | 1.13 | 0.17 | -0.32 | 0.68 |
| | $\epsilon_0$(-) | 11.07 | 0.98 | -1.10 | 0.80 |
| | J(kw/m) | 0.09 | 0.27 | 0.03 | 0.45 |
| | $\theta_{Jmax}$(degrees) | 47.57 | N/A | -3.20 | 0.55 |
| | $d_\theta$ (-) | 0.17 | 0.25 | 0.10 | 0.27 |
| 44025 | Hs(m) | 0.40 | 0.30 | 0.15 | 0.87 |
| | Te(s) | 1.12 | 0.17 | 0.42 | 0.73 |
| | $\epsilon_0$(-) | 10.11 | 1.25 | 2.15 | 0.77 |
| | J(kw/m) | 0.10 | 0.28 | -0.03 | 0.44 |
| | $\theta_{Jmax}$(degrees) | 43.81 | N/A | -0.29 | 0.58 |
| | $d_\theta$ (-) | 0.19 | 0.26 | 0.11 | 0.11 |

**Summary of metrics for all 18 stations**

| Parameter | Type of statistics | RMSE | SI | Bias | R |
|---|---|---|---|---|---|
| Hs(m) | | | | | |
| | Mean | 0.39 | 0.28 | 0.11 | 0.88 |
| | Max | 0.51 | 0.44 | 0.24 | 0.95 |
| | Min | 0.29 | 0.19 | 0.01 | 0.78 |
| Te(s) | | | | | |
| | Mean | 1.15 | 0.18 | 0.58 | 0.75 |
| | Max | 1.61 | 0.26 | 0.98 | 0.88 |
| | Min | 0.75 | 0.10 | 0.30 | 0.59 |
| J(kw/m) | | | | | |
| | Mean | 10.51 | 1.05 | 2.06 | 0.83 |
| | Max | 17.71 | 1.75 | 4.54 | 0.93 |
| | Min | 4.70 | 0.68 | 0.04 | 0.73 |
| $\epsilon_0$(-) | | | | | |
| | Mean | 0.08 | 0.25 | 0.02 | 0.47 |
| | Max | 0.13 | 0.36 | 0.03 | 0.65 |
| | Min | 0.05 | 0.17 | 0.00 | 0.22 |
| $\theta_{Jmax}$(degrees) | | | | | |
| | Mean | 42.36 | N/A | 4.37 | 0.59 |
| | Max | 47.57 | N/A | 10.55 | 0.70 |
| | Min | 39.50 | N/A | 0.29 | 0.33 |
| $d_\theta$ (-) | | | | | |
| | Mean | 0.16 | 0.22 | 0.10 | 0.32 |
| | Max | 0.22 | 0.30 | 0.19 | 0.77 |
| | Min | 0.09 | 0.12 | 0.02 | 0.04 |

NDBC 41013- NC coastal

NDBC 44025- Off the NY Harbor

Observation Model Observation Model

Summer

Non-Summer

**Simulated wave heights over the modeling area**

[Figure]

Significant Wave Height(m)  January 2009  1-10-2009:0:00:00  1-30-2009:0:00:00

Monthly average

Significant Wave Height(m)  July 2009  7-20-2009:0:00:00  7-30-2009:0:00:00

Monthly average

Hurricane Ophelia- September 2005
9-12-2005:0:00:00  9-16-2005:0:00:00  9-17-2005:0:00:00

**Monthly mean wave power for 2009**

[Figure]

**Summary and conclusions**

◈ An ultra high-resolution wave climate model with coastal resolution of 200 meters was setup and verified

◈ Unstructured SWAN using a mesh with triangular elements was used

◈ CFSR wind field(spatial resolution of 0.312 ° and temporal resolution of 1 hour) along with WWIII-ST4( spatial resolution of 0.5 ° and temporal resolution of 3 hours) as the boundary condition were used to force the model.

◈ Seasonality of wind with generally higher wind speeds in winter

◈ Model was calibrated and verified for whitecapping dissipation

◈ Six IEC parameters were used for model verification

◈ Based on model calibration and verification results at the location of 18 NDBC buoys, Janssen(1991) with $C_{ds}$=2.7 $\delta$=0.9 was selected for whitecapping dissipation. The 3-year verification results (2007-2009) were comparable with the test bed study along the West Coast

◈ Seasonality in wind caused seasonality in the simulated wave power with largest amounts of coastal wave power in winter and early spring and the lowest power in summer.

**Acknowledgments**

Sandia National Laboratories is a multi-mission laboratory managed and operated by National Technology and Engineering Solutions of Sandia LLC, a wholly owned subsidiary of Honeywell International Inc. for the U.S. Department of Energy's National Nuclear Security Administration under contract DE-NA0003525

Renewable Energy xxx (2017) 1—13

[Figure]

Contents lists available at ScienceDirect

**Renewable Energy**

journal homepage: www.elsevier.com/locate/renene

[Figure]

**A wave model test bed study for wave energy resource characterization**

Zhaoqing Yang [a, *], Vincent S. Neary [b], Taiping Wang [a], Budi Gunawan [b], Annie R. Dallman [b], Wei-Cheng Wu [a]

[a] *Pacific Northwest National Laboratory, 1100 Dexter Ave North, Suite 400, Seattle, WA 98109, USA*
[b] *Sandia National Laboratories, P.O. Box 5800, Albuquerque, NM 87185 − MS1124, USA*

**ARTICLE INFO**

*Article history:*
Received 30 August 2016
Received in revised form
20 December 2016
Accepted 22 December 2016
Available online xxx

*Keywords:*
Wave energy
Resource characterization
Test bed study
Numerical wave modeling
WaveWatch III
SWAN

**ABSTRACT**

This paper presents a test bed study conducted to evaluate best practices in wave modeling to characterize energy resources. The model test bed off the central Oregon Coast was selected because of the high wave energy and available measured data at the site. Two third-generation spectral wave models, SWAN and WWIII, were evaluated. A four-level nested-grid approach—from global to test bed scale—was employed. Model skills were assessed using a set of model performance metrics based on comparison of six simulated wave resource parameters and observations from a wave buoy inside the test bed. Both WWIII and SWAN performed well at the test bed site and exhibited similar modeling skills. The ST4 physics package with WWIII, which represents better physics for wave growth and dissipation, outperformed ST2 physics and improved wave power density and significant wave height predictions. However, ST4 physics tended to over-predict the wave energy period. The newly developed ST6 physics did not improve the overall model skill for predicting the six wave resource parameters. Sensitivity analysis using different wave frequencies and direction resolutions indicated the model results were not sensitive to spectral resolutions at the test bed site, likely due to the absence of complex bathymetric and geometric features.

© 2016 Published by Elsevier Ltd.

**1. Introduction**

The recently published International Electrotechnical Commission Technical Specification (IEC TS) provides a standardized methodology for consistent and accurate wave resource assessment and characterization [1]. The methodology relies primarily on spectral wave model hindcasts for deriving recommended wave energy resource parameters. It also includes best modeling practices that depend on the desired class of wave resource characterization and assessment, including model selection, period of simulation, open boundary conditions, grid resolution, forcing (spatial and temporal) resolution, and model validation.

Although buoy observations can provide realistic directional wave spectra data for accurate resource assessment at a particular site, they are often constrained by spatial and temporal distributions. Existing buoy stations may not be close enough to the study

site to be representative of the wave climate; or they may have an insufficient period of record to accurately characterize the wave climate statistics. Long-term measurement records are especially important for characterizing extreme sea states, as well as normal sea states when inter-period climate oscillations occur on the order of a few years or decades [2—6]. A minimum 10 years of record is often recommended for characterizing normal sea states, and 20 years for extreme sea states [1]. However, it is rare to find buoy observations that are representative of the wave climate at the study site and have periods of records greater than 10 years. Model hindcasts of the wave climate, therefore, offer an attractive alternative for characterizing wave energy resources [7—14].

Even if a wave model captures all of the key physics (e.g., wave generation, growth and dissipation, nonlinear interactions), accurate wave modeling still highly depends on model configurations such as source term selection and spectral resolutions, specification of forcing inputs, model grid resolutions, proper model calibration, and validation. When selecting models for wave resource characterization it is important to understand the key processes affecting wave dynamics near the shore where wave energy conversion

\* Corresponding author.
  *E-mail address:* zhaoqing.yang@pnnl.gov (Z. Yang).

http://dx.doi.org/10.1016/j.renene.2016.12.057

0960-1481/© 2016 Published by Elsevier Ltd.

devices are expected to be deployed. The most popular third-generation phase-averaged spectral models include the Wave Action Model (WAM) [15], Simulating WAve Nearshore (SWAN) [16], WAVEWATCH III® (WWIII) [17], TOMAWAC [18], and MIKE-21 Spectral Wave models (MIKE-21 SW) [19].

The overall goal of this study was to establish a wave model test bed to benchmark, test, and evaluate modeling methodologies and model skills for predicting the wave energy resource parameters recommended by IEC TS. The following sections review current wave modeling best practices, third-generation wave models, and evaluate model capability in predicting normal and extreme sea states, and recommend future research to improve wave modeling for resource characterization.

**2. Methods**

This section describes the model test bed site, the selection of wave models, and model setup, which includes data inquiries and processing, grid generation, specification of open-boundary conditions, and input configurations.

**2.1. Model domain — test bed**

The model test bed for wave resource characterization was selected primarily based on its meeting three criteria: 1) high wave energy resource site with potential for future wave energy converter development, 2) availability of long-term and high-quality wave measurement data, and 3) existing information from previous studies. The Oregon Coast is among the highest wave energy regions along the U.S. coasts, based on the U.S. nationwide wave resource assessment conducted by the Electric Power Research Institute [20]. Therefore, a wave modeling test bed was selected near the central Oregon Coast, approximately centered offshore from Newport, Oregon (Fig. 1). The test bed site covers an area of $44.45° - 45°$ N and $124.75° - 124°$ W (61,105 m × 59,401 m) and has annual average wave power densities that range between 35 and 50 kW/m [20]. The test bed site also includes Tier 1 wave energy converter test sites, such as the active North Energy Test Site (NETS) managed by the Pacific Marine Energy Center [10]. An operational real-time wave buoy (46050) owned and maintained by the National Oceanic and Atmospheric Administration's (NOAA's) National Data Buoy Center (NDBC) is located inside the test bed (Fig. 1). The NDBC Buoy 46050 is a 3-m discus meteorological ocean platform moored at a deep water depth of 137.2 m. The buoy station has been collecting standard meteorological data, including wind speed and direction, gust speed, air temperature, sea surface temperature since 1991, and high-quality wave spectral data since 2008.

There are some previous studies along the Oregon Coast with areas inside or overlapped with the test bed site. An initial effort was made to characterize the wave energy resource of the US Pacific Northwest by Lenee-Bluhm et al. [21] using archived spectral records from ten wave measurement buoys operated and maintained by NDBC and the Coastal Data Information Program (CDIP). García-Medina et al. [9,22] conducted a wave resource assessment along the Pacific Northwest coast using WWIII and SWAN models with a nested-grid approach. Model results from the 7-year hindcast with a 30 arc-second grid resolution were used to evaluate the temporal and spatial variability and trends of wave resource in the Pacific Northwest coast. Dallman and Neary [10] used historical data from buoy NDBC 46050 inside the test bed to present representative spectra and predict extreme sea states. Different from previous studies, the present study focuses on establishing a wave model test bed to evaluate approaches and wave models for simulating wave resource parameters recommended by IEC TS.

**2.2. Wave models**

A wide range of numerical models exist for simulating surface wave dynamics based on different physical assumptions and numerical frameworks. Wave models can be divided into two major categories based on different governing equations in time and frequency domains: 1) phase-resolving models and 2) phase-averaged models. Phase-resolving models are based on fundamental wave equations that involve rigorous approximations. Evolution of the sea state over time is simulated using a model grid resolution much smaller than the wavelength and fine model time step, which typically requires huge computational resources. In addition, some of the phase-resolving models, such as Boussinesq type models, are only applicable in the simulation of waves for shallow water. Therefore, phase-resolving models are impractical for hindcasts for long-term simulations (multiple years) and relatively large model domains (dimension >10 km). In contrast, phase-averaged models provide a statistical description of the wave conditions in spatial and temporal domains by solving the phase-averaged wave energy action balance equation, and they compute the distribution of wave energy in the frequency and direction domain and its evolution over time. Therefore, use of phase-averaged wave models is the most practical approach for characterizing wave resources.

Since the 1990s, third-generation wave models explicitly account for all the relevant physics for the development of ocean waves in two dimensions. WAM, WWIII SWAN, TOMWAC and MIKE-21 SM are the five most popular third-generation models that have been widely validated in many applications around the world. The present study focused on evaluation of structured-grid wave models. Among the aforementioned five third-generation wave models, TOMWAC and MIKE-21 SM are unstructured-grid models and will not be considered in the present study. WAM is very similar to WWIII and the main difference is the numerical schemes. Therefore only SWAN [16,23] and WWIII [24–26], the two most widely used third-generation, phase-averaged wave models, were evaluated in this study. Both SWAN and WWIII have been used to simulate wave climate and resource characterization around the world [9,10,13,14,20,27–32]. One of the fundamental differences between WWIII and SWAN is the numerical scheme used to solve the spectral wave action balance equation. WWIII uses explicit numerical schemes, so the model time steps are constrained by the Courant–Friedrichs–Lewy (CFL) stability criteria. SWAN uses implicit schemes, which allows much larger time steps for high computational efficiency.

WWIII was developed and is maintained by NOAA's National Centers for Environmental Prediction (NCEP) [17,25,33], as part of the marine operational forecast system. The current version of WWIII (version 4.18) consists of a collection of physics packages, including curvilinear grids, structured and unstructured-grids, effects of sea ice, and various wind-wave interaction and dissipation packages, such as the source term 2 (ST2), ST4, and ST6 physics package options [34–37]. The ST2 physics package was developed by Tolman and Chalikov [37] based on previously developed input and nonlinear interaction source terms and a new dissipation source term for low and high frequencies. The ST4 physics package consists of new parameterizations for spectral dissipation of wind-generated waves based on known properties of swell dissipation and wave breaking statistics that are consistent with observations [34]. The ST6 physics package, or the so-called BYDRZ (abbreviation for Babanin-Young-Donelan-Rogers-Zieger) source term, implements observation-based physics for wind input source term and sink terms due to negative wind input, whitecapping dissipation and wave-turbulence interactions [17,38].

In contrast to WWIII, SWAN solves the action balance equation

[Figure]

**Fig. 1.** Study domain and nested grids for a wave modeling test bed for resource characterization on the coast of central Oregon, USA.

using implicit numerical schemes, so it is more computationally efficient for simulating wave climate in high-resolution model grids [16]. SWAN models nearshore wave dynamics that include nonlinear wave interactions, refraction and shoaling due to bathymetry or ambient currents, and whitecapping and depth-induced breaking. In addition, SWAN can solve the steady form of the action balance equation by running in the stationary mode, which greatly reduces computational requirements and run times. This option is generally applicable when the model domain has a dimension of approximately less than 100 km [16]. As a result, SWAN is the most commonly used model for wave resource characterization in the nearshore regions [10,22,29–32,39].

**2.3. Model setup**

A nested-grid approach was employed for this study. Four different levels of rectangular grids were considered. The level 1 (L1) grid is the global grid with a spatial resolution of 0.5° in both the longitudinal and latitudinal directions. The global model domain (L1) covers most of the globe from latitude 77.5° S to 77.5° N. The level 2 (L2) grid, which is nested into the global grid, covers the West Coast continental shelf (120° W to 128° W) from Southern California at 35° N to Vancouver Island, British Columbia, at 50° N. The level 3 (L3) grid centers on the Oregon Coast and is nested inside the L2 grid from 43.6° to 45.9°N and 125.6° to 123.8°W. The test bed level 4 (L4) grid is nested inside the L3 grid from 44.45° to 45°N and 124.75° to 124°W. The nested-grid scaling ratio (the ratio of the coarse-grid resolution to the fine-grid resolution) is set to a value of approximately five to six to maintain a smooth transition between model results along the nested-grid boundaries. The L2 and L3 grid resolutions are 6 arc-minutes and 1 arc-minute, respectively. The test bed (L4) grid resolution is 12 arc-seconds in the longitudinal direction and 10 arc-seconds in the latitudinal direction, which approximately corresponds to 265 m by 308 m and follows recommendations of the IEC TS [1]. The L3 grid provides open-boundary conditions for the high-resolution test bed model. In this study, estuaries and coastal bays along the coast were not considered and therefore they are not represented in the model grids. The model domains for the L4 to L2 nested grids are shown in Fig. 1. The domain coordinates (in latitude and longitude), spatial resolution (grid dimensions), and grid size (number of grid points) for all four model grids are summarized in Table 1.

Model grid and bathymetry files for the global wave model were obtained directly from NOAA's Environmental Modeling Center, Marine Modeling and Analysis Branch. The model bathymetry for the nested L2 to L4 grid was interpolated from NOAA's 3 arc-second (~90 m) Coastal Relief Model for the inner-shelf region, and NOAA's 1 arc-minute ETOPO1 Global Relief Model for the outer-shelf region and deep ocean. The resolution of the Coastal Relief Model data set is sufficient for the inner-shelf region because the local model grid resolution is ~300 m. The eastern Pacific Ocean coast has a narrow continental shelf and a deep outer-shelf basin. The model bathymetry for the test bed domain (L4) was further interpolated from NOAA's high-resolution (1/3 arc-second) tsunami bathymetry data. The average water depth for the test bed model domain is 165 m, and the maximum water depth is approximately 600 m. The distribution of model bathymetry for the test bed L4 grid is shown in Fig. 2.

Surface wind forcing is one of the most important factors for simulating wave generation and propagation from the outer continental shelf to the inner shelf; therefore, it is critical for all three classes (reconnaissance, feasibility and design) of wave resource characterization and assessment [1]. In this study, wind speed and direction were obtained from the Climate Forecast System Reanalysis (CFSR), which is a coupled atmosphere-ocean-land surface-sea ice system developed by NOAA's NCEP [40,41]. The CFSR data cover the entire global domain at a 1-h temporal resolution and a 0.5-degree spatial resolution for a 32-year period from January 1979 to March 2011. Fig. 3 shows the monthly-averaged global CFSR wind distributions in July and November 2009. A distinct seasonal pattern observed in Fig. 3 is that wind in the northern hemisphere is stronger in November (winter) than in July (summer), while opposite conditions exist in the southern hemisphere because July and November correspond respectively to winter and summer there. Sea-ice data required for the global model were also obtained from CFSR data sets. Sea-ice data have the same spatial and temporal resolutions as the wind data.

A full-year simulation allows for an evaluation of the seasonal effects on wave resource parameters. Directional spectral data at NDBC Buoy 46050 are available starting on March 5, 2008. Wave data from NDBC Buoy 46050 show strong seasonal variations of significant wave height in 2009; a series of storms occurred in winter, and relatively calm seas occurred during summer. Calendar year 2009 was selected as the model simulation period based on the availability and completeness of wind forcing data and met-ocean data for model validation at NDBC Buoy 46050. WWIII was started from 12/1/2008 to allow sufficient spin-up time.

Although tidal currents can be strong along the West Coast, especially in estuaries and bays along the Pacific Northwest coast, wave-current interaction induced by tides or ocean currents was not the focus of the study reported here. The IEC TS [1] on Wave Resource Characterization recommends including ocean current data in wave models only if depth-averaged current speeds exceed 1.5 m/s. In general, this is unlikely for points of investigation during normal sea states because the model test bed is not close to any estuaries or bays with strong currents.

**3. Results and discussion**

**3.1. Baseline simulation**

A baseline simulation was first conducted using WWIII with the ST2 physics package and SWAN in stationary mode as well as non-stationary mode (SWAN-NS). The growth and dissipation in the ST2 physics are based on the physics by Tolman and Chalikov [37]. Both WWIII and SWAN used default settings only; i.e., neither model was calibrated by tuning model parameters. The baseline simulation used 29 frequency bins with a minimum frequency of 0.035 Hz and a logarithmic increment factor of 1.1, which gives the maximum frequency of 0.505 Hz. In the directional domain, 24 direction bins were specified with a resolution of 15°. The spectral resolution meets the minimum requirements specified by IEC TS [1]; i.e., a minimum of 25 frequency components and 24 to 48 directional components, and a frequency range covering at least 0.04−0.5 Hz.

To the extent possible, the source term models for SWAN were selected to agree with those in WWIII. Common source/sink terms include those for linear wave growth by Cavaleri and Rizzoli [42], exponential wave growth by Janssen [43,44], dissipation due to bottom friction (JONSWAP [Joint North Sea Wave Project]), depth-induced breaking by Battjes and Janssen [45], and nonlinear wave-wave of quadruplets by Hasselmann et al. [46]. The formulations for the whitecapping and turbulent boundary layer dissipation are different in SWAN and WWIII (ST2). SWAN models dissipation due to whitecapping using the formulation by Komen et al. (1984), while WWIII (ST2) uses the turbulent boundary layer

**Table 1**
Summary of nested WWIII model grids for the wave model test bed.

| Grid name | Coverage | Resolution (long., lat.) | Cell number |
|---|---|---|---|
| Global Grid L1 | 77°S–77°N | 0.5° × 0.5° (30′ × 30′) | 223,920 |
| Nested Grid L2 | 35°–50°N; 128°–120°W | 0.1° × 0.1° (6′ × 6′) | 12,231 |
| Nested Grid L3 | 43.6°–45.9°N; 125.6°–123.8°W | 1′ × 1′ | 15,151 |
| Test Bed Grid L4 | 44.45°–45°N; 124.75°–124°W | 12″ × 10″ (265 m × 308 m) | 44,974 |

[Figure]

**Fig. 2.** Bathymetry of the test bed off central Oregon Coast. Red circle indicates the buoy location of NDBC 46050. NETS = North Energy Test Site. (For interpretation of the references to colour in this figure legend, the reader is referred to the web version of this article.)

[Figure]

**Fig. 3.** Global wind distribution from the Climate Forecast System Reanalysis system at 0.5° resolution for July 2, 2009 (a) and November 7, 2009 (b).

dissipation model described by Tolman and Chalikov [37]. As noted by Tolman et al. [17], "The wind-wave growth and dissipation are separate, but interrelated processes, because the balance of these two source terms governs the integral growth characteristics of the wave model." Several combinations of these basic source terms are

available in the different physics packages of WWIII, as described by Tolman et al. [17].

Both WWIII and SWAN employ source terms needed for the 'design class resource assessment' (highest resolution) in the IEC TS [1], including terms for linear and exponential wind growth, and dissipation terms that simulate whitecapping, quadruplet wave interaction, wave breaking, and bottom friction. Default parameter settings for these source term models were used in all of the simulations presented herein.

Due to different numerical schemes used in WWIII and SWAN, the model run time steps are also very different. WWIII uses a time-splitting approach with four different time steps, including the global time step $\Delta t_g$, the spatial propagation time step $\Delta t_{xy}$, the intra-spectral propagation time step $\Delta t_k$, and the source term time step $\Delta t_s$ [17]. The important time step that controls model stability is the CFL time step $\Delta t_{xy}$ for spatial propagation for the specific model grid resolution. The computational time step in SWAN is not restricted by CFL stability criteria, because of an implicit scheme used in the model, and one global time step is used for the model run. The time steps used in WWIII and SWAN simulations for all of the grids are shown in Table 2. Clearly, the model run time for WWIII is constrained by the small spatial propagation time step $\Delta t_{xy}$ of 8 s for the test bed (L4 grid). In this study, a time step of 60 s

**Table 2**
Model run time steps for WWIII (L1−L4 grids) and SWAN (L4 grid).

| WWIII Nested Grid | $\Delta t_g$ (s) | $\Delta t_{xy}$ (s) | $\Delta t_k$ (s) | $\Delta t_s$ (s) |
|---|---|---|---|---|
| Global Grid L1 | 3600 | 480 | 1800 | 30 |
| Nested Grid L2 | 600 | 240 | 300 | 15 |
| Nested Grid L3 | 100 | 45 | 50 | 15 |
| Test Bed Grid L4 | 20 | 8 | 10 | 15 |
| SWAN Grid | | $\Delta t_{xy}$ (s) | | |
| Test Bed Grid L4 | | 60 | | |

was used for the SWAN simulations for the test bed domain.

A year-long simulation with the nested-grid WWIII model was conducted to generate wave resource parameters for the test bed. Wave spectral data along the open boundary of the L4 grid were also outputted at hourly intervals to drive the SWAN simulation for the test bed. Fig. 4 shows the global distributions of monthly-averaged significant wave height in July and November 2009. High significant wave heights are shown in the Southern Ocean during July (Fig. 4a) and in the Eastern Pacific Ocean and Atlantic Ocean (Fig. 4b) during November, which correspond to the seasonal distribution patterns of wind field (Fig. 3). The monthly-averaged significant wave heights in the West Coast region show a smooth transition of the wave field across the nested-grid boundaries from global scale to test bed local scale (Fig. 5a and b), thereby demonstrating the successful implementation of the nested-grid approach. Significant wave heights in July in the Pacific Northwest coastal region are generally much smaller than those observed in November.

The IEC TS [1] recommends six parameters for characterizing the wave energy resource, which were used in recent studies analyzing the wave resource off the U.S. Pacific Northwest coast [9,10,21]. These six parameters are omnidirectional wave power $J_{omni}$, significant wave height $H_{m0}$, energy period $T_e$, spectral width $\in_0$, direction of maximum directionally resolved wave power $\theta$, and the directionality coefficient $d$. The formulations of these six wave resource parameters are given in IEC TS [1], and repeated in Ref. [10]. The six IEC TS wave resource parameters were calculated based on model outputs from the baseline simulations using the WWIII ST2 physics package and SWAN. Here, only model results from WWIII are presented and compared to those calculated from buoy measurements, because the results from SWAN are similar to those from WWIII (Fig. 6). Overall, model results match the data for all six parameters quite well. Omnidirectional power, significant wave height, and energy period show strong seasonal variations. Because winds are strong in the winter and weak in the summer, omnidirectional wave power and significant wave height are large in the winter and small in the summer. The wave energy period tends to be longer as a result of the large swell caused by high wind

in the winter. In contrast, the wave energy period becomes shorter in the summer because of calm sea states and low winds.

Scatter plots of the six wave resource parameters are presented in Fig. 7. In general, model results for wave power and significant wave height become more scattered and less accurate under large wave conditions (Fig. 7a and b). Simulated wave energy periods do not show strong heteroscedasticity, as do wave power $J_{omini}$ and significant wave height $H_s$; rather they tend to be slightly over-predicted in comparison to observed data (Fig. 7c). Wave energy period changes from approximately 5 s during summer to as high as 15 s during winter. The spectral width generally varies between 0.2 and 0.6, with low spreading of wave energy during winter and high spreading in the summer (Fig. 7d). The wave direction of maximum directionally resolved wave power is typically in the range of 215°−315°, indicating that waves propagate dominantly from the west direction (Fig. 7e). The directional coefficient is very scattered and tends to overpredict most of the time during the year (Fig. 7f). The larger values of the directional coefficient in the winter indicate low directional spreading.

**3.2. Model skills**

To evaluate the performance of the model in predicting the six wave resource parameters, four statistics were computed to quantify the discrepancies between the simulated results and observed data: the root-mean-square-error, model bias, scatter index, and linear correlation coefficient. The root-mean-square-error (RMSE), is defined as

$$RMSE = \sqrt{\frac{\sum_{i=1}^{N}(P_i - M_i)^2}{N}}$$

where $N$ is the number of observations, $M_i$ is the measured value, and $P_i$ is the predicted value.

RMSE represents the sample standard deviation of the differences between predicted values and measured values. Model bias, which represents the average difference between the predicted and measured value, is defined as

$$Bias = \frac{1}{N}\sum_{i=1}^{N}(P_i - M_i),$$

The scatter index (SI) is the RMSE normalized by the average of all measured values over the value of comparison:

$$SI = \frac{RMSE}{\overline{M}},$$

where the overbar indicates the mean of the measured values. The linear correlation coefficient (R) is a measure of the strength of the linear relationship between the predicted and measured values, and is defined as

$$R = \frac{\sum_{i=1}^{N}(M_i - \overline{M})(P_i - \overline{P})}{\sqrt{\left(\sum_{i=1}^{N}(M_i - \overline{M})^2\right)\left(\sum_{i=1}^{N}(P_i - \overline{P})^2\right)}}$$

The model performance metrics for each of the six IEC TS parameters are shown in Table 3 for the baseline WWIII simulation with the ST2 physics package and for the SWAN and SWAN-NS simulations with WWIII outputs as open-boundary conditions. Note that no scatter index (SI) values were provided for direction of maximum directionally resolved wave power $\theta$ in Table 3 because the discontinuity from 360 to 0° for the mean angle in the SI equation may give misleading results. The values of these model

[Figure]

**Fig. 4.** Global distributions of monthly-averaged significant wave height simulated by WWIII for July 2009 (a) and November 2009 (b).

[Figure]

**Fig. 5.** Distributions of monthly-averaged significant wave height simulated by WWIII in the US West Coast region for July 2009 (a) and November 2009 (b).

[Figure]

**Fig. 6.** Comparisons of the six IEC observed and modeled parameters at Buoy 46050 for the WWIII baseline condition.

performance metrics for all three model runs are very similar, indicating the model skills for WWIII, SWAN, and SWAN-NS are about the same at the test bed site. The error statistics for the baseline model runs are similar to those in other previous studies conducted in the region [9,22], indicating that all three models performed well and the model results are in good agreement with observations at NDBC Buoy 46050. In particular, the RMSEs for $J_{omni}$, $H_s$, and $T_e$ are about 20.0 (kW/m), 0.42−0.45 m, and below 1 s, respectively. The linear correlation coefficients for $J_{omni}$, $H_s$, and $T_e$ are all above 0.9. However, it is noticeable that the correlation co-efficients of modeled and measured spectral width $\in_0$, direction of maximum directionally resolved wave power $\theta$, and the direc-tionality coefficient $d$ are relatively low. The low correlation of $\in_0$ is because it is a function of higher order moment of variance spec-trum. Therefore it is much more difficult to simulate spectral width $\in_0$. The low correlations of modeled and measured $\theta$ and $d$ are generally due to the high uncertainty in both modeled and measured wave direction [9,12,21]. The standard accuracy required for wave direction measurement of NDBC buoys is only up to 10° [47]. Large bias in measured direction of maximum directionally

resolved wave power was seen in Fig. 6, especially in June and December. Therefore, it is important to consider effects of the complexity of wave resource parameters and the uncertainty of measured data when evaluating the model skills for wave resource characterization and assessment.

**3.3. Sensitivity analysis**

One main difference between the WWIII and SWAN models' source term configurations is the different treatment of wave growth and dissipation. The recent development of new physical packages in WWIII, such as ST4 and ST6 physics, has improved model prediction for different growth and dissipation processes, including swell dissipation [17,24,34,38,48]. Sensitivity analysis was conducted to evaluate the effects of the ST4 and ST6 physics package on wave climate.

In the WWIII ST4 physics package, the swell dissipation is esti-mated based on global satellite Synthetic Aperture Radar data and a combination of a viscous and turbulent boundary layer [24]. In the ST4 sensitivity run, the ST4 physics package was turned on in all four WWIII nested grids. However, model configurations for the SWAN-ST4 run remained the same as the baseline stationary run except that open-boundary conditions were specified with outputs from the WWIII-ST4 simulation along the boundary grid points of the L4 domain.

To evaluate the performance of the ST4 physics package in comparison to the baseline simulations with the ST2 physics package, the four error statistics for the six IEC TS wave resources were calculated based on the ST4 WWIII and SWAN simulations and are shown in Table 4. Compared to model performance metrics with the ST2 physics package (Table 3), simulations using WWIII and SWAN with the ST4 physics package show better model skills for predicting wave power $J_{omni}$ and significant wave height $H_s$. Improvement in model accuracy for WWIII-ST4 is clearly seen to be greater than that for SWAN-ST4. This is simply because the ST4 physics package was activated in all four domains for the WWIII-ST4 simulation, while the SWAN-ST4 simulation was only forced with WWIII-ST4 outputs along the L4 domain boundary. The model skills using ST4 for predicting wave energy period $T_e$, however, were worse than those using ST2. The RMSEs for energy period were increased by about 25%, from below 1 s to greater than 1.2 s. Error statistics for the other three resource parameters ($\in_0$, $\theta$ and $d\theta$) are about the same for the ST2 and ST4 physics packages

Please cite this article in press as: Z. Yang, et al., A wave model test bed study for wave energy resource characterization, Renewable Energy (2017), http://dx.doi.org/10.1016/j.renene.2016.12.057

[Figure]

**Fig. 7.** Scatter plots of the six IEC observed and modeled wave resource parameters using the WWIII model with the ST2 physics package. The red line is the linear regression. The black dash line represents the locus of where predicted and observed values are the same. (For interpretation of the references to colour in this figure legend, the reader is referred to the web version of this article.)

(Tables 3 and 4).

A sensitivity run with WWIII using ST6 physics was also conducted. Error statistics for model simulation with WWIII ST6 was calculated and given in Table 4. Clearly, the ST6 physics did not improve the overall model skills for simulating the six IEC TS wave resources parameters compared to the baseline simulation with ST2 physics. Similar to ST4 physics, ST6 predicted better omnidirectional wave power than ST2. However, error statistics with ST6

physics for other parameters were slightly worse than those with ST2 and ST4 physics.

To further evaluate the differences in the performance of the ST2 and ST4 physics packages, comparisons of significant wave heights between observed data and modeled results using the ST2 and ST4 physics packages are plotted for July and November 2009, representing the summer and winter conditions, respectively (Fig. 8). In July when the sea state was calm, the differences between ST2 and

Please cite this article in press as: Z. Yang, et al., A wave model test bed study for wave energy resource characterization, Renewable Energy (2017), http://dx.doi.org/10.1016/j.renene.2016.12.057

**Table 3**
Performance metrics for baseline simulations.

| Parameter | Model | RMSE | SI | Bias | R |
|---|---|---|---|---|---|
| $J$ (kW/m) | WWIII | 20.0 | 0.64 | 6.1 | 0.91 |
| | SWAN | 20.0 | 0.63 | 6.5 | 0.91 |
| | SWAN-NS | 19.0 | 0.62 | 6.3 | 0.91 |
| $H_s$ (m) | WWIII | 0.42 | 0.19 | 0.16 | 0.94 |
| | SWAN | 0.45 | 0.20 | 0.19 | 0.94 |
| | SWAN-NS | 0.44 | 0.20 | 0.18 | 0.94 |
| $T_e$ (s) | WWIII | 0.98 | 0.11 | 0.50 | 0.90 |
| | SWAN | 0.96 | 0.11 | 0.51 | 0.91 |
| | SWAN-NS | 0.95 | 0.11 | 0.52 | 0.91 |
| $\varepsilon_0$ (−) | WWIII | 0.07 | 0.20 | 0.01 | 0.68 |
| | SWAN | 0.07 | 0.20 | 0.00 | 0.71 |
| | SWAN-NS | 0.06 | 0.19 | 0.00 | 0.72 |
| $\theta$ (degrees) | WWIII | 22.87 | n/a | −6.87 | 0.74 |
| | SWAN | 22.62 | n/a | −6.65 | 0.74 |
| | SWAN-NS | 22.24 | n/a | −6.62 | 0.75 |
| $d_\theta$ (−) | WWIII | 0.10 | 0.13 | 0.05 | 0.48 |
| | SWAN | 0.10 | 0.12 | 0.04 | 0.55 |
| | SWAN-NS | 0.10 | 0.12 | 0.04 | 0.55 |

**Table 4**
Performance metrics for sensitivity runs with the ST4 and ST6 physics package.

| Parameter | Model | RMSE | SI | Bias | R |
|---|---|---|---|---|---|
| $J$ (kW/m) | SWAN-ST4 | 17.12 | 0.55 | 3.20 | 0.91 |
| | WWIII-ST4 | 16 | 0.51 | 2.0 | 0.92 |
| | WWIII-ST6 | 16 | 0.52 | −1.5 | 0.91 |
| $H_s$ (m) | SWAN-ST4 | 0.43 | 0.19 | 0.07 | 0.93 |
| | WWIII-ST4 | 0.38 | 0.17 | 0.01 | 0.94 |
| | WWIII-ST6 | 0.43 | 0.19 | −0.11 | 0.93 |
| $T_e$ (s) | SWAN-ST4 | 1.20 | 0.13 | 0.79 | 0.89 |
| | WWIII-ST4 | 1.23 | 0.14 | 0.86 | 0.90 |
| | WWIII-ST6 | 1.35 | 0.15 | 0.88 | 0.85 |
| $\varepsilon_0$ (−) | SWAN-ST4 | 0.07 | 0.21 | 0.01 | 0.67 |
| | WWIII-ST4 | 0.07 | 0.20 | 0.01 | 0.65 |
| | WWIII-ST6 | 0.08 | 0.25 | 0.02 | 0.53 |
| $\theta$ (degrees) | SWAN-ST4 | 23.08 | n/a | −7.33 | 0.73 |
| | WWIII-ST4 | 23.44 | n/a | −7.62 | 0.73 |
| | WWIII-ST6 | 29.64 | n/a | −12.49 | 0.57 |
| $d_\theta$ (−) | SWAN-ST4 | 0.10 | 0.12 | 0.04 | 0.51 |
| | WWIII-ST4 | 0.10 | 0.13 | 0.05 | 0.54 |
| | WWIII-ST6 | 0.11 | 0.14 | 0.03 | 0.28 |

ST4 results were small. However, in November, when large swells were present, using the ST4 physics package improved the model skill in predicting the peak wave height and timing of large waves, because of the better ST4 representation of peak frequency. Monthly distributions of RMSEs for $J_{omni}$ and $H_s$ with ST2 and ST4 physics are shown in Fig. 9. Seasonal variations in RMSEs for $J_{omni}$ and $H_s$ are evident; low RMSE values occurred in the summer and high values in the winter. The RMSEs with ST4 simulations are similar to those with ST2 simulations in the summer, but smaller in the winter, indicating that the ST4 physics package performs better at simulating swell growth and dissipation. Cumulative frequency distributions of significant wave heights calculated based on observations and modeled results using the ST2 and ST4 physics packages are presented in Fig. 10. Again, the cumulative frequency distribution when using ST4 is significantly better than ST2 in comparison to field observations, especially for significant wave heights greater than 1.5 m, which once again indicates ST4 physics is better at simulating large waves than ST2 physics.

To better understand the effect of ST4 physics on simulating wave climate in the frequency and directional domain, two-dimensional wave energy spectra were calculated based on observations and model results using ST2 and ST4 physics for July 15, 2009 (Fig. 11) and November 22, 2009 (Fig. 12), respectively. Although at a broader scale distribution patterns of wave energy

spectra with ST2 and ST4 physics are consistent with observations, dominant energy propagates from the northwest in July and from the southwest in November, and ST4 physics showed overall better performance in producing the wave energy spectra. ST4 physics predicted dominant spectra in the direction ranging from 300 to 350° (Fig. 11b) in July, which is similar to the observation in the range from 300 to 360° (Fig. 11c), while ST2 physics predicted the wave spectra in a narrower range from 315 to 340° (Fig. 11a). ST2 physics also overpredicted the peak frequency at 0.21 Hz, while ST4 and observations both showed a peak frequency below 0.2 Hz. Similarly, ST4 showed greater skill in predicting the direction range of peak spectra and the peak magnitude in comparison to observations in November (Fig. 12).

Although the configurations of frequency and direction resolution specified in the baseline simulations met the minimum requirement recommended in the IEC TS [1], higher spectral resolution, both in frequency and direction, generally has the potential to improve model prediction of energy advection from swell over long distances. In particular, complex geometry and bathymetry in shallow-water regions will alter the frequency-directional characteristic of incoming waves. However, increasing the spectral resolution in frequency and direction will also proportionally increase the computational time. Therefore, it is useful to assess the balance of model prediction accuracy versus computational cost.

Sensitivity model runs were conducted using the WWIII model by varying the number of frequency and direction bins to evaluate the effect of frequency and direction resolutions on the accuracy of wave prediction. The number of frequency bins was increased from 29 to 50; with a 1.07 logarithmic increment factor and a minimum frequency of 0.035 Hz, the maximum frequency is nearly 1 Hz at 0.96 Hz. The number of directional bins was increased from 24 (15-degree resolution) to 36 (10-degree resolution).

A time-series comparison of the predicted significant wave height from WWIII and SWAN models with finer spectral resolution results from the baseline model, and observations at NDBC Buoy 46050 show that increasing the spectral resolution provides no improvement in WWIII and SWAN model skill. Model insensitivity to spectral resolution in this study was likely due to the large water depth (128 m) and the absence of any bathymetric or geometric features at the point of comparison (NDBC Buoy 46050). However, wave models may be sensitive to spectral resolutions at other locations that have complex geometry features and shallow-water depths.

**3.4. Computation efficiency**

One of the common challenges in modeling is the significant computational resources needed to perform model simulations of wave climates at a high resolution. Because the WWIII and SWAN models use very different numerical schemes, their requirements for computational platforms and simulation times are also different. The SWAN simulation requirements assume 16-core CPU platforms, which are widely available at a reasonable cost. SWAN simulations in the present study were performed using 16-core RHEL 6.4 Linux-based operating system platforms with Intel Xeon E7-4880 processors rated at 2.5 GHz clock speeds, with 37.5 MB of L3 Cache, and 1 TB of RAM. The WWIII simulations were performed using 8 nodes on a supercomputer consisting of 692 nodes. Each node is dual socket with 16 cores per socket and an AMD Interlagos processor running at 2.1 GHz with 64 GB of 1600 MHz memory per node (2 GB/socket).

Table 5 provides a summary of the computational times for WWIII and SWAN in both stationary and non-stationary modes. These results indicate that the computation time requirement for WWIII is significantly greater than it is for SWAN in stationary

[Figure]

**Fig. 8.** Comparisons of the observed and modeled significant wave heights at Buoy 46050 using the WWIII ST2 and ST4 physics packages for July (a) and November (b) 2009.

[Figure]

**Fig. 9.** Monthly RMSEs of omnidirectional $J_{omni}$ (a) and significant wave height $H_s$ (b) of WWIII simulations with ST2 and ST4 physics at Buoy 46050.

mode, although WWIII runs included all four-nested grids. Even performing runs on a cluster using 256 cores, it took more than 5 days to complete a 1-year simulation using WWIII. In contrast, SWAN, in stationary mode, took less than 2 days to complete a 1-year simulation on a 16-core CPU platform, which shows great efficiency over WWIII. CPU hours increased significantly, nearly 19 times, when modeling the unsteady-state term in the action balance equation using the non-stationary mode of SWAN. As expected, a significant increase in CPU hours was observed when increasing the spectral resolution, proportional to the ratio of increase in resolution.

**4. Summary and conclusions**

A wave model test bed off the central Oregon Coast was established to evaluate the performance of third-generation, phase-averaged spectral wave models and different modeling approaches for simulating the six wave energy resource parameters recommended by IEC TS [1]. The overarching goal of the test bed study was to provide industry with guidance for model selection and modeling best practices, depending on the wave site conditions and desired class of wave resource assessment. This paper presents the results from the initial effort of the test bed study to evaluate two of the most widely used third-generation spectral models, WWIII and

[Figure]

**Fig. 10.** Cumulative frequency distributions of significant wave heights derived from observations and WWIII simulations at Buoy 46050.

**Table 5**
Summary of WWIII and SWAN computational times for the baseline simulation.

| Model Run | Description | Clock Time | Total CPU-hour |
|---|---|---|---|
| WWIII[b] | Baseline | 5.1 days | 31,488 |
| SWAN-S[a] | Baseline, Stationary | 1.9 days | 731 |
| SWAN-NS[a] | Baseline, Non-stationary | 35.3 days | 13,572 |

[a] Model run time on a 16-core CPU platform.
[b] Model run time on a cluster using 256 cores (8 nodes with 32-core per node).

SWAN, and the nested-grid modeling approach that uses a structured-grid framework.

A nested structured-grid approach, with three levels of outer grids, was employed to provide open-boundary conditions for the test bed domain (L4). The three outer grids included the global domain as the outermost grid (L1), the Pacific Northwest coastal region as the second-level outer grid (L2) nested within L1 grid, and the central Oregon Coast as the third outer grid (L3) nested within L2 grid. For WWIII simulations, all four-level models were two-way nesting and run at the same time. The SWAN model for the test bed domain was driven by WWIII outputs from the L3 domain (one-way nesting). The four-level nested-grid modeling framework, using NOAA's WWIII global model as the lowest-level model, provides an accurate and efficient approach that is suitable for the feasibility class resource assessment at a spatial resolution of ~300 m.

Model performance was evaluated using standard performance metrics based on a comparison of predictions of six IEC wave resource parameters derived by model hindcasts to those derived from buoy measurements. Comparisons of baseline model results derived from WWIII and SWAN to observed data for the six IEC parameters indicate good agreement between model simulations and buoy measurements. Differences between WWIII and SWAN predictions at the NDBC Buoy 46050 location were negligible. Better representations of growth and dissipation using ST4 physics in the WWIII model generally improved model performance. Notably, model skill for predicting omnidirectional wave power density and significant wave height for large waves, which are important for wave resource assessment, was significantly improved with ST4 physics. In contrast, model skill for predicting energy period was slightly reduced with the ST4 physics. Sensitivity analysis with WWIII ST6 physics indicated that the ST6 physics did not improve the model performance in predicting the six IEC TS wave resource parameters at the test bed site. Therefore, use of ST4 physics is recommended for wave resource assessment, even if the WWIII model is used to provide open-boundary conditions.

Sensitivity analysis also indicated that increasing the spectral resolution in both frequency and direction domains provided no improvement in WWIII or SWAN model skills, suggesting that in the area with relatively smooth bathymetry and coastlines, spectral resolution with 29 frequency bins and 24 directional bins may be sufficient. However, this insensitivity was likely due to the large water depth (128 m) at the point of comparison and the absence of any complex bathymetric or geometric features in the vicinity.

[Figure]

**Fig. 11.** Distributions of 2D wave energy spectra computed using WWIII with ST2 physics (a), ST4 physics (b), and observation data (c) on July 15, 2009.

[Figure]

**Fig. 12.** Distributions of 2D wave energy spectra computed from WWIII with ST2 physics (a), ST4 physics (b) and observation data (c) on November 22, 2009.

[Figure]

**Fig. 13.** Comparison of observed and CFSR-predicted winds at NDBC buoy station 46002 (a), 46050 (b), and 46027 (c).

One of the most important factors that affect model accuracy is surface wind forcing, particularly for large wave prediction. The PNW coast is dominated by strong winds and swells in the winter months, as shown in Fig. 5. However, CFSR winds are generally underpredicted in the winter, which is likely the cause of underestimate of the swell peaks (Fig. 8). Fig. 13 shows the comparison of observed and CFSR-predicted winds at NDBC 46002 in the deep open ocean at a water depth of 3,368 m, NBDC 46050 at the inner shelf (also in the test bed) at 128 m, and station 46027 in the nearshore at a shallow-water depth of 46 m (see locations in Fig. 1). Clearly, the accuracy of CFSR-predicted wind speed decreases gradually as the locations become close to the shore. CFSR wind speed prediction matches observations the best at NDBC 46002, with a RMSE of 1.31 (m/s) and linear correlation coefficient R of 0.93. The RMSE and R are 1.81 (m/s) and 0.88 for NDBC 46050, and 2.89 (m/s) and 0.77 for NDBC 46027, respectively. Therefore, improvement in wind prediction capability, especially in the nearshore region, is important for wave resource characterization [49,50]. The present study was limited to evaluation of a structured-grid model skill at a deep offshore site in a northwest wave climate with default model settings. Future studies should include the evaluation of model performance in shallow-water environments, unstructured-grid models, and potential improvement in model predictions of wave energy periods for large waves.

**Acknowledgements**

This study was funded by the Wind and Water Power Technologies Office within the Office of Energy Efficiency and Renewable Energy, U.S. Department of Energy under contract DE-AC05-76RL01830 to Pacific Northwest National Laboratory and contract DE-AC04-94AL8500 to Sandia National Laboratories. The authors thank members of a steering committee charged with external review of this study, including Tuba Ozkan-Haller of Oregon State University, who served as chairperson, Bryson Robertson of the Institute of Energy Systems at the University of Victoria, Arun Chawla from NCEP at NOAA, Tim Mundon from Oscilla Power™, Inc., and Levi Kilcher from the National Renewable Energy Laboratory.

**References**

[1] IEC, Marine energy – Wave, Tidal and Other Water Current Converters – Part 101: Wave Energy Resource Assessment and Characterization, International Electrotechnical Commission, Geneva, Switzerland, 2015.

[2] X.L.L. Wang, V.R. Swail, Changes of extreme wave heights in Northern Hemisphere oceans and related atmospheric circulation regimes, J. Clim. 14 (10) (2001) 2204–2221.

[3] S.M. Brooks, T. Spencer, A. McIvor, I. Moller, Reconstructing and understanding the impacts of storms and surges, southern North Sea, Earth Surf. Process. Landforms 41 (6) (2016) 855–864.

[4] P.D. Bromirski, D.R. Cayan, Wave power variability and trends across the North Atlantic influenced by decadal climate patterns, J. Geophys. Res. Oceans 120 (5) (2015) 3419–3443.

[5] S. Caires, A. Sterl, 100-year return value estimates for ocean wind speed and significant wave height from the ERA-40 data, J. Clim. 18 (7) (2005) 1032–1048.

[6] I. Grabemann, N. Groll, J. Moller, R. Weisse, Climate change impact on North Sea wave conditions: a consistent analysis of ten projections, Ocean. Dyn. 65 (2) (2015) 255–267.

[7] R. Carballo, M. Sanchez, V. Ramos, F. Taveira-Pinto, G. Iglesias, A high resolution geospatial database for wave energy exploitation, Energy 68 (2014) 572–583.

[8] A. Chawla, D.M. Spindler, H.L. Tolman, Validation of a thirty year wave hindcast using the Climate Forecast System Reanalysis winds, Ocean. Model. 70 (2013) 189–206.

[9] G. García-Medina, H.T. Özkan-Haller, P. Ruggiero, Wave resource assessment in Oregon and southwest Washington, USA, Renew. Energy 64 (2014) 203–214.

[10] A.R. Dallman, V.S. Neary, Characterization of U.S. Wave Energy Converter (WEC) Test Sites: a Catalogue of Met-ocean Data, second ed., Sandia National Laboratories, Albuquerque, New Mexico, 2015.

[11] B. Liang, F. Fan, F. Liu, S. Gao, H. Zuo, 22-Year wave energy hindcast for the China East Adjacent Seas, Renew. Energy 71 (2014) 200–207.

[12] B.R.D. Robertson, C.E. Hiles, B.J. Buckham, Characterizing the near shore wave energy resource on the west coast of Vancouver Island, Canada, Renew. Energy 71 (2014) 665–678.

[13] W.E. Rogers, J.M. Kaihatu, L. Hsu, R.E. Jensen, J.D. Dykes, K.T. Holland, Forecasting and hindcasting waves with the SWAN model in the Southern California Bight, Coast. Eng. 54 (1) (2007) 1–15.

[14] N. Li, K.F. Cheung, J.E. Stopa, F. Hsiao, Y.L. Chen, L. Vega, P. Cross, Thirty-four years of Hawaii wave hindcast from downscaling of climate forecast system reanalysis, Ocean. Model. 100 (2016) 78–95.

[15] WAMDI, The WAM model – a third generation ocean wave prediction model, J. Phys. Oceanogr. 18 (12) (1988).

[16] SWAN, SWAN: User Manual, Cycle III Version 41.01A, Delft University of Technology, Delft, The Netherlands, 2015.

[17] H.L. Tolman, WAVEWATCH III Development Group, User Manual and System Documentation of Wavewatch III® Version 4.18, National Oceanic and Atmospheric Administration, National Weather Service, National Centers for Environmental Prediction, College Park, MD 20740, 2014, p. 311.

[18] M. Benoit, F. Marcos, F. Becq, TOMAWAC: a prediction model for offshore and nearshore storm waves, in: Environmental and Coastal Hydraulics: Protecting the Aquatic Habitat, Proceedings of Theme B, Vols 1 & 2, 1997, pp. 1316–1321, 27.

[19] DHI, Mike 21 Spectral Waves FM – Short Description, DHI, Horsholm Denmark, 2012, p. 16.

[20] EPRI, Mapping and assessment of the United States ocean wave energy resource, in: EPRI 2011 Technical Report to U.S. Department of Energy, Electric Power Research Institute, Palo Alto, California, 2011.

[21] P. Lenee-Bluhm, R. Paasch, H.T. Ozkan-Haller, Characterizing the wave energy resource of the US Pacific Northwest, Renew. Energy 36 (8) (2011) 2106–2119.

[22] G. García-Medina, H.T. Özkan-Haller, P. Ruggiero, J. Oskamp, An inner-shelf wave forecasting system for the U.S. pacific northwest, Weather Forecast. 28 (3) (2013) 681–703.

[23] N. Booij, R.C. Ris, L.H. Holthuijsen, A third-generation wave model for coastal regions – 1. Model description and validation, J. Geophys. Res. Oceans 104 (C4) (1999) 7649–7666.

[24] F. Ardhuin, B. Chapron, F. Collard, Observation of swell dissipation across oceans, Geophys. Res. Lett. 36 (2009).

[25] H.L. Tolman, A new global wave forecast system at NCEP, Ocean Wave Meas. Anal. 1 and 2 (1998) 777–786.

[26] A. Chawla, H.L. Tolman, V. Gerald, D. Spindler, T. Spindler, J.H.G.M. Alves, D.G. Cao, J.L. Hanson, E.M. Devaliere, A multigrid wave forecasting model: a new paradigm in operational wave forecasting, Weather Forecast. 28 (4) (2013) 1057–1078.

[27] B.C. Liang, X. Liu, H.J. Li, Y.J. Wu, D.Y. Lee, Wave climate hindcasts for the Bohai Sea, Yellow Sea, and East China Sea, J. Coast. Res. 32 (1) (2016) 172–180.

[28] M.A. Hemer, D.A. Griffin, The wave energy resource along Australia's Southern margin, J. Renew. Sustain. Energy 2 (4) (2010) 043108.

[29] G. Chang, K. Ruehl, C.A. Jones, J. Roberts, C. Chartrand, Numerical modeling of the effects of wave energy converter characteristics on nearshore wave conditions, Renew. Energy 89 (2016) 636–648.

[30] N. Guillou, G. Chapalain, Numerical modelling of nearshore wave energy resource in the Sea of Iroise, Renew. Energy 83 (2015) 942–953.

[31] M. Monteforte, C. Lo Re, G.B. Ferreri, Wave energy assessment in Sicily (Italy), Renew. Energy 78 (2015) 276–287.

[32] D. Silva, A.R. Bento, P. Martinho, C.G. Soares, High resolution local wave energy modelling in the Iberian Peninsula, Energy 91 (2015) 1099–1112.

[33] H.L. Tolman, A generalized multiple discrete interaction approximation for resonant four-wave interactions in wind wave models, Ocean. Model. 70 (2013) 11–24.

[34] F. Ardhuin, E. Rogers, A.V. Babanin, J.-F. Filipot, R. Magne, A. Roland, A. van der Westhuysen, P. Queffeulou, J.-M. Lefevre, L. Aouf, F. Collard, Semiempirical dissipation source functions for ocean waves. Part I: definition, calibration, and validation, J. of Physical Oceanography 40 (9) (2010) 1917–1941.

[35] H.L. Tolman, Treatment of unresolved islands and ice in wind wave models, Ocean. Model. 5 (3) (2003) 219–231.

[36] A. Roland, Development of WWM II: Spectral Wave Modelling on Unstructured Meshes, Institute of Hydraulic and Water Resources Engineering, 2009 (Technische Universitat Darmstadt).

[37] H.L. Tolman, D. Chalikov, Source terms in a third-generation wind wave model, J. Phys. Oceanogr. 26 (11) (1996) 2497–2518.

[38] S. Aijaz, W.E. Rogers, A.V. Babanin, Wave spectral response to sudden changes in wind direction in finite-depth waters, Ocean. Model. 103 (2016) 98–117.

[39] M.M. Amrutha, V.S. Kumar, K.G. Sandhya, T.M.B. Nair, J.L. Rathod, Wave hindcast studies using SWAN nested in WAVEWATCH III - comparison with measured nearshore buoy data off Karwar, eastern Arabian Sea, Ocean. Eng. 119 (2016) 114–124.

[40] S. Saha, S. Moorthi, H.L. Pan, X.R. Wu, J.D. Wang, S. Nadiga, P. Tripp, R. Kistler, J. Woollen, D. Behringer, H.X. Liu, D. Stokes, R. Grumbine, G. Gayno, J. Wang, Y.T. Hou, H.Y. Chuang, H.M.H. Juang, J. Sela, M. Iredell, R. Treadon, D. Kleist, P. Van Delst, D. Keyser, J. Derber, M. Ek, J. Meng, H.L. Wei, R.Q. Yang, S. Lord, H. Van den Dool, A. Kumar, W.Q. Wang, C. Long, M. Chelliah, Y. Xue, B.Y. Huang, J.K. Schemm, W. Ebisuzaki, R. Lin, P.P. Xie, M.Y. Chen, S.T. Zhou, W. Higgins, C.Z. Zou, Q.H. Liu, Y. Chen, Y. Han, L. Cucurull, R.W. Reynolds, G. Rutledge, M. Goldberg, The NCEP climate forecast system reanalysis, Bull. Am. Meteorological Soc. 91 (8) (2010) 1015–1057.

[41] S. Saha, S. Moorthi, X.R. Wu, J. Wang, S. Nadiga, P. Tripp, D. Behringer, Y.T. Hou, H.Y. Chuang, M. Iredell, M. Ek, J. Meng, R.Q. Yang, M.P. Mendez, H. Van Den Dool, Q. Zhang, W.Q. Wang, M.Y. Chen, E. Becker, The NCEP climate forecast system version 2, J. Clim. 27 (6) (2014) 2185–2208.

[42] L. Cavaleri, P.M. Rizzoli, Wind wave prediction in shallow-water - theory and applications, J. Geophys. Res. Oceans Atmos. 86 (Nc11) (1981) 961–973.

[43] P.A.E.M. Janssen, Wave-induced stress and the drag of air-flow over sea waves, J. Phys. Oceanogr. 19 (6) (1989) 745–754.

[44] P.A.E.M. Janssen, Quasi-linear theory of wind-wave generation applied to wave forecasting, J. Phys. Oceanogr. 21 (11) (1991) 1631–1642.

[45] J.A. Battjes, J.P.F.M. Janssen, Energy loss and set-up due to breaking random waves, in: 16th Conference on Coastal Engineering, ASCE, Hamburg, Germany, 1978.

[46] S. Hasselmann, K. Hasselmann, J.H. Allender, T.P. Barnett, Computations and parameterizations of the nonlinear energy-transfer in a gravity-wave spectrum .2. Parameterizations of the nonlinear energy-transfer for application in wave models, J. Phys. Oceanogr. 15 (11) (1985) 1378–1391.

[47] NDBC, Handbook of Automated Data Quality Control Checks and Procedures, NDBC Technical Document 09-02, 2009, p. 78.

[48] F. Bi, J.B. Song, K.J. Wu, Y. Xu, Evaluation of the simulation capability of the WAVEWATCH III model for Pacific Ocean wave, Acta Oceanol. Sin. 34 (9) (2015) 43–57.

[49] J.E. Stopa, K.F. Cheung, Intercomparison of wind and wave data from the ECMWF reanalysis interim and the NCEP climate forecast system reanalysis, Ocean. Model. 75 (2014) 65–83.

[50] C.R. Sampson, P.A. Wittmann, E.A. Serra, H.L. Tolman, J. Schauer, T. Marchok, Evaluation of wave forecasts consistent with tropical cyclone warning center wind forecasts, Weather Forecast. 28 (1) (2013) 287–294.

Please cite this article in press as: Z. Yang, et al., A wave model test bed study for wave energy resource characterization, Renewable Energy (2017), http://dx.doi.org/10.1016/j.renene.2016.12.057

---

## Author Comment (AC4) · 2 Jan 2019

Dear Reviewer

Thank you for your feedback and final comment on our paper.

Regards Allahdadi et al

---

## Author Response (AR2)

Dear Reviewer

Thank you again for your very constructive comments. Your comments on revision#1 of the paper are addressed below and the line numbers addressing the modifications in the manuscript are presented. In the manuscript the revisions are highlighted with light blue:

Comments and Responses
* * *
**26 It's rather the shortcomings of various source terms, not only the balance**

Line #26 was modified to reflect the above fact.

**57 Correct reference to Van Vledder et al., 2016.**

The reference was corrected (Ln#56)

**65 As discussed in W007 his new source term was primarily developed for coastal and inland waters with relatively short fetches.**

"Coastal and inland waters"  was  mentioned in Ln#65

**78-#81 This is incorrect. The OMP version of SWAN4120 contains ST6. Please drop this fake argument. Implementation issues can never be a proper argument for not doing something. Please concentrate on scientific/physical arguments. A stronger argument is twofold. There is limited experience with ST6 in coastal waters, and the range of applicability of W007 for larger fetches with inhomogeneous wind fields is poorly known. As said before, the crucial difference between Komen type and W007 whitecapping source term is the use (or not) of a mean wave steepness, which gives modelling errors in multi-peaked seas. Note that W007 did not really test his source term for multi-peaked seas.**

The sentences were modified based on the above comment (Ln #76-80)

**82 please state which complications may arise. This is now too vaguely formulated.**

The sentence was rephrased to avoid confusion Ln#81)

**87 The apparent contradiction is just another argument that no generally applicable source term setting exists requiring tuning. In the North Sea also shallow water effects like bottom friction are a significant component of the total source term balance.**

The above statement was added to make it more clear(Ln#87-90)

**#104** If you know about the gustiness effect, why did you not use it to 'correct' the CFSR winds?

This clarification was added to the manuscript: (Ln#105-107)

  "To be consistent with the practical modeling efforts and real applications, we did not correct the wind field for gustiness. Therefore, in our simulation we used CFSR wind field with 1 hour temporal resolution that cannot include such short term variations."

**#108** Now it becomes clear why you used W007. When you stick to the available SWAN-ADCIRC executable you run into unnecessary problems.

We had to use SWAN-ADCIRC because we needed the ADCIRC ability for domain decomposition of the unstructured grid which is necessary for implementing high performance computing over a very large numbers of computational grids (more than 4,300,000). It is mentioned in page 4 Ln#112 of the revised manuscript.

**#112** what are these?: … each Komen method.. This is not yet introduced. I can imagine choosing delta=0 or delta=1 may be choices.

In the revised manuscript we referred to section 2 for more details on whitecapping formulations : Ln#116-117: "(see section 2 for details of formulations)"
Details on whitecapping parameters are mentioned in section 5, Ln#253-254 of the revised manuscript.

**#116** You should add also the relatively large fetches and instationary large wind fields as an argument. No reason to state that no new form is suggested. You may put that in the discussion.

Was modified according to the comment (Ln#119-121)

**#124** Only using January, although representative for the winter months, is still a bit meager. You can also reverse the argument, because you want to study some features of the wind fields, you choose this month for studying. Moreover, this also fits better in view of the limitations of a fetch-limited approach. This is a stronger argument and better fits in the purpose of this manuscript.

The following statement was added in Ln#129-132 to show what specific features we need to study by choosing January:

"The persistent offshore-ward wind field during this month (Allahdadi et al., 2019) along with large fetch lengths over the modeling region provide an appropriate condition to study different features of wind field and waves including fetch-limited and fully developed sea states."

**134 It is better to discuss the physical aspects, viz. the impact on lower or higher frequencies, see Rogers et al. (2003) for a discussion**

The conclusion from Rogers et al(2002) was mentioned in Ln #141-142

**186 typo: July 2009**
The Typo was fixed  (Ln#193)

**195 typo: wind roses (2 words)**

The typo was fixed (Ln#203)

**208 …was developed… suggest that a new model setup was created. This seems to contradict the statement that you were forced to use an existing SWAN-ADCIRC approach.**

We actually used unstructured swan that is coupled to ADCIRC for domain decomposition used for high performance computing. We added the clarification to section 4, Ln#215-216

**216 unit of bias is wrong, should be m/s. Lower panels in Figure 4 shows systematic trend of under-prediction with higher wind speeds. This should be noted, as well as the question whether you corrected for this or not.**

Unit of bias was fixed (Ln#224)
Underestimation trend of the CFSR for higher win speeds and the statements that "no correction" was applied to the wind field are mentioned  Ln#225-228

**218 before -> around**
Fixed(Ln#228)

**218 Make the buoy position markers and text larger for visibility**

Fixed. See the modified Figure 5

**222 timestep is the wrong word here. A time step is an interval, whereas here you mean a moment in time. Please rephrase.**

All "timesteps"  words related to this matter were changed to "time"

**236 which two approaches? From earlier remarks I count 3. One W007 and two Komen-type formulations?. If only one Komen is used, what delta is used?**

Only two approaches were used as mentioned in section 2: Komen (the default SWAN approach), and Westhuysen as a Saturated-based method ( as mentioned in section 2). The default values for both method is used. So, the delta regarding the Komen method is 1. The default parameters were added to section 5, Ln #253-254

**238 as defaults may change over time (as they did), you can better explicitly state the settings used.**

The default parameters were explicitly mentioned (section 5, Ln #253-254)

**242 It is still unclear which delta is used in the Komen formulation. Please note that the choice of delta may have significant effects.**

Delta value was added to section 5, Ln #253-254)

**252 Crucial pieces of information are missing here. Which type of buoys are used? Over which frequency range have the buoy spectra been integrated? Now it seems that different frequency intervals have been used. In case SWAN table output for Tm02 has been used, the integration is up to 10 Hz (using the prognostic and parametric spectrum). Buoys usually deliver reliable spectra up to 0.5 Hz. This mismatch may cause significant differences for especially Tm02, see section 4.3 Akpinar et al. (2012) for a discussion on this topic. In such a comparison, one should always use the same frequency interval to derive parameters. For Hm0 the effect is often insignificant, but the higher the frequency moment, the larger the effect. This effect may partly explain the under-prediction. Please check carefully, the consequences.**

We have already mentioned the frequency ranges of the buoy data (0.02-0.485) in page 9(Ln#304-305) of the manuscript (highlighted). We agree that the difference between the integration range of the buoy data in the frequency range with the Prognostic range of SWAN may cause some differences for Tm02. We will mention that in the result part of page 8. However, as Akpinar et al (2012) showed, the differences decrease for higher values of Tm02. In fact for measured wave periods larger than three seconds, the differences are generally negligible. Since the measured wave periods during our simulations at all four buoys are larger than three seconds for most cases (Figure 6e-h), we can safely neglect this discrepancy for the wave period comparison. Furthermore, we picked two events with large wave height and period for discussion (times t1 and t2, see table 2 for values).  .We have added some short discussion about this effect and the probable effect on our results in page 8, Ln#267-274.

**271 Note that this under-prediction may also be due to the fact that no calibration was performed.**

That is absolutely right. However, in this paper we only examined the default parameter. In the discussion part (section 6.2) we mentioned about the necessity for recalibration of models for different sea state conditions.

**300 Can you find a reason why Komen spectra are larger than those of W007. Is this due to the use of a mean wave steepness which in the presence of low-frequency wave components results in an over-prediction of the higher frequencies?**

In the manuscript this behavior was attributed to the larger algebraic sum of the source terms from Komen compared to the Westhuysen Whitecapping:

"Komen simulated a larger sum of source terms at the peak frequency and all frequencies below that (Figure 12d). This result is consistent with Figures 10a that shows higher spectral energies at this time by Komen compared to Westhuysen" (page 12, Ln#386-388 )

**309 Author Clyson does not exist. Check reference**

The correct reference is *Kahma and Calkoen, 1992* that was corrected in the manuscript (see Ln#326).

**326 Can you estimate how much wind strength errors contribute to the total prediction error. As noted, there is an inherent prediction error in each source term, but here also the numerical determination of Tm02 may contribute to the total error. Without such a quantitative consideration, the discussion is a bit pointless.**

In this paper the main purpose is comparing the performance of two models and present different reasons for these comparative differences. Since we used the same wind field for simulation using both whitecapping approach this comparison is meaningful. However, determining and quantifying the effect of wind field discrepancies can be a great idea, but needs thorough simulations. We will suggest it as a topic for further researches in the conclusion part. However, based on a calibrated model for the same area(Allahdadi et al., 2019) some preliminary quantifications on contribution of the wind on model discrepancies were presented(please see section 6.1, Ln#343-348)

Figure 9 Add proper name and symbol of wave parameters in legend 'significant' wave height Hm0 , .. mean wave period Tm02

The proper names and symbol were added to Figure caption.

**332 Specify the kind of source terms shown in Figure 11. I guess they are integrated source term magnitudes. This should be explained.**

Yes. They are integrated. Clarification was added in Ln# 354)

Figure 11 This figure seems inconsistent with what I expect in relation to Figure 9.
1) The shape of the spatial distributions seems inconsistent with those in Figure 9.
2) The upper panels have a color band in yellow, whereas this is not present in the lower panels.
3) Using the unit w/m2 is wrong, as these are integrated source terms of the rate of change of wave variance. No multiplication with rho g has to be done.
4)

1.  Figure 9 shows the simulated maps of wave height and period for times t1 and t2. Figure 11 shows the source term variation over the modeling area only for time t1. If you compare panels in Figure 11 with figures 9(a,b) and 9(e,f) that are onlr related to time t1 they would be consistent.
2.  Lack of yellow color in the lower panel(Westhuysen source terms) is due to the fact that the whitecapping and wind input and consequently the quadruplet terms in Westhuysen approach are scaled based on larger factors that results in larger source terms than Komen(have been mentioned in section 6.1).

3. The unit for the source terms was changed to $m^2/s$ (Ln#360)

**336 As mentioned earlier, the source terms are different because the underlying variance density spectra are also different. Still, I agree that for the same spectrum, the trend is similar. This was easily checked for a JONSWAP spectrum.**

Thank you for the confirmation.

**344 This choice is related to keeping the total balance appropriate.**

The statement was added to the text in page 11, Ln#367.

**357 The chosen spectrum and related wcap source terms do not really show some effect for small frequencies. I did a test for a JONSWAP spectrum and then subtle for significant differences popup.**

This is consistent with W007 as the observed the same differences for energy at low frequencies.

**360 A comment on the strange shape of Snl4 is missing**

The comment was added to the bottom of page 11, Ln#378-380.

**368 How can a model be more than reality? Please rephrase**

The sentence was modified (Ln#393-394)

**393 are resulted -> result**

We suspect that "resulted" maybe the right word.

**430 result -> results**
Fixed, Ln#455

**442 Such a distinction will be difficult in practice. This is another indication that these formulations are not generally applicable. This is an interesting conclusion. As noted in Bingolbali, tuning for a large basin may give different results when tuning is done for a local area.**

The phrase " if possible" was added before this statement to show the difficulty of implementing this suggestion (Ln#466-467)

**447 rephrase … too much…, -> occur often?**

Correction was applied ( Ln#472 )

**475 directional spectra cannot be observed, only reconstructed from the Fourier coefficients that can be distilled from the buoys time series. So, state how the buoy spectra were reconstructed**

It was mentioned in the manuscript that the directional specra is "reconstructed from Fourier coefficients" (Ln#504-505)

**489 How do you quantify directional spreading?**
We estimated the directional spreading as "the total angle for which wave energy exists within the scale to 360 degrees". This clarification was added to page 15, Ln#519-520

**468 Note that the quadruplet term here is the crude DIA. See Ardhuin et al. (2007) and Bottema and Van Vledder (2008) for possible effects on slanting fetch situations when using the accurate Snl4.**

The DIA method used in this research and differece with results from the exact method (Xnl) based on Bottena and Vledder(2008)was mentioned in pages 15, Ln#499-503.

**515 This is a welcome addition to the manuscript, leading to a nice conclusion about the inclusion of stability effects in wave modelling.**

Thanks for approving.

**557 calibrated by whom and in what way? This is an important notion**
.
We meant "calibrated models based on growth curve of Kahma and Calkoen(19912)". We added this to page 17 of the manuscript Ln#588-589

**571 The logic here is a bit flawed. Variabilities in the wind field itself are no cause for slanting fetch effects. Please rephrase.**

This phrase was modified and the wind field effect was removed (Ln# 601-602)

**585 What are 'real values'?**

The word "Unrealistic" was used at the beginning of the sentence and "than real value" was removed to avoid complications (Ln# 615)

**588 technical -> physical ?**

We think "scientific" maybe a better word, so used that instead of physical in the page 18 (Ln# 619)

**590 omit the distinction between serial and parallel, that is not relevant**

"serial and parallel" modes were removed (Ln# 620-621)

---

## Author Response (AR3)

Dear Professor Neil Wells

Thank you very much for handling our paper and sending us your constructive comments. The comments are addressed as follows:

Comments to the Author:
lines 87-88 Sentence ambiguous. A contradiction is not argument. Could replace with" This suggests that there is no generally applicable source term. "

The sentence was modified according to the editor's suggestion. Highlighted in lines #87-88 of the revised manuscript

Figure 9 caption line 2
delete "e-h" before" and simulation..."

Fixed. Highlighted in Figure caption.

Figure 11 Colour in lower panels is inconsistent with colour bar below. Where is yellow ?

This comment has also made by reviewer 1 and we responded to it in the previous revision as follows:

*"Lack of yellow color in the lower panel(Westhuysen source terms) is due to the fact that the whitecapping and wind input and consequently the quadruplet terms in Westhuysen approach are scaled based on larger factors that results in larger source terms than Komen(have been mentioned in section 6.1)"*

In plotting the maps for both upper and lower panels, the same codes with the same colorbar and plot configurations have been used and different scaling of the whitecapping formula account for the jump over the yellow color for lower panels. Other types of colorbar show similar pattern  as the present Figure 11.

Non-public comments to the Author:
The reviewer has been very helpful in reviewing your paper and I would appreciate an acknowledgement in your paper to the reviewer.

Reviewers have already been appreciated in the last revision that we submitted. Please see highlighted lines in the acknowledgment part in the new submission.

Best Regards

Allahdadi et al